



# Drivers and implications of declining fossil fuel CO₂ in Chinese cities revealed by radiocarbon measurements

Pingyang Li[1,2], Boji Lin[1,2,3], Zhineng Cheng[1,2], Jing Li[1,2], Jun Li[1,2], Duohong Chen[4,*], Tao Zhang[4], Run Lin[1,2], Sanyuan Zhu[1,2], Jun Liu[4], Yujun Lin[4], Shizhen Zhao[1,2], Guangcai Zhong[1,2], Zhenchuan Niu[5,6], Ping Ding[7], Gan Zhang[1,2,*]

[1] State Key Laboratory of Advanced Environmental Technology, Guangzhou Institute of Geochemistry, Chinese Academy of Sciences, Guangzhou 510640, People's Republic of China
[2] Guangdong Key Laboratory of Environmental Protection and Resources Utilization, and Joint Laboratory of the Guangdong-Hong Kong-Macao Greater Bay Area for the Environment, Guangzhou Institute of Geochemistry, Chinese Academy of Sciences, Guangzhou 510640, People's Republic of China
[3] University of Chinese Academy of Sciences, Beijing 100049, People's Republic of China
[4] Environmental Key Laboratory of Regional Air Quality Monitoring, Ministry of Ecology and Environment, Guangdong Ecological and Environmental Monitoring Center, Guangzhou 510308, People's Republic of China
[5] State Key Laboratory of Loess, Institute of Earth Environment, Chinese Academy of Sciences, Xi'an 710061, People's Republic of China
[6] Institute of Global Environmental Change, Xi'an Jiaotong University, Xi'an 710061, People's Republic of China
[7] State Key Laboratory of Deep Earth Processes and Resources, Guangzhou Institute of Geochemistry, Chinese Academy of Sciences, Guangzhou 510640, People's Republic of China

*Correspondence to*: Gan Zhang (zhanggan@gig.ac.cn), Duohong Chen (chenduohong@139.com)

**Abstract.** China's clean air policies have successfully mitigated fossil fuel CO₂ (CO₂ff or Cff) emissions in bottom-up inventories since 2013. Yet, evidence from top-down measurements and their underlying drivers remains lacking. Here, we quantify Cff concentrations and fuel-specific contributions using atmospheric $\Delta(^{14}CO_2)$ and $\delta(^{13}CO_2)$ measurements across representative Chinese cities. We found distinct regional trends: megacities like Guangzhou show significant Cff declines (35 % decrease from 2011 to 2022) along with their source regions, while smaller cities have yet to demonstrate similar reductions. These improvements can be attributed to a 23 % coal consumption reduction, 17 % increased natural gas use (evidenced by stable isotope analysis), and improved combustion efficiency (indicated by 63 % falling $R_{CO/CO2ff}$ ratios). Notably, the three-decade observational record shows steeper declines in urban $R_{CO/CO2ff}$ ratios than inventory estimates, suggesting current emission inventories may underestimate combustion efficiency improvements and CO emission reductions relative to Cff mitigations. These findings indicate nationwide progress toward Cff emission peaks, with megacities leading the transition. They also underscore how coal-to-gas transitions and technological upgrades simultaneously advance air quality and climate goals. Importantly, our results highlight the critical need to integrate top-down observational frameworks (e.g. radiocarbon measurements) with traditional inventories to better capture rapid, policy-driven emission changes and inform future co-benefit optimization strategies.



## 1 Introduction

As the world's largest energy consumer, China's heavy reliance on fossil fuels has resulted in severe air pollution and substantial fossil fuel $CO_2$ ($CO_{2ff}$ or $C_{ff}$) emissions, accounting for 31 % of global fossil $CO_2$ emissions in 2022 (Friedlingstein et al., 2023b). These emissions pose critical threats to public health and ecological stability. In response, China has enacted progressive policies including the 2013 Clean Air Action Plan (Zheng et al., 2018; Zhang et al., 2019), 2018 Blue Sky Defense Battle, and 2022 Pollution-Carbon Synergy Plan, achieving co-benefits in air quality improvement and $C_{ff}$ mitigation as quantified through bottom-up inventories like Multi-resolution Emission Inventory for China (MEIC) (Shi et al., 2022). However, the effectiveness of these policies in reducing atmospheric $C_{ff}$ concentrations, and the underlying drivers of these reductions, remains unverified and unexplored through top-down observational approaches, creating a critical knowledge gap in climate policy assessment.

Bottom-up inventories and top-down measurements are approaches commonly used to determine atmospheric $C_{ff}$ emissions, but each has inherent limitations that can affect accuracy and reliability. Although bottom-up inventories are available at increasingly higher spatiotemporal resolution (Han et al., 2020), they are time-consuming to compile and update promptly, often lack quantitative estimation of uncertainty (Lo Vullo and Monforti, 2019), and frequently debated in attributing emissions to specific sources (Gurney et al., 2021). In contrast, top-down studies encompass all existing sources within a geographic region but struggle to achieve accurate partitioning of the fossil fuel and biospheric $CO_2$ contributions. This methodological impasse can be resolved by $^{14}C$ analysis, which exploits the unique $^{14}C$-depletion signature of $C_{ff}$ compared to contemporary biogenic sources (Levin et al., 2003; Turnbull et al., 2006), enabling unambiguous fossil fuel emission quantification.

Urban areas, occupying merely 3 % of global land yet responsible for 75 % of global $C_{ff}$ emissions (reaching 80 % in China) (Dhakal, 2009; Duren and Miller, 2012), represent strategic priorities for emission mitigation. Recent advances in analytical tools can help identify key drivers of urban $C_{ff}$ reductions. $\delta(^{13}CO_2)$ signatures successfully distinguished coal, oil, and natural gas contributions in cities like Beijing and Xi'an (Wang et al., 2022b), while $\Delta CO/C_{ff}$ ratios tracked combustion efficiency variations across national (China, South Korea) and urban (Paris, Heidelberg) scales (Turnbull et al., 2011; Lee et al., 2020; Lopez et al., 2013; Rosendahl, 2022). To address the research gaps mentioned above, we performed spatiotemporal mapping of 2022 $C_{ff}$ concentrations across representative Chinese cities using dual-carbon isotope constraints ($\Delta(^{14}CO_2)$ + $\delta(^{13}CO_2)$) for fuel-specific source attribution. By integrating multi-source inventories with extended $\Delta CO/C_{ff}$ observations through 2022, we developed a robust framework for top-down verification of policy-driven emission reductions. Our methodology not only quantifies $C_{ff}$ concentration decreases but also identifies the key mechanisms behind these reductions, offering critical insights for refining climate mitigation strategies and supporting sustainable urban development.





## 2 Data and methods

### 2.1 Sample collection

We selected representative Chinese cities of varied population sizes for this study: Guangzhou, Shenzhen, and Beijing for megacities (urban permanent resident populations >10 million), Xi'an for supercities (5−10 million), Zhanjiang for large cities (1−5 million), and Shaoguan for medium and small cities (<1 million), which is retrieved from the Tabulation on 2020 China Population Census by County (Council, 2022). Since we could obtain results in Beijing and Xi'an from previous studies, we conducted field sampling in the four cities in Guangdong Province, China (Fig. 1). Guangdong Province is located south of the Nanling Mountains and on the coast of the South China Sea, lying within subtropical and tropical low-latitude regions. The area experiences a prevailing southeast monsoon from the ocean during summer and a northeast monsoon from the continent during winter. The four cities in Guangdong Province differ in terms of area, population, gross domestic product (GDP), $C_{ff}$ inventory emissions, population density, topographic elevation, and land use/land cover. Guangzhou and Shenzhen represent two of China's seven megacities — approximately 45 exist globally — within the Pearl River Delta (PRD), the world's largest urban agglomeration. Guangzhou, the capital of Guangdong Province, has a population of 18.7 million, GDP of 2 884 billion Yuan, and built-up area covering 35.2 %. Shenzhen, a high-tech hub transformed by post-1978 reforms, hosts 17.7 million people with GDP reaching 3 239 billion Yuan and 53.8 % built-up coverage. In contrast, Zhanjiang (large city) and Shaoguan (medium and small city) have smaller populations — 7.0 million and 2.9 million respectively — and lower GDPs of 371.3 billion Yuan and 156.4 billion Yuan. Zhanjiang features extensive cultivated land (47.8 %) and coastal ports, while Shaoguan is distinguished by 78.4 % vegetation coverage.

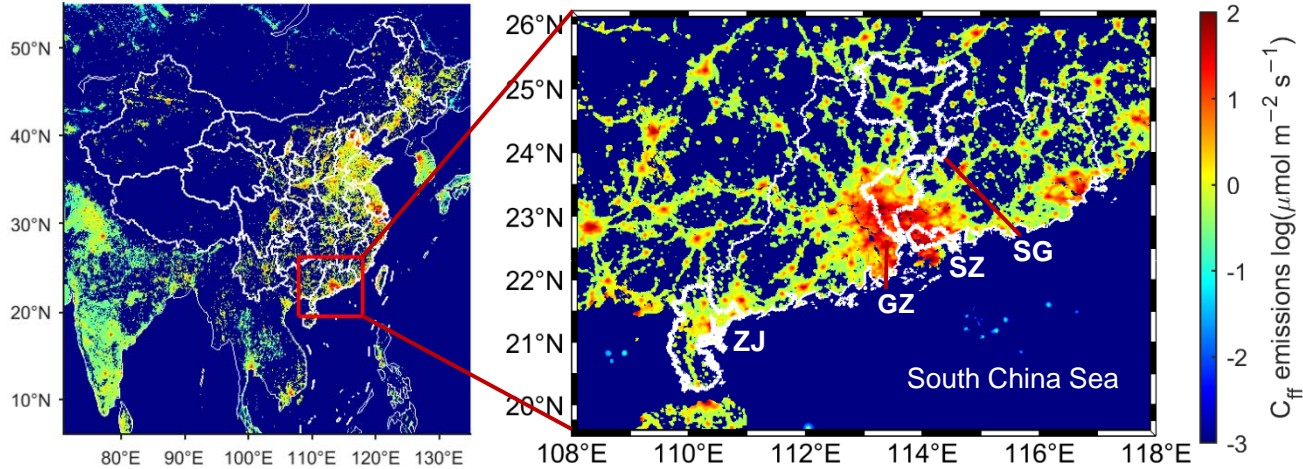







**Figure 1: Locations of sampling sites and spatial distribution of $C_{ff}$ concentrations during summer (s) and winter (w) in (a and b, GD) Guangdong Province and the cities of (c and d, GZ) Guangzhou, (e and f, SZ) Shenzhen, (g and h, ZJ) Zhanjiang, and (i and j, SG) Shaoguan. The colored circles represent the observations, while the shading indicates the $C_{ff}$ inventory emissions from the Open-source Data Inventory for Anthropogenic CO₂ (ODIAC) (Oda and Maksyutov, 2024) in August and December with 1×1 km grid spacing. White lines indicate boundaries of cities in Guangdong Province (a and b), and boundaries of districts in the four cities (c−j). In (a) and (b), bold white lines indicate boundaries of nine cities of the Pearl River Delta. The left color bar represents the $C_{ff}$ inventory emissions, while the right color bar represents the $C_{ff}$ observations.**

We collected 240 air samples from 30 sites during summer (28 July − 30 August 2022) and winter (12 December 2022 − 6 January 2023) seasons, with weekly sampling in both campaigns. The locations and details of these sampling sites are given





in Fig. 1 and Table A1. Ten sampling sites were located in Guangzhou (GZ1–GZ10), ranging from urban downtown to suburban areas, based on distributed $C_{ff}$ emissions from the Open-source Data Inventory for Anthropogenic $CO_2$ (ODIAC) (Oda and Maksyutov, 2011; Oda and Maksyutov, 2024). Another 10 sampling sites were distributed uniformly throughout Shenzhen (SZ1–SZ10). In Zhanjiang (ZJ1–ZJ5) and Shaoguan (SG1–SG5), five sampling sites were selected in each city,

primarily in urban areas, and distributed according to the first and second most dominant wind directions. These sites are located on the tower or on the roof of the building with 10–12 m extendable masts and are chosen to be free from any modifying effects of surrounding skyscrapers. Most of our sampling sites are generally no more than 300 m from the nearest air quality monitoring station. The sampling height is usually kept above 30 m above the ground level to avoid the influence of local sources. We assume that the measurements at the sampling sites in Guangzhou and Shenzhen are statistically

sufficient to assess the whole cities, while those in Zhanjiang and Shaoguan are sufficient to assess the urban areas. Air sampling occurred between 13:00 and 17:00 local time, coinciding with the deepest planetary boundary layer and well-mixed atmospheric conditions. Post-filtration samples were transferred into pre-evacuated/flushed 3 L borosilicate flasks using 12 V micro-diaphragm pumps. These delivered a flow rate of 6 L min$^{-1}$ at 25 °C and 101.3 kPa, with pressurization to 172.4–206.8 kPa. The duration of the sampling was approximately 15–20 min in total.


## 2.2 Measurement of atmospheric $CO_2$ and $\delta(^{13}C)$

Whole-air samples were dried using magnesium perchlorate at a constant flow rate of 25 mL min$^{-1}$, controlled by a mass flow controller. The $CO_2$ concentrations and $\delta(^{13}C)$ values were then measured using a Picarro G2201-i high-precision carbon isotope analyzer (Picarro, Inc., Santa Clara, CA, USA) with cavity ring-down spectroscopy. Each sample was

measured for 10 min, and only data from the final 5 min were used to calculate the average $CO_2$ concentration and $\delta(^{13}C)$ value. Calibration for the $CO_2$ concentrations and the $\delta(^{13}C)$ values was performed using the method described by Wen et al. (2013) with three standards: (a) (409.47 ± 0.02) ppm (µmol mol$^{-1}$; similar hereafter), (−8.717 ± 0.013) ‰; (b) (447.78 ± 0.01) µmol mol$^{-1}$, (−9.759 ± 0.006) ‰; and (c) (503.65 ± 0.01) µmol mol$^{-1}$, (−11.456 ± 0.004) ‰, obtained from the Chinese Academy of Meteorological Sciences. The $CO_2$ concentrations of the standards are traceable to the X2019 standard scale

maintained by the Central Calibration Laboratory of the World Meteorological Organization, and the $\delta(^{13}C)$ values are traceable to the stable isotope laboratory of the Institute of Arctic and Alpine Research based on the NBS-19 and NBS-20 standards. The $\delta^{13}C$ values were reported relative to the international Vienna Pee Dee Belemnite standard (Coplen, 1996). The precision was better than 0.2 µmol mol$^{-1}$ for $CO_2$ concentrations and 0.1 ‰ for $\delta(^{13}C)$ values.

## 2.3 Measurement of atmospheric $\Delta(^{14}C)$

The residual air samples were transferred into a vacuum system at a flow rate of 300 mL min$^{-1}$. It was then first passed through a cold trap consisting of dry ice and ethanol slurry to freeze out water, followed by passage through a liquid nitrogen



cold trap (−196 °C) to freeze down the $CO_2$ (Xu et al., 2007). The extracted and purified $CO_2$ was converted into graphite using the hydrogen reduction method. The graphite was then pressed into an aluminum holder for $^{14}C$ measurements using

an NEC 0.5MV 1.5SDH-2 accelerator mass spectrometer (AMS, National Electrostatics Corporation, USA) (Zhu et al., 2015). Each measurement wheel typically comprises 13 primary standards (oxalic acid II), 13 secondary standards (IAEA-C7), 13 solid process blanks (*p*-phthalic acid), 6 gas process blanks ($^{14}C$-free $CO_2$ in synthetic air from a cylinder), and some authentic air samples. The results are presented as $\Delta(^{14}C)$, which is the per mill (‰) deviation from the absolute radiocarbon reference standard, corrected by fractionation and decay (Stuiver and Polach, 1977). We analyzed 17 pairs of parallel air

samples to evaluate the quality control and assurance of the entire sampling and laboratory analysis process, including sampling, extraction, graphitization, and AMS measurement. The AMS measurement uncertainty and the average deviation are (2.1 ± 0.3) ‰ and (0.2 ± 2.9) ‰, respectively (see Fig. A1). We thus specify a one-sigma measurement uncertainty of 2.9 ‰ for $\Delta(^{14}C)$ based on these repeat measurements of air samples.

**2.4 $C_{ff}$ concentration estimation (incorporated biomass burning emissions)**

Recently added atmospheric $CO_2$ ($CO_{2obs}$ or $C_{obs}$) is thought to consist of background $CO_2$ ($CO_{2bg}$ or $C_{bg}$) and excess $CO_2$ ($CO_{2xs}$ or $C_{xs}$). The $C_{xs}$ mainly includes $C_{ff}$ and biogenic $CO_2$ ($CO_{2bio}$ or $C_{bio}$). The corresponding $\Delta(^{14}C)$ values are expressed as $\Delta_{obs}$, $\Delta_{bg}$, $\Delta_{ff}$ (−1000 ‰, zero $^{14}C$ content), and $\Delta_{bio}$, respectively. The mass balance equations for atmospheric $CO_2$ and $\Delta(^{14}C)$ are expressed as follows (Levin et al., 2003):

$$C_{obs} = C_{bg} + C_{xs} = C_{bg} + C_{ff} + C_{bio} \qquad (1)$$

$$C_{obs}\Delta_{obs} = C_{bg}\Delta_{bg} + C_{ff}\Delta_{ff} + C_{bio}\Delta_{bio} \qquad (2)$$

$$C_{ff} = \frac{C_{obs}(\Delta_{bg}-\Delta_{obs})}{\Delta_{bg}+1000‰} - \frac{C_{bio}(\Delta_{bg}-\Delta_{bio})}{\Delta_{bg}+1000‰} = \frac{C_{obs}(\Delta_{bg}-\Delta_{obs})}{\Delta_{bg}+1000‰} - \beta \qquad (3)$$

The added $C_{ff}$ component is determined using Eq. (3). The $CO_2$ and $\Delta(^{14}C)$ from other sources, such as air-sea exchange (Graven et al., 2018) and nuclear facilities (see Appendix C1), have been neglected owing to their relatively small amounts.

The second term (β) represents a disequilibrium correction for the effect of $CO_2$ sources from biospheric exchange with distinct $\Delta(^{14}C)$ signatures relative to atmospheric values, primarily attributed to heterotrophic respiration (Rh) and biomass burning (BB). We quantified β using integrated modeling frameworks (see Appendixes B and C). The heterotrophic respiration correction ($\beta_{Rh}$) was derived from FLEXPART simulations (Pisso et al., 2019) with CASA-GFED4s data (Randerson et al., 2017; Van Der Werf et al., 2017), yielding values of (−0.06 ± 0.03) µmol mol$^{-1}$ in summer and (−0.11 ±

0.04) µmol mol$^{-1}$ in winter. The biomass burning corrections ($\beta_{BB}$) was calculated under two assumptions: (1) $\Delta(^{14}C)$ endmembers assume 100 % perennial biomass, and (2) $C_{BB}$ emissions represent 100 % of $C_{bio}$ in EDGAR2024 (covering open and closed combustion) (Edgar, 2024). $\beta_{BB}$ showed maximum values of (–0.09 ± 0.08) µmol mol$^{-1}$ during summer and (–0.24 ± 0.12) µmol mol$^{-1}$ during winter. The combined correction (β = $\beta_{Rh}$ + $\beta_{BB}$) displayed seasonal patterns: (–0.16 ± 0.09) µmol mol$^{-1}$ in summer and (–0.35 ± 0.15) µmol mol$^{-1}$ in winter, contrasting with (−0.5 ± 0.2) µmol mol$^{-1}$ during



summer and (−0.2 ± 0.1) µmol mol⁻¹ during winter reported by Turnbull et al. (2009). Notably, this study is the first to integrate BB emissions into $C_{ff}$ estimation frameworks. BB has a greater impact than Rh on disequilibrium corrections, likely due to inventory and assumption differences. While the total corrections align broadly with literature values, the seasonal trends differed inversely from past studies. We applied literature-based corrections to maintain methodological consistency, and the simulated disequilibrium terms further support our conclusions.


### 2.5 $C_{ff}$ footprint by FLEXPART model

Surface flux sensitivity of $C_{ff}$ were conducted using the FLEXible PARTicle (FLEXPART) dispersion model (version 10.4) (Pisso et al., 2019). The model produced source−receptor relationships, often referred to as "footprints" for atmospheric surface measurements, which represent the response of the observations at a measuring station to a source emission. The

footprints are calculated driven by global meteorological fields from the National Centers for Environmental Prediction's Climate Forecast System (CFSv2) Reanalysis model (Saha et al., 2011). They are computed by releasing 10 000 virtual particles from each receptor at each sampling time and tracking them backward for 30 days over the domain of 0°–60° N, 70° E–150° E, with resolution of 0.1°×0.1°.

### 2.6 Fuel-specific fractions of $C_{ff}$ by Keeling plot and Bayesian mixing model

The method to determine coal, oil, and natural gas (i.e., fossil fuel type) fractions of $C_{ff}$ is described briefly using a Keeling plot (Miller and Tans, 2003) and the Bayesian mixing model (MixSIAR) (Stock et al., 2018). We calculated the excess $\delta(^{13}C)$ (intercepts $\delta_{xs}$, Eq. (4)) above the background level based on the best-fit lines in the Keeling plot. To determine the $\delta(^{13}C)$ of the fossil fuel source ($\delta_{ff}$, Eq. (5)), we estimated the weighted averages of the fossil fractions $F_{ff}$ using a two end-member

mixing analysis on $C_{xs}$. The $\delta(^{13}C)$ of the biogenic source ($\delta_{bio}$) was set to −26.1 ‰, which is the average $\delta(^{13}C)$ value of the background air plus the −16.8 ‰ discrimination by the terrestrial ecosystem (Bakwin et al., 1998). We then estimated the coal, oil, and natural gas fractions of $C_{ff}$ ($F_{coal}$, $F_{oil}$, and $F_{ng}$, Eqs. (6) and (7)) using a Bayesian tracer mixing model framework implemented as an open-source R package. The model used the $\delta_{ff}$ values as mixing data and the end-member $\delta(^{13}C)$ signatures of coal, oil, and natural gas as the source data.

We adopted the end-member $\delta(^{13}C)$ signatures measured in Beijing: $\delta_{coal}$ = −24.3 ‰, $\delta_{oil}$ = (−28.9 ± 0.5) ‰ and $\delta_{ng}$ = (−33.2 ± 0.9) ‰ (Wang et al., 2022a). This selection was based on three considerations: First, coal $\delta(^{13}C)$ signatures exhibit remarkable regional stability in China. Second, oil signatures from the Pearl River Mouth Basin of (−28.1 ± 1.6) ‰ (Cheng et al., 2013) show close agreement with Beijing values of (−28.9 ± 0.5) ‰. Third, measured natural gas signatures like (−33.2 ± 0.9) ‰ in Beijing and (−32.0 ± 0.1) ‰ in Xi'an are significantly enriched compared to literature averages [−39.5 ‰





in Beijing and (−38.9 ± 2.6) ‰ in Pearl River Mouth Basin] (Ping et al., 2018; Quan et al., 2018), as using the lower
literature values would lead to underestimation of natural gas contributions.

$$\delta_{obs} = C_{bg}(\delta_{bg} - \delta_{xs}) \times \frac{1}{C_{obs}} + \delta_{xs} \qquad (4)$$

$$\delta_{xs} = F_{ff}\delta_{ff} + (1 - F_{ff})\delta_{bio} \qquad (5)$$

$$\delta_{ff} = F_{coal}\delta_{coal} + F_{oil}\delta_{oil} + F_{ng}\delta_{ng} \qquad (6)$$

$$1 = F_{coal} + F_{oil} + F_{ng} \qquad (7)$$

### 2.7 Correlation of $C_{ff}$ and CO and its emission ratio

We calculated Pearson correlation coefficient (*r*) between $C_{ff}$ and CO enhancement ($\Delta CO = CO_{obs} - CO_{bg}$), and
observational emission ratio of $\Delta CO$ to $C_{ff}$ ($R_{CO/CO2ff}$) [ppb ppm$^{-1}$ (nmol µmol$^{-1}$; similar hereafter)] using linear least squares
regression. The $R_{CO/CO2ff}$ ratios were derived from the regression slopes of $\Delta CO$ versus $C_{ff}$ concentrations. For comparison,
the inventory emission ratio of CO to $C_{ff}$ ($I_{CO/CO2ff}$) [ppb ppm$^{-1}$ (nmol µmol$^{-1}$)] was calculated as (Lee et al., 2020):

$$I_{CO/CO2ff} = E_{CO}/E_{CO2ff} \times M_{CO2}/M_{CO} \qquad (8)$$

where $E_{CO}$ and $E_{CO2ff}$ represent the total CO and $C_{ff}$ emissions, respectively, in teragrams per year (Tg a$^{-1}$) from the bottom-
up national and urban inventory; and $M_X$ refers to the molar masses of CO and $CO_2$ in grams per mole (g mol$^{-1}$).

## 3 Results and discussions

### 3.1 Background selection

We conducted atmospheric observations of $CO_2$ and its carbon isotope composition ($\Delta(^{14}C)$ and $\delta(^{13}C)$) in Guangzhou,
Shenzhen, Zhanjiang, and Shaoguan in Guangdong Province, South China, during the summer and winter of 2022. To
attribute $CO_2$ enhancements ($C_{xs}$) to a particular region, it is necessary to isolate the component of the observed
concentration attributable to fluxes within the region by removing the background (Karion et al., 2021). High-elevation
mountains, representing the free troposphere, were considered ideal background locations for use in this study (Turnbull et
al., 2009). Specifically, the Nanling site (NL, 1700 m above sea level (a.s.l.)), one of the 30 sampling sites of this study (SG5;
Table A1), was selected because it serves as the nearest regional background site for the study areas with relatively complex
boundary conditions (for more reasons see Appendix D). The "annual" $CO_2$ and $\Delta(^{14}C)$ averages at NL station, calculated as
averages of summer and winter measurements, were (418.5 ± 7.3) µmol mol$^{-1}$ and (−7.1 ± 3.9) ‰, respectively. These
values closely match those observed at Jungfraujoch (JFJ, 3580 m a.s.l.) and appear in the upper-right section of the Keeling
plot of $\Delta(^{14}C)$ and $CO_2$ (i.e., scatter plot between $\Delta(^{14}C)$ and inverse of $CO_2$ mole fractions) representing background
conditions (Pataki et al., 2003). This positioning becomes evident when comparing with Waliguan (WLG, 3890 m a.s.l.)
station data (Fig. 2). The advantage of using the Keeling plot method to screen background data is that it simultaneously



accounts for both higher values of $\Delta(^{14}C)$ and lower values of $CO_2$ (Zhou et al., 2024). The $\Delta(^{14}C)$ averages at NL were the highest among the 30 sampling sites considered in this study, with values of $(-3.7 \pm 1.3)$ ‰ in summer and $(-10.6 \pm 0.8)$ ‰ in winter (Table A1). We selected a regional (rather than urban) background site in 2022 to capture higher $C_{ff}$ concentrations. This approach establishes a lower baseline for analyzing $C_{ff}$ reduction in subsequent sections.

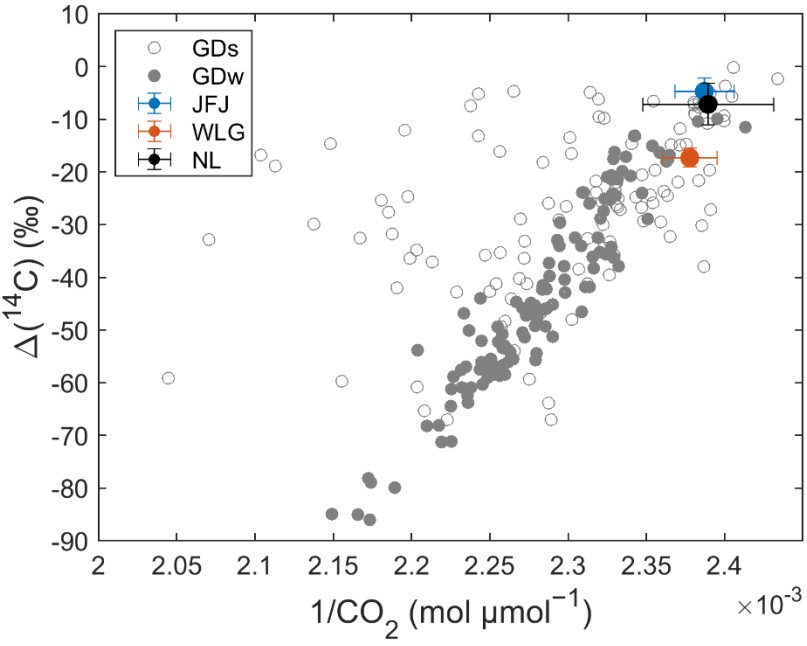

**Figure 2: Keeling plot of $CO_2$ and $\Delta(^{14}C)$ measurements from Guangdong Province in summer (GDs) and winter (GDw), and background stations including JFJ (Jungfraujoch) (Emmenegger et al., 2024b, a), WLG (Waliguan) (Liu et al., 2024; Lan et al., 2023), and NL (Nanling, this study) in 2022. The $CO_2$ concentrations at the WLG station were obtained from the World Data Centre for Greenhouse Gases (WDCGG, https://gaw.kishou.go.jp/, last accessed: April 21, 2024).**

**3.2 $CO_2$, $\Delta(^{14}C)$, $C_{xs}$ and $C_{ff}$ concentrations**

$CO_2$ concentrations in Guangzhou, Shenzhen, Zhanjiang, and Shaoguan were $(438.8 \pm 12.3)$, $(435.0 \pm 12.7)$, $(444.2 \pm 17.2)$, and $(431.6 \pm 10.5)$ µmol mol$^{-1}$ (multisite mean and one-sigma standard deviation), respectively; the corresponding $\Delta(^{14}C)$ values were $(-40.7 \pm 13.4)$ ‰, $(-37.2 \pm 24.1)$ ‰, $(-28.8 \pm 13.8)$ ‰, and $(-25.0 \pm 14.9)$ ‰, respectively. Relative to the background, $CO_2$ concentrations in the four cities were enhanced by $(20.3 \pm 12.5)$, $(16.5 \pm 13.5)$, $(25.8 \pm 16.7)$, and $(13.1 \pm$

$10.1)$ µmol mol$^{-1}$ ($C_{xs}$), respectively; the mean $\Delta(^{14}C)$ was depleted by $(-33.6 \pm 11.3)$ ‰, $(-29.9 \pm 22.3)$ ‰, $(-21.5 \pm 11.7)$ ‰, and $(-17.8 \pm 15.7)$ ‰ ($\Delta\Delta(^{14}C)$), respectively, reflecting the marked influence of $^{14}C$-free $CO_2$ emissions from fossil fuel combustion. The fossil fuel and biogenic fractions of $C_{xs}$, $C_{ff}$ and $C_{bio}$, were determined using a two end-member mixing analysis. The $C_{ff}$ fractions were $(79 \pm 5)$ %, $(73 \pm 6)$ %, $(59 \pm 4)$ %, and $(53 \pm 13)$ % during winter in Guangzhou,



Shenzhen, Zhanjiang, and Shaoguan, respectively. In comparison with other cities worldwide (Table E1, Fig. E1), we observed higher $C_{ff}$ fractions (>70 %) in some megacities and supercities compared with large and medium-sized cities. Noting that the $C_{ff}/C_{xs}$ ratio is critically sensitive to background selection. Regional backgrounds (as implemented here) introduce $C_{bio}$ contributions from surrounding rural/ agricultural sources to $C_{xs}$, whereas local urban backgrounds effectively isolate urban emissions by filtering out these external biogenic signals, thereby increasing the apparent $C_{ff}$ fraction compared to regional background approaches. The consistent adoption of regional background methodologies across all studies in Table E1 ensures the comparative validity of the results, as they share a common framework for accounting for $C_{bio}$ influences from peripheral non-urban sources. The derived annual $C_{ff}$ averages are (15.3 ± 5.2), (13.7 ± 10.2), (10.0 ± 5.2), and (8.2 ± 7.0) µmol mol$^{-1}$ in Guangzhou, Shenzhen, Zhanjiang, and Shaoguan, respectively, based on the mass balance equations of $CO_2$ and $\Delta(^{14}C)$. These $C_{ff}$ concentrations were low to moderate compared with those in other cities globally (Table E2, Fig. E1), despite the high emissions in Guangzhou and Shenzhen from inventories (Fig. 1).

### 3.3 Spatial distribution and seasonal variations

We identified potential source regions of $C_{ff}$ by analyzing its spatial distribution and seasonal variations. $C_{ff}$ is found to be emitted dominantly from densely populated urban downtown areas (GZ6 and GZ5; SG3 and SG2) in Guangzhou during summer (GZs) and Shaoguan during winter (SGw), forming an "urban $C_{ff}$ dome" (Fig. 1cj). This was further supported by significant positive correlations between the $C_{ff}$ measurements and the corresponding gridded ODIAC (Oda and Maksyutov, 2011; Oda and Maksyutov, 2024) inventory emissions in GZs ($r = 0.53$, $p = 0.1$) and SGw ($r = 0.91$, $p = 0.03$). The "urban $C_{ff}$ dome" indicates that $C_{ff}$ is mainly derived from the localized fossil fuel combustion, which is likely to be influenced by the urban topography. That is, downtown Guangzhou and downtown Shaoguan are surrounded by mountains to the east, north, and west. In contrast, we found that $C_{ff}$ was mainly emitted from western industrial areas (SZ2) in Shenzhen during summer (SZs), and from port areas (ZJ5>ZJ2; ZJ2>ZJ3>ZJ4) in Zhanjiang during winter (ZJw) and summer (ZJs, by atmospheric transport) (Fig. 1ehg).

Atmospheric transmission of $C_{ff}$ from potential source regions was observed at large spatial scales combined with air mass back trajectories by the Hybrid Single-Particle Lagrangian Integrated Trajectory (HYSPLIT) model (Stein et al., 2015) and emission footprints by the FLEXPART dispersion model (Pisso et al., 2019). Shaoguan exhibited higher $C_{ff}$ concentrations in summer ((10.7 ± 8.3) µmol mol$^{-1}$) than in winter ((5.8 ± 4.4) µmol mol$^{-1}$). This was likely attributable to atmospheric trapping of emissions from the Pearl River Delta (PRD) urban agglomeration, as illustrated by air mass back trajectories (HYSPLIT, Fig. F1a) and emission footprints (FLEXPART, Fig. 3g), rather than high local emissions due to lower summer emissions from inventories. In contrast, we found higher $C_{ff}$ concentrations in winter compared with those in summer in Guangzhou ((17.7 ± 3.5) > (12.9 ± 5.6) µmol mol$^{-1}$), Shenzhen ((18.0 ± 9.9) > (9.2 ± 8.5) µmol mol$^{-1}$), and Zhanjiang ((12.1 ± 5.1) > (7.6 ± 4.3) µmol mol$^{-1}$), consistent with the values in 14 other Chinese cities (Zhou et al., 2020). The higher winter concentrations found in Guangzhou, Shenzhen, and Zhanjiang in this study likely resulted from atmospheric trapping of



emissions in the shallow planetary boundary layer, and high local emissions, because ODIAC (MEIC) indicates that winter emissions were 8 %, 10 %, and 11 % (17 %, 22 %, and 14 %) higher, respectively, than those in summer (Oda and Maksyutov, 2024; Meic, 2023). The atmospheric trapping of emissions is higher than local emissions during winter in
Guangzhou (GZw) and Shenzhen (SZw), which is supported by higher $C_{ff}$ concentrations occurring in downwind areas (GZ2, GZ6, and GZ10; SZ3 and SZ4) compared with upwind areas (GZ1 and GZ3; SZ8 and SZ9). The air mass back trajectories (HYSPLIT, Fig. F1b) and emission footprints (FLEXPART, Fig. 3bd) showed that the major source region was traced to the Yangtze River Delta (YRD) urban agglomeration in East China, and a portion from North China via long-range transport (Fig. F2ef). The major source region from the YRD was also reported in a study of CFC-11 in Shenzhen (Chen et al., 2024).





**Figure 3: FLEXPART footprints simulating C_ff emissions in summer (s) and winter (w) for (a and b, GZ) Guangzhou, (c and d, SZ) Shenzhen, (e and f, ZJ) Zhanjiang, and (g and h, SG) Shaoguan at heights from 0–100 m a.s.l. over a period of 30 days. Blue**




**squares are shown as enlarged maps in the right figures. Blue points represent the locations of sampling sites. Black lines indicate**
**the boundaries of continents (left), Chinese provinces (left, bold), and the nine cities of the PRD (right, bold) taken from Natural**
**Earth (https://www.naturalearthdata.com/, last accessed: 9 March 2024).**

## 3.4 Historical variations

We observed a 35 % decline in $C_{ff}$ concentrations in Guangzhou from $(23.7 \pm 12.9)$ µmol mol$^{-1}$ in 2010–2011 (Ding et al., 2013) to $(15.3 \pm 5.2)$ µmol mol$^{-1}$ in 2022 in this study ($p < 0.01$, two-tailed t-test) (Fig. 4a, Table E2). To conservatively

validate this reduction, we employed a dual-method analytical framework for 2022 data designed to maximize $C_{ff}$ estimates: (1) adopting a regional (instead of urban) background to establish a lower baseline, and (2) applying both literature-derived and simulation-based disequilibrium corrections (β) to address potential biases. This validation confirmed 2022 $C_{ff}$ values remain statistical significantly lower than 2010–2011 levels across all the observational data ($p < 0.01$). Additional site-specific validation at GZ7 station showed a 41 % winter reduction from $(28.4 \pm 15.0)$ µmol mol$^{-1}$ in 2010 to $(16.8 \pm 3.4)$

µmol mol$^{-1}$ in 2022, with equivalent statistical significance ($p < 0.01$). By adjusting baseline assumptions and correction parameters, this methodology ensures that the reported decline is robust against potential overestimations in 2022. It provides reliable evidence for Guangzhou's decarbonization progress, even under maximized $C_{ff}$ calculation scenarios in 2022.

We found that reductions in $C_{ff}$ concentration also occurred in other Chinese cities, and some cities outside China.

Statistically significant reductions were observed in Chinese megacities and supercities (Fig. 4a, Table E2): Beijing showed a 32 % decrease from $(39.7 \pm 36.1)$ µmol mol$^{-1}$ in 2014 to $(27.0 \pm 0.3)$ µmol mol$^{-1}$ in 2014–2016 ($p < 0.05$, two-tailed t-test) and a 50 % decrease to $(19.7 \pm 22.0)$ µmol mol$^{-1}$ by winter 2020 ($p < 0.01$); Xi'an decreased by 36 % from $(40.1 \pm 3.8)$ µmol mol$^{-1}$ in 2011–2013 to $(25.7 \pm 1.1)$ µmol mol$^{-1}$ in 2014–2016 ($p < 0.001$), with suburban areas declining by 12 % from $(23.5 \pm 6.5)$ µmol mol$^{-1}$ in 2016 to $(13.1 \pm 10.9)$ µmol mol$^{-1}$ in 2021 ($p < 0.05$). These trends are consistent with tree-ring

$\Delta(^{14}C)$ records showing peaks in urban $C_{ff}$ emissions for Beijing (2010) and Xi'an (2013) (Niu et al., 2024). Globally, Krakow, Poland showed a 93 % decrease from 27.5 µmol mol$^{-1}$ in 1989 to 1.98–2.18 µmol mol$^{-1}$ in 2005–2009 ($p < 0.001$), Heidelberg, Germany a 2 % decrease from $(11.09 \pm 0.24)$ µmol mol$^{-1}$ in 1986–1996 to $(10.92 \pm 0.34)$ µmol mol$^{-1}$ in 1997–2007 ($p = 0.04$), and Los Angeles, USA a 42 % decrease from $(22.9 \pm 5.6)$ µmol mol$^{-1}$ in 2006–2013 to $(13.2 \pm 9.4)$ µmol mol$^{-1}$ in 2014–2016 ($p < 0.01$) (Table E2, Fig. E1). The consistent statistical significance ($p < 0.05$ threshold unless noted)

across atmospheric measurements and proxy records underscores the effectiveness of emission mitigation policies in these cities.

We found that $C_{ff}$ concentration reduction did not only occur in the above-mentioned cities, but also probably in their source regions. The winter $C_{ff}$ emission source regions of Guangzhou and Shenzhen are highly overlapped with China's high $C_{ff}$ emitting areas: North China, the YRD in East China, and the PRD in South China (Figs. 3 and F2). These areas are also

mostly consistent with the provinces that attained the higher $C_{ff}$ emission reductions due to stricter pollution control policies.



These provinces include Hebei, Shandong, Zhejiang, Shanxi, and Henan in North and East China according to the MEIC inventory (Shi et al., 2022). Therefore, it's very likely that $C_{ff}$ emissions are also decreasing or in a declining growth rate in China's high $C_{ff}$ emitting areas.

Similar reductions were found in $C_{ff}$ emissions from 2012 to 2020 according to the MEIC inventory (Shi et al., 2022), such

as Guangzhou (by 16 % from 2011), Shenzhen (by 3 %), Zhanjiang (by 0.1 %), Beijing (by 16 %), and Xi'an (by 9 %) (Fig. 4b), particularly in the industrial and power sectors (Li et al., 2017). We also found such declines in the MIXv2 Asian emission inventory (MIXv2, excluding Shenzhen and Shaoguan) (Li et al., 2024) and another carbon inventory for most Chinese cities (Zhang et al., 2024), but not in the ODIAC (Oda and Maksyutov, 2024) and the Emissions Database for Global Atmospheric Research (EDGAR) (Crippa et al., 2023). In fact, the mitigation of $C_{ff}$ emissions in China's MEIC

inventory was primarily driven by heterogeneous trends across cities: 38 % exhibited sustained emission reductions, 29 % showed an initial decline followed by a rebound, while 33 % maintained increasing trajectories. Notably, cities achieving sustained reductions were disproportionately concentrated in larger cities, comprising 86 % of megacities, 43 % of supercities, and 43 % of Type I large cities (populations of 3–5 million). In contrast, smaller cities showed lower mitigation prevalence, with only 34 % of Type II large cities (1–3 million) and 38 % of medium/ small cities attaining emission

decreases.

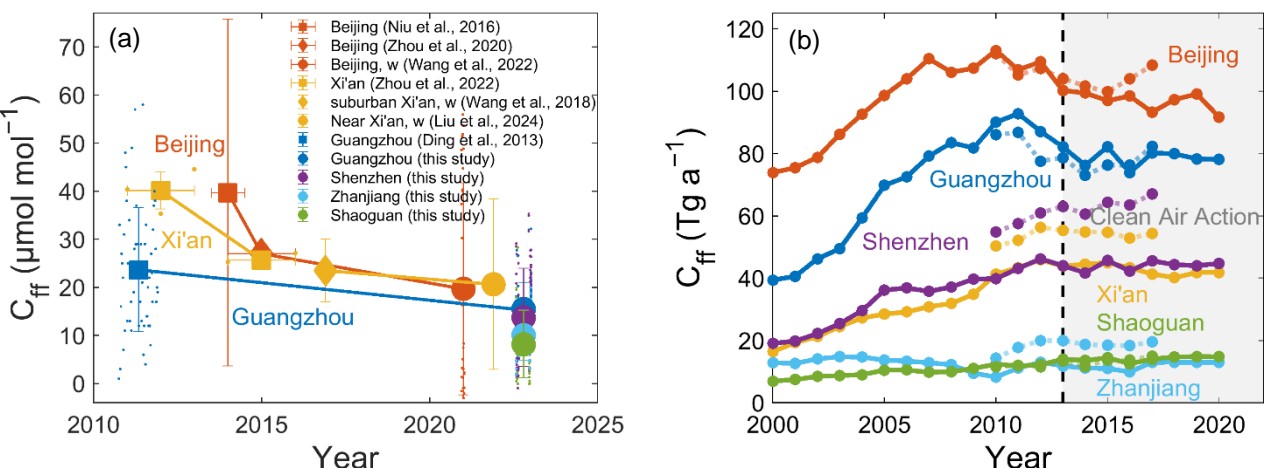

**Figure 4: (a) $C_{ff}$ concentrations from atmospheric measurements (Niu et al., 2016; Wang et al., 2022b; Zhou et al., 2022; Ding et al., 2013; Zhou et al., 2020; Wang et al., 2018) in Beijing, Xi'an, Guangzhou, Shenzhen, Zhanjiang, and Shaoguan. The large symbols**
**indicate annual means, multiyear averages, or winter means (w), and the small symbols represent the corresponding scattered measurements. $C_{ff}$ observations are calculated as enhancements over background Nanling for Guangzhou and Waliguan for Beijing and Xi'an. The $C_{ff}$ concentrations in winter in Beijing obtained from Wang et al.[40] were estimated based on the background $\Delta(^{14}CO_2)$ measurements from Waliguan (Liu et al., 2024). The y-axis error bars indicate uncertainty, and the x-axis error bars represent the observed period. (b) $C_{ff}$ emissions from the MEIC (solid lines) (Li et al., 2017; Meic, 2023; Zheng et al.,**
**2018) and MIXv2 (dotted lines) (Li et al., 2024) inventories in Beijing, Xi'an, Guangzhou, Shenzhen, Zhanjiang, and Shaoguan since 2010. The vertical dashed line indicates the year 2013 when China's Clean Air Action Plan was implemented.**



### 3.5 Driver factors

### 3.5.1 Coal-to-gas transition

We first determined the coal, oil, and natural gas fractions of $C_{ff}$ using the Keeling plot of $\delta(^{13}C)$ and $CO_2$ (i.e., scatter plot between $\delta(^{13}C)$ and inverse of $CO_2$ mole fractions) and the Bayesian mixing model (MixSIAR) (Stock et al., 2018) during winter 2022. The fractions in winter were $(49 \pm 25)$ %, $(29 \pm 22)$ %, and $(22 \pm 19)$ %, respectively, for Guangzhou, $(47 \pm 25)$ %, $(29 \pm 21)$ %, and $(24 \pm 20)$ % for Shenzhen, $(43 \pm 24)$ %, $(29 \pm 21)$ %, and $(28 \pm 21)$ % for Zhanjiang, and $(39 \pm 24)$ %, $(34 \pm 23)$ %, and $(27 \pm 21)$ % for Shaoguan. Coal combustion was the largest contributor to $C_{ff}$ emissions, followed in

descending order by oil combustion and natural gas combustion. Compared with other cities around the world (Table G1), we found natural gas was the primary fuel type consumed in Paris (70 %) (Lopez et al., 2013) and Beijing $[(55 \pm 9)$ %] (Wang et al., 2022b), whereas oil was the main fuel type consumed in Los Angeles (>50 %) (Djuricin et al., 2010; Newman et al., 2016). Coal remains the primary fossil fuel used in Xi'an $[(72.6 \pm 10.4)$ % in 2014 and $(54 \pm 4)$ % in 2019) (Wang et al., 2022b; Zhou et al., 2014), Guangzhou (49 % in 2022), and Shenzhen (47 % in 2022). Notably, cities with high $C_{ff}$

emissions consume all three types of fossil fuels, with the dominant fuel type varying by city. Coal remains the primary fossil fuel used in many Chinese cities.

The reduction in $C_{ff}$ concentrations can be attributed to changes in energy systems as a result of China's clean air measures (Shi et al., 2022). A major contribution has been the reduction in coal usage and the shift to low-carbon energy sources such as natural gas. During 2013–2022, the share of coal in the energy mix decreased by 4.9 % in China and by 7.1 % in

Guangdong Province, whereas the share of natural gas increased by 3.0 % in China and by 7.2 % in Guangdong Province, according to the MEIC inventory (Li et al., 2017; Zheng et al., 2018; Meic, 2023; Xu et al., 2024). By applying the coal, oil, and natural gas fractions of $C_{ff}$ derived from our measurements, it's likely that coal usage in Guangdong Province since 2013 have decreased ≥21 %, and natural gas usage have increased by ≥16 % (Fig. 5a). Similarly, in Guangzhou city, it's likely that coal usage since 2011 has decreased by 23 % instead of by 8.8 % (Fig. 5b), and natural gas usage has increased by 17 %

instead of by 7.9 %, assuming that the fuel type fractions of $C_{ff}$ in Guangzhou city were the same as those in Guangdong Province in the inventory.



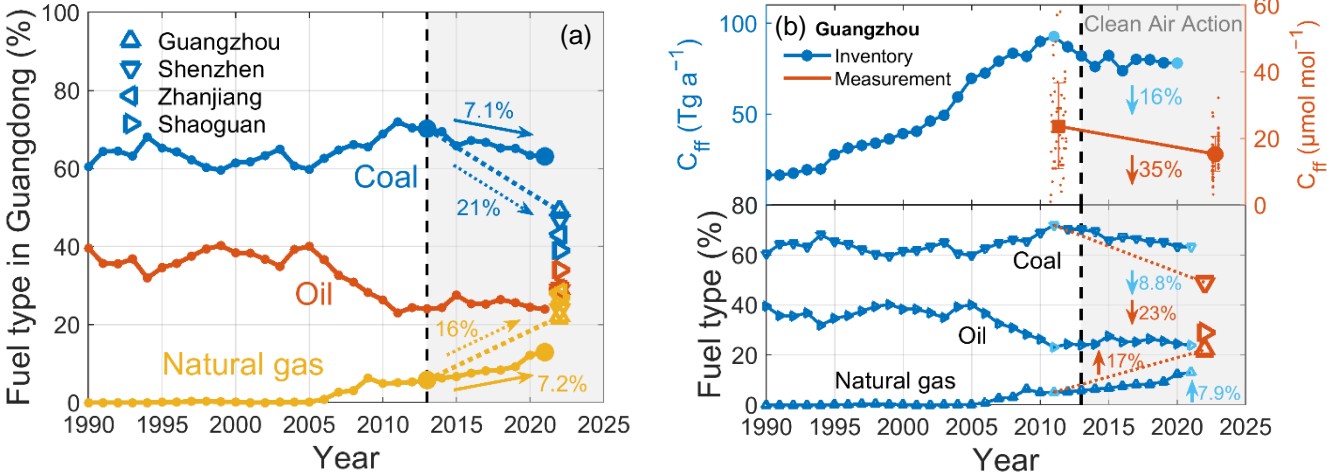

**Figure 5: (a) Coal, oil, and natural gas fractions of $C_{ff}$ in Guangdong Province from the MEIC inventory from 1990 to 2021 (points), and in the cities of Guangzhou, Shenzhen, Zhanjiang, and Shaoguan from measurements in this study in 2022 (triangles). (b) (Top) Comparison of reductions in $C_{ff}$ inventory emissions (blue) and measured $C_{ff}$ concentrations (red) resulting from (Bottom) reduced coal usage and increased natural gas usage in Guangzhou. The vertical dashed line indicates the year 2013 when China's Clean Air Action Plan was implemented.**

### 3.5.2 Combustion efficiency improvement

We first calculated $R_{CO/CO2ff}$ ratios at each measurement site and found higher ratios in summer than winter (Fig. H1). However, we focused only on observations in winter for four reasons. First, summer CO shows greater instability as its atmospheric lifetime depends on OH radical production, which is enhanced through photochemical reactions (e.g., $CH_4$ oxidation) under intense solar radiation, making CO a less reliable fossil fuel tracer (Rosendahl, 2022). Second, winter exhibits stronger $\Delta CO$-$C_{ff}$ correlations ($r > 0.6$, $p < 0.01$; Fig. H2) with better regional representativeness due to extended CO atmospheric lifetime from slower CO oxidation rates. Third, the winter $\Delta CO$-$C_{ff}$ relationship better captures anthropogenic emission characteristics compared to other seasons. Fourth, weaker vertical mixing in winter accentuates local emission impacts (Wang et al., 2010).

We then estimated winter 2022 $R_{CO/CO2ff}$ ratios across Chinese cities using $\Delta CO$-$C_{ff}$ regression slopes (Fig. H2), with spatial variations primarily attributed to differences in fuel composition and combustion efficiency (Graven et al., 2009). CO is generated through incomplete combustion of both fossil fuels and biomass. These spatial patterns are consistent with combustion characteristics showing biomass burning produces higher CO emissions per unit energy than fossil fuel combustion (Akagi et al., 2011). As shown in Fig. H1, suburban/rural sites (GZ1, SZ9, ZJ1, SG1) exhibited significantly higher ratios than urban sites (GZ5, SZ7, ZJ4, SG3): GZ1 > GZ5 ((30.4 ± 10.0) > (19.8 ± 4.6) nmol µmol$^{-1}$), SZ9 > SZ7 ((41.3 ± 23.0) > (14.8 ± 2.4) nmol µmol$^{-1}$), ZJ1 > ZJ4 ((41.2 ± 3.6) > (11.9 ± 6.4) nmol µmol$^{-1}$), and SG1 > SG3 ((26.7 ± 2.9) > (15.1 ± 3.6) nmol µmol$^{-1}$). This pattern is in agreement with previous studies attributing elevated ratios in non-urban





areas to biomass burning contributions (Rosendahl, 2022). In contrast, megacities showed 35–40 % lower ratios (Guangzhou: 13.3 nmol µmol$^{-1}$, Shenzhen: 14.1 nmol µmol$^{-1}$) compared to smaller cities (Zhanjiang: 21.3 nmol µmol$^{-1}$, Shaoguan: 21.7 nmol µmol$^{-1}$; $p < 0.01$; Fig. H2), suggest higher fossil fuel combustion efficiency and/or lower biomass burning inputs.

Guangzhou's ratios are dominated by improved fossil fuel combustion efficiency due to having the highest biomass burning emissions among the four studied cities in the EDGAR2024 inventory, while Shenzhen's ratios are attributed to both factors with nearly negligible biomass contributions corresponding to its 2017 biomass boiler phase-out policy.

We retrieved historical $R_{CO/CO2ff}$ data from observations in China by estimation from $\varDelta(^{14}C)$ measurements and correction from $R_{CO/CO2}$ (increased by 20 %) (Table H1), and $I_{CO/CO2ff}$ data from the MEIC, MIXv2, and EDGAR inventories (Fig. 6).

Given the minor contribution of biomass burning (BB)-related CO emissions across all inventories, with $I_{CO/CO2ff}$ ratio below 0.003 (MEIC, incorporating OBBEIC data (Song et al., 2009; Huang et al., 2012)), less than 1.0 (MIXv2), and declining from 3.8 (1990) to 1.1 (2022) in EDGAR, we assume that interannual variability in BB emissions has negligible influence on the overall emission ratios. Observational and inventory data show a sustained decline in China's $R_{CO/CO2ff}$ and $I_{CO/CO2ff}$ ratios over the past 30 years (Fig. 6a), demonstrating that efforts to improve fossil fuel combustion efficiency are effective (Wang

et al., 2010; Lee et al., 2020), which is another factor contributing to the reduction in $C_{ff}$ concentrations. The MEIC inventory attributes this trend to spatiotemporally heterogeneous mitigation pathways: 72 % of the cities started $I_{CO/CO2ff}$ reductions during 1990–1994, while the remaining 28 % (mainly concentrated in the western provinces) exhibited a delayed start until 1995–2004. The implementation of China's clean air policies since 2013 has systematically phased out small, inefficient combustion facilities and replaced them with centralized, high efficient, and clean energy infrastructure (Shi et al.,

2022). The phase-out of coal-fired industrial boilers during 2013–2020 reduced $CO_2$ emissions by $(1.5 \pm 0.3)$ Gt, accounting for 12 % of the national industrial emission reduction (Li, 2023). These technological transitions enhanced combustion efficiency by >10 %, and reduced coal-dominated energy intensity by 40 % across the sector. The MEIC inventory showed that these synergistic measures resulted in significant energy savings, with a net reduction of 0.25 gigatonnes of coal equivalent (Gtce) in 2020 and a cumulative reduction of 1.06 Gtce over the policy implementation period (Shi et al., 2022).

Critically, the efficiency-driven transition decoupled energy demand from $C_{ff}$ emissions, with combustion optimization directly reducing coal consumption 1–2 % and $C_{ff}$ emissions by 1–3 Gt per year after 2015 (Le Quéré et al., 2016; Friedlingstein et al., 2023a).

We systematically compared observational $R_{CO/CO2ff}$ values with inventory $I_{CO/CO2ff}$ estimates. Our 2022 measurements of the $R_{CO/CO2ff}$ ratios in megacities (Guangzhou and Shenzhen) were consistent with EDGAR estimates (14.9 nmol µmol$^{-1}$, 2022),

while those in smaller cities (Zhanjiang and Shaoguan) were closer to MEIC values (19.2 nmol µmol$^{-1}$, 2020) (Fig. 6a) and independent field measurements near Xi'an ($23 \pm 6$ nmol µmol$^{-1}$, 2021) (Liu et al., 2024). City comparisons of observations against MEIC estimates revealed systematic deviations: Shenzhen's observed ratio fell 24 % below inventory estimates (23.4 nmol µmol$^{-1}$), whereas Shaoguan's exceeded projections (12.7 nmol µmol$^{-1}$) by 38 %; Guangzhou's and Zhanjiang's are similar to inventory estimates (14.2 and 23.8 nmol µmol$^{-1}$, respectively) (Fig. 6b).



The three-decade observational $R_{CO/CO2ff}$ ratios are closer to (higher than) the MEIC estimates with a difference of $(22 \pm 23)$ %
compared with the MIXv2 and EDGAR estimates, when focusing on the ratios over time and ignoring the local deviations
caused by the specific cities. These findings indicate that the MEIC inventory is more accurate than the EDGAR inventory
for China. For specific cities, we found that the MEIC inventory estimates were deviated less from the observed $R_{CO/CO2ff}$
(based on $\Delta(^{14}C)$ measurements) in recent years than the corrected $R_{CO/CO2ff}$ (using $R_{CO/CO2}$) in earlier years for Beijing and

Guangzhou (Fig. 6b). For example, in Beijing, the discrepancy in the ratios between observations and inventories decreased
from 22 % in 2006–2007 ($R_{CO/CO2}$-corrected) (Wang et al., 2010) to 8.7 % in 2009–2010 ($\Delta(^{14}C)$-derived) (Turnbull et al.,
2011), and further declined to 7.0 % by 2014 ($\Delta(^{14}C)$-derived) (Niu et al., 2018). Similarly, in Guangzhou, the discrepancy
dropped from 84 % in 2009–2010 ($R_{CO/CO2}$-corrected) (Silva et al., 2013) to 34 % in 2014–2017 ($R_{CO/CO2}$-corrected) (Mai et
al., 2021), and eventually reached 6.4 % by 2022 ($\Delta(^{14}C)$-derived). These results suggest that $R_{CO/CO2}$ corrections should be

carefully interpreted, as the effect of $CO_2$ from non-fossil sources can significantly bias the results, even in megacities with
high $C_{ff}$ emissions. For example, human respiration could bias $R_{CO/CO2}$ low by about 9 % at a rural site near Beijing (Wang et
al., 2010; Turnbull et al., 2011).

Despite the relatively good agreement of ratios between observations ($R_{CO/CO2ff}$) and MEIC inventory ($I_{CO/CO2ff}$) at the
national scale, observational data exhibited significantly greater $R_{CO/CO2ff}$ reduction rates than inventory estimates when

examined at the city level. From observations (Fig. 6b), in Guangzhou, $R_{CO/CO2ff}$ decreased by 36 % from 35.8 nmol µmol$^{-1}$
in 2009–2010 (Silva et al., 2013) to 23.8 nmol µmol$^{-1}$ in winter of 2014–2017 (Mai et al., 2021) and by 63 % to 13.3 nmol
µmol$^{-1}$ in 2022; in Beijing, $R_{CO/CO2ff}$ decreased by 58 % from72.3 nmol µmol$^{-1}$ in 2004 (Han et al., 2009) to 30.4 nmol
µmol$^{-1}$ in 2014 (Niu et al., 2018); in Xi'an, $R_{CO/CO2ff}$ decreased by 50 % from $(46 \pm 13)$ nmol µmol$^{-1}$ in 2016 (Wang et al.,
2018) to $(23 \pm 6)$ nmol µmol$^{-1}$ in 2021 (Liu et al., 2024). The MEIC estimates for the above three cities decreased by 36 %,

52 %, and 21 %, respectively, over the same period. Larger reductions of the ratios were found from observations than those
from the MEIC inventory (i.e., 63 % > 36 % for Guangzhou, 58 % > 52 % for Beijing, and 50 % > 21 % for Xi'an). This
conclusion holds even after artificially biasing the $R_{CO/CO2ff}$ ratio downward by about 9 % to account for human respiration in
Beijing (2004) and in Guangzhou (2009–2010 and 2014–2017). These findings suggest that the MEIC inventory may
insufficiently capture, or lag, the rapid improvement in combustion efficiency and energy structure transformation in China.

The three-decade decline in China's $R_{CO/CO2ff}$ ratios demonstrates both improved fossil fuel combustion efficiency and
successful implementation of air pollution control policies i.e., the success of air pollution emission reduction efforts. Our
observations reveal significantly greater urban $R_{CO/CO2ff}$ reductions than those estimated by the MEIC inventory, indicating
potential underestimation of CO emission reductions relative to $C_{ff}$ mitigations in current inventories. This finding aligns
with previous reports of inventory underestimates for real-world CO reductions. Mai et al. (2021) showed that the MEIC

inventory may underestimate cumulative reductions from fleet turnover and catalytic converter upgrades, despite China's
National V standards having achieved the $\leq 1$ g km$^{-1}$ CO emission limit since 2013. Together, these results imply that the
MEIC inventory might systematically underestimate the actual effectiveness of clean air policies in reducing air pollutant
emissions.



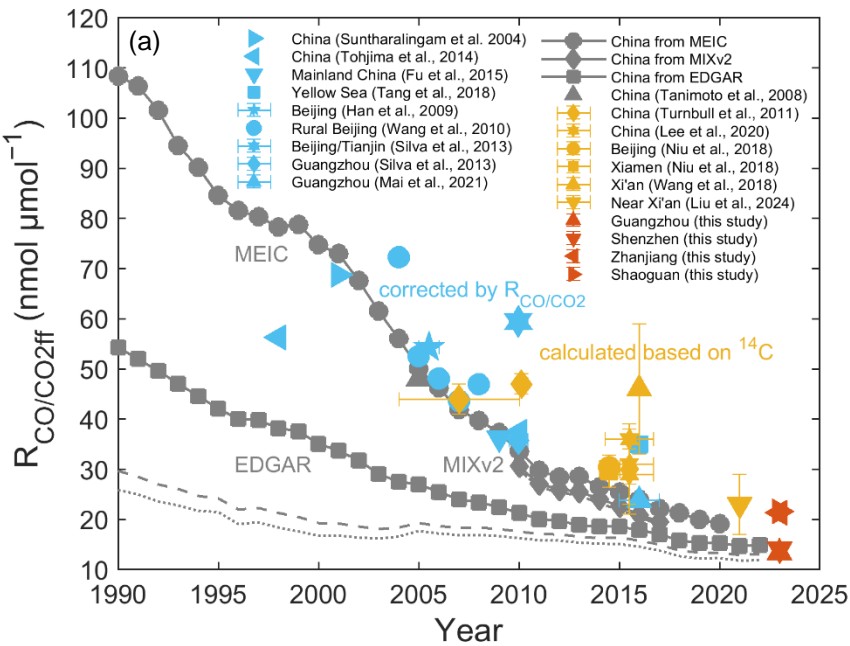

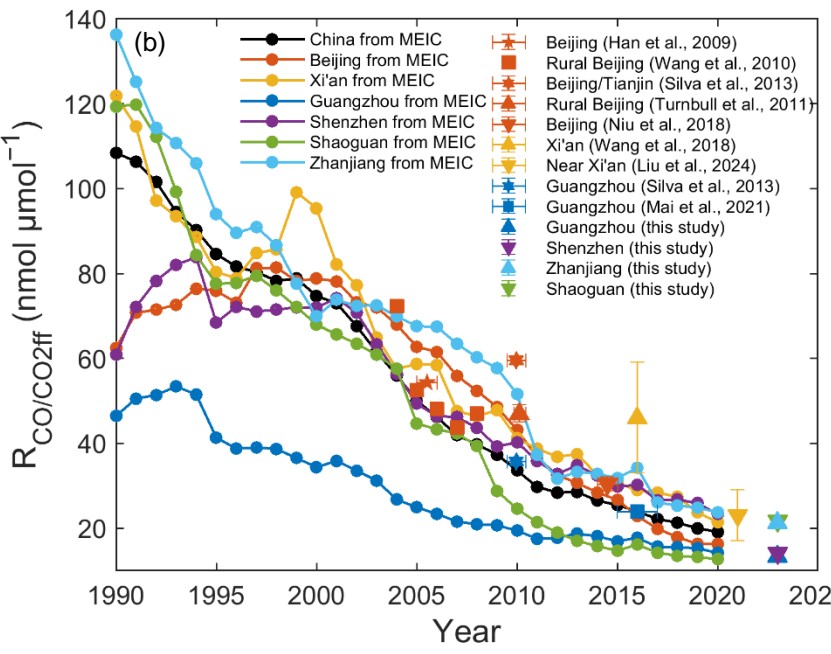


**Figure 6:** $R_{CO/CO2ff}$ for (a) China and for (b) Chinses cities obtained from inventories and observations (values refer to Table H1). For (a), the gray symbols represent data from the emission inventories (Tanimoto et al., 2008), including MEIC (Meic, 2023; Xu et al., 2024; Li et al., 2019; Li et al., 2017), MIXv2 (Li et al., 2024), and EDGAR2024 (Edgar, 2024). The $I_{CO/CO2ff}$ emission ratios





derived from the three inventories are shown with distinct approaches: (1) MEIC calculated the $I_{CO/CO2ff}$ ratio for all anthropogenic sectors (represented by solid line with point symbols); (2) MIXv2 computed two variants: combining anthropogenic sectors with open biomass burning (solid line with diamond symbols) and anthropogenic-only emissions (dash-dotted line); while (3) EDGAR2024 provided three ratios: fossil + biogenic CO (solid line with square symbols), fossil + biomass burning CO (dashed line), and fossil-only CO (dotted line), all relative to $C_{ff}$ emissions. The light blue symbols represent $R_{CO/CO2ff}$ corrected by $R_{CO/CO2}$ from observational studies (Wang et al., 2010; Tohjima et al., 2014; Suntharalingam et al., 2004; Tang et al., 2018; Han et al., 2009; Fu et al., 2015), assuming that 20 % of the $CO_2$ enhancement was from sources other than $C_{ff}$. The orange symbols represent $R_{CO/CO2ff}$ calculated based on atmospheric $^{14}CO_2$ measurements from previous studies (Turnbull et al., 2011; Niu et al., 2018; Lee et al., 2020; Wang et al., 2018; Liu et al., 2024). The red symbols depict the values observed in this study. For (b), the Chinese cities include Beijing, Xi'an, Guangzhou, Shenzhen, Zhanjiang, and Shaoguan from the MEIC inventory (filled circles) and observations from previous studies (Wang et al., 2018; Liu et al., 2024; Wang et al., 2010; Silva et al., 2013; Niu et al., 2018; Mai et al., 2021; Han et al., 2009; Turnbull et al., 2011) and this study since 1990. The up and down triangles represent $R_{CO/CO2ff}$ estimated based on atmospheric $\Delta(^{14}CO_2)$ measurements. Other symbols represent the $R_{CO/CO2ff}$ corrected by $R_{CO/CO2}$ from observational studies, assuming that 20 % of the $CO_2$ enhancement is from sources other than $C_{ff}$. The y-axis error bars indicate uncertainty, and the x-axis error bars represent the observed period.

## 3.6 Implication

Since 2013, China has implemented a series of measures with the explicit aim of improving air quality. While the initial goal of China's clean air targets was to address air pollution, they also served as a powerful catalyst for the simultaneous transformation of energy systems and the mitigation of $C_{ff}$ emissions. As a result, we have observed $C_{ff}$ concentration and emission reductions in some Chinese megacities and supercities, such as Guangzhou, Beijing, and Xi'an. The achievement of peak emissions in Beijing (2010) and Xi'an (2013) marks a pivotal transition for China, signaling that cities across the nation, from megacities to small cities, are gradually reaching their emission peaks. This milestone has profound implications for both China's sustainable development and global climate governance, as China has dominated the global trend since 2010 (Friedlingstein et al., 2023b).

Despite China's remarkable success in reducing $C_{ff}$ emissions, continued efforts are needed to optimize the nation's energy system and economic structure in order to facilitate future green growth. It is imperative that common solutions to climate change and air pollution are formulated and implemented with urgency, as China has set a goal for all cities to meet current air quality standards by 2035 and has pledged to achieve carbon peak by 2030 and carbon neutrality by 2060. One available solution is to control the common key sources and dominant source regions of air pollution and $CO_2$ emissions (Wu et al., 2022; Zheng et al., 2024). In future policymaking, it is essential to adopt a co-beneficiary strategy that co-ordinates clean air measures and addresses climate change measures. This strategy, together with the associated assessment approach, will be an essential part of achieving sustainable development.

## 4 Conclusions

This study advances the understanding of urban $C_{ff}$ concentration changes in China through three key contributions. First, we pioneer the integration of biomass burning emissions into $C_{ff}$ estimation frameworks, significantly improving



methodological accuracy. Second, we identify substantial $C_{ff}$ reductions in cities and their source regions, driven by coal-to-
gas transitions (evidenced by stable isotope analysis) and combustion efficiency improvements (confirmed by declining
$R_{CO/CO2ff}$ ratios), where megacities and supercities lead this decline. Finally, through systematic analysis of long-term
$R_{CO/CO2ff}$ trends, we reveal current emission inventories may underestimate combustion efficiency gains and CO emission
reductions relative to $C_{ff}$ mitigations. These findings provide critical support for refining emission accounting systems and

developing evidence-based climate policies. The integrated approach offers new insights into urban emission dynamics and
mitigation effectiveness.

This study has some limitations in sampling and source attribution. First, current sampling only covers summer and winter;
future work should include all seasons to better capture annual trends. Second, the $\delta(^{13}C)$ signatures used here mainly rely on
published data, introducing potential uncertainties. Direct measurements of source-specific isotopic values would help refine

the analysis. Additionally, future studies could incorporate atmospheric modelling and inversion methods to improve
emission estimates. This would require high resolution prior flux data and validation against direct measurements (e.g.,
radiocarbon analysis). Addressing these gaps would enhance source apportionment accuracy and enable a more robust
integration of top-down (e.g., inversions) and bottom-up (e.g., inventories) approaches for evaluating urban emission
mitigation strategies.




## Appendix A: Seasonal averages and quality control of $\Delta(^{14}C)$ and $\delta(^{13}C)$ measurements

**Table A1** $\Delta(^{14}C)$ and $\delta(^{13}C)$ averages and standard deviations at 30 sampling sites

| City | Site code | Summer | | Winter | | Altitude (m a.s.l.) | Elevation (m a.g.l.) | Site description |
|------|-----------|--------|--------|--------|--------|----------|-----------|------------------|
| | | $\Delta(^{14}C)$ (‰) | $\delta(^{13}C)$ (‰) | $\Delta(^{14}C)$ (‰) | $\delta(^{13}C)$ (‰) | | | |
| Guangzhou | GZ1 | −15.3 ± 9.8 | −9.0 ± 0.7 | −37.6 ± 8.2 | −8.9 ± 0.2 | 212 | 25 | Suburban rooftops |
| | GZ2 | −21.7 ± 4.8 | −9.0 ± 0.3 | −48.4 ± 5.6 | −8.9 ± 0.3 | 19 | 20 | Suburban rooftops |
| | GZ3 | −16.4 ± 8.9 | −8.7 ± 0.3 | −38.0 ± 4.0 | −8.7 ± 0.2 | 120 | 30 | Suburban rooftops |
| | GZ4 | −16.1 ± 4.8 | −9.0 ± 0.7 | −41.3 ± 9.4 | −9.0 ± 0.3 | 23 | 35 | Urban rooftops |
| | GZ5 | −34.3 ± 12.4 | −9.6 ± 0.8 | −39.6 ± 8.3 | −8.5 ± 0.3 | 46 | 35 | Urban rooftops |
| | GZ6 | −33.7 ± 16.8 | −9.5 ± 1.0 | −44.4 ± 11.3 | −8.9 ± 0.1 | 53 | 60 | Urban rooftops |
| | GZ7 | −17.0 ± 5.1 | −8.8 ± 0.4 | −39.6 ± 7.3 | −8.9 ± 0.3 | 120/ 75 | 118/ 40 | Urban tower/ Urban rooftops |
| | GZ8 | −24.4 ± 9.5 | −9.4 ± 0.5 | −41.2 ± 8.1 | −9.2 ± 0.2 | 12 | 30 | Urban rooftops |
| | GZ9 | −27.4 ± 5.4 | −9.8 ± 0.4 | −42.7 ± 7.0 | −8.7 ± 0.3 | 50 | 30 | Urban rooftops |
| | GZ10 | −26.6 ± 20.0 | −9.5 ± 1.0 | −42.8 ± 5.9 | −9.0 ± 0.2 | 54 | 40 | Suburban rooftops |
| Shenzhen | SZ1 | −14.9 ± 18.1 | −8.6 ± 0.5 | −38.0 ± 27.3 | −9.2 ± 0.2 | 40 | 30 | Suburban tower |
| | SZ2 | −55.2 ± 3.9 | −9.2 ± 0.4 | −43.5 ± 9.0 | −9.4 ± 0.1 | 28 | 15 | Rooftops in Industrial area |
| | SZ3 | −12.3 ± 22.7 | −8.7 ± 0.5 | −48.0 ± 16.2 | −9.4 ± 0.3 | 14 | 15 | Urban rooftops |
| | SZ4 | −15.2 ± 28.1 | −8.8 ± 0.6 | −45.3 ± 15.3 | −9.5 ± 0.3 | 42 | 40 | Urban campus rooftops |
| | SZ5 | −17.9 ± 17.7 | −8.8 ± 0.3 | −42.1 ± 12.5 | −9.4 ± 0.3 | 40 | 30 | Urban campus rooftops |
| | SZ6 | −17.3 ± 10.4 | −8.7 ± 0.5 | −43.1 ± 32.2 | −9.1 ± 0.4 | 60 | 30 | Suburban tower |
| | SZ7 | −11.3 ± 6.1 | −8.8 ± 0.1 | −40.3 ± 33.5 | −9.3 ± 0.6 | 210 | 200 | Urban rooftops |
| | SZ8 | −9.8 ± 12.8 | −8.6 ± 0.5 | −35.5 ± 32.0 | −9.0 ± 0.3 | 150 | 110 | Suburban rooftops at the boundary site |
| | SZ9 | −4.5 ± 11.6 | −8.5 ± 0.4 | −36.8 ± 32.3 | −9.2 ± 0.3 | 60 | 30 | Suburban tower |
| | SZ10 | 1.9 ± 1.9 | −9.0 ± 0.8 | −44.4 ± 7.8 | −9.4 ± 0.2 | 60 | 20 | Suburban rooftops |
| Zhanjiang | ZJ1 | −10.5 ± 9.5 | −10.2 ± 0.5 | −27.3 ± 13.0 | −9.1 ± 0.4 | 8 | 20 | Rural rooftops |
| | ZJ2 | −15.0 ± 14.7 | −9.5 ± 0.6 | −27.1 ± 12.7 | −9.0 ± 0.2 | 24 | 40 | Urban rooftops |
| | ZJ3 | −9.8 ± 4.9 | −10.2 ± 0.6 | −28.6 ± 12.8 | −9.1 ± 0.1 | 44 | 40 | Urban rooftops |
| | ZJ4 | −5.6 ± 7.1 | −8.9 ± 0.3 | −30.3 ± 12.4 | −9.1 ± 0.2 | 25 | 40 | Urban rooftops |
| | ZJ5 | −14.6 ± 13.7 | −8.8 ± 0.2 | −31.9 ± 11.8 | −9.4 ± 0.4 | 41/ 46 | 50/ 30 | Suburban campus site/ Site near the port area |
| Shaoguan | SG1 | −15.9 ± 16.1 | −8.9 ± 0.3 | −11.7 ± 5.2 | −9.0 ± 0.1 | 114 | 30 | Suburban rooftops |
| | SG2 | −20.7 ± 15.2 | −9.5 ± 0.3 | −19.7 ± 3.8 | −8.9 ± 0.2 | 60 | 40 | Urban campus rooftops |
| | SG3 | −18.0 ± 17.5 | −9.2 ± 0.5 | −26.6 ± 10.4 | −9.0 ± 0.2 | 68 | 40 | Urban rooftops |
| | SG4 | −35.2 ± 21.0 | −9.0 ± 0.2 | −11.1 ± 6.3 | −8.8 ± 0.2 | 95 | 30 | Rural site |
| | SG5/ NL | 5.1 ± 1.3 | −9.2 ± 0.2 | −2.0 ± 0.8 | −9.3 ± 0.2 | 1700 | 15 | Rooftops at the background site |




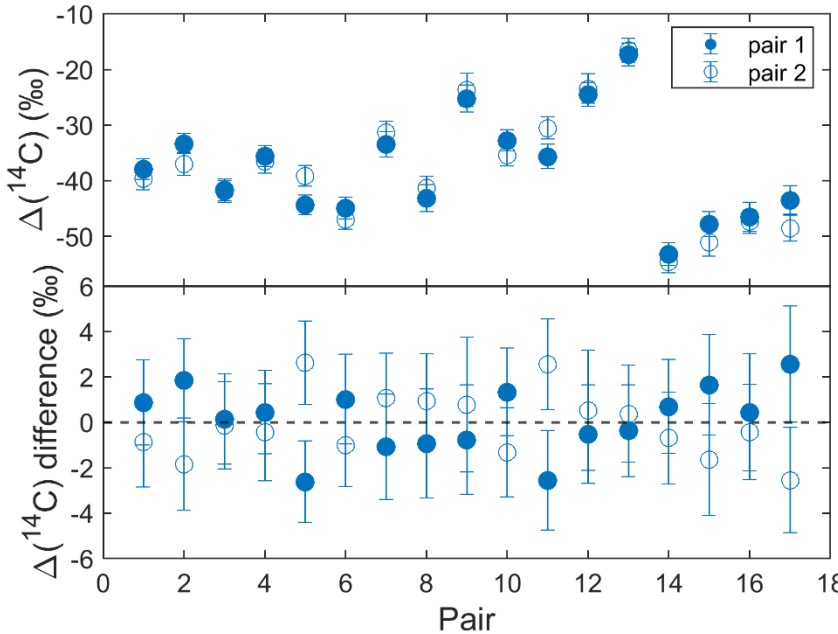

**Figure A1** Pair differences of $\Delta(^{14}C)$ for replicate measurements. Replicates were obtained from parallel air samples. The difference of each individual measurement from its pair mean is shown. Closed and open symbols are the first and second group taken from each pair, respectively. Error bars are the 1-sigma uncertainty on each measurement.


**Appendix B: Radiocarbon isotope endmembers for biomass burning**

Atmospheric $^{14}CO_2$ is assimilated by plants via photosynthesis, imprinting atmospheric $\Delta(^{14}CO_2)$ signatures into plant tissues. This creates a bidirectional link: plant $\Delta(^{14}C)$ reflects atmospheric $\Delta(^{14}CO_2)$ levels, while atmospheric $\Delta(^{14}CO_2)$ dynamics can be inferred from plant biomass archives (e.g., tree rings). Annual biomass $\Delta(^{14}C)$ closely matches contemporaneous atmospheric $\Delta(^{14}CO_2)$ (due to rapid carbon turnover within a single growing season). Multi-year biomass $\Delta(^{14}C)$ represents an integrated signal, blending atmospheric $\Delta(^{14}CO_2)$ variations over its growth period (e.g., tree rings capture annual $\Delta(^{14}CO_2)$ fluctuations).

**B1 Annual biomass.** The $\Delta(^{14}C)$ for annual biomass in 2022 was estimated as $-14.8 \pm 2.2$ ‰ (mean ± MSE; $\Delta_a$), derived from a linear regression model of atmospheric $\Delta(^{14}CO_2)$ decline ($-4.4$ ‰ $a^{-1}$) observed in Northern Hemisphere zone 3 between 2010 and 2018 (Hua et al., 2021).

**B2 Multi-year biomass.** The $\Delta(^{14}C)$ for multi-year biomass is related with its age; the year it was growing, the annual increase in biomass, and atmospheric $^{14}CO_2$ during its growth cycle. The $\Delta(^{14}C)$ for multi-year biomass can be determined (Lewis et al., 2004):





$$\Delta^{14}C = \frac{\int_{t_1}^{t_2} \Delta^{14}C(t)w(t)dt}{\int_{t_1}^{t_2} w(t)dt} \qquad (B1)$$

where $\Delta^{14}C(t)$ is the atmospheric $\Delta(^{14}CO_2)$ at age t, and the weighting function w(t) is the growth rate of carbon in biomass

at age t, which can be determined by the Chapman-Richards growth model (Lewis et al., 2004):

$$V = A(1 - e^{-\frac{t-t_0}{\tau}})^m \qquad (B2)$$

$$w(t) = \frac{dV}{dt} \qquad (B3)$$

where V is the volume of a tree at age *t* (V = 0 at t = $t_0$), and the parameters A, $\tau$, and m can be chosen empirically to fit

measured tree growth characteristics. The Chapman-Richards growth model describes cumulative growth of V.

It is assumed that the multi-year biomass was partitioned into five age cohorts (10-, 20-, 40-, 65-, and 85-year-old trees) with

relative share of 20 ± 10 %, 20 ± 10 %, 40 ± 20 %, 10 ± 5 % and 10 ± 5 %, respectively (Mohn et al., 2008). The

corresponding $\Delta(^{14}C)$ values were calculated as 20.9 ± 5.4 ‰, 52.9 ± 4.0 ‰, 137.5 ± 35.1 ‰, 261.2 ± 50.4 ‰, and 203.1 ±

17.4 ‰, respectively. Consequently, the $\Delta(^{14}C)$ signature of the multi-year biomass for the year 2022 was estimated as 116.2

± 17.6 ‰ (mean ± 1σ; $\Delta_m$) using the Chapman-Richards growth model ($\tau = 50, m = 3$) and long-term tree-ring $\Delta(^{14}C)$

measurements (Hua et al., 2021).

**B3 Biomass burning.** The $\Delta(^{14}C)$ endmember for biomass burning ($\Delta_{BB}$) was calculated using the two biomass types:

$$\Delta_{BB} = f_a \cdot \Delta_a + (1 - f_a) \cdot \Delta_m \qquad (B4)$$

where $\Delta_a$ and $\Delta_m$ represent the $\Delta(^{14}C)$ signatures of annual biomass (e.g., crop residues) and multi-year biomass (e.g., woody

waste), respectively, and $f_a$ is the annual biomass fraction.

Using this framework, we estimated the 2022 $\Delta(^{14}C)$ endmembers for biomass burning as 116.2 ± 17.6 ‰, 103.1 ± 15.8 ‰,

90.0 ± 14.1 ‰, 76.9 ± 12.3 ‰, 63.8 ± 10.6 ‰, and 50.7 ± 8.9 ‰ for $f_a$ values of 0 %, 10 %, 20 %, 30 %, 40 %, 50 %,

respectively.

**Appendix C: Bias correction for $C_{ff}$ calculation**

**C1 Nuclear facilities.** All operational (Daya Bay, Ling'ao, Yangjiang, Taishan) and under-construction (Lufeng,

Taipingling, Lianjiang) nuclear power plants (NPPs) along the coastline of Guangdong Province employ pressurized water

reactor (PWR) technology. Airborne $^{14}C$ releases from these facilities are predominantly hydrocarbons (75–95 %), mainly

$CH_4$, with only a small fraction in the form of $CO_2$ (Iaea, 2004). In fact, almost all reactors in China (> 95 %) are the PWR

type, which has the lowest emission factor for $^{14}CO_2$ release (Graven and Gruber, 2011; Yang, 2024). Graven and Gruber

(2011) note that most of China and the western US are the areas with very little potential bias in the derived $C_{ff}$, owing to

intense fossil fuel emissions but little to no nuclear activity. Graven et al. (2018) confirm this view that emissions from

reactors in the western US have a negligible effect on their samples, simulating an effect on inferred $C_{ff}$ of less than 0.1 ppm.

Although we can't obtain the $\Delta(^{14}CO_2)$ emission monitoring data from NPPs in China, *National Report on Radiation*



*Environment Quality* confirms that no incidents of excessive radiation dose discharges have occurred at NPPs in Guangdong

Province. For example, data from 2011 to 2014 show that liquid radionuclide discharges from Daya Bay and Ling'ao NPPs were only 0.09–0.38 % of legal thresholds. The 2019 report highlighted that public radiation doses at NPPs such as Daya Bay and Taishan remained "significantly below regulatory limits" (Meec, 2021). We thus assume that the $^{14}CO_2$ effect from the NPPs in Guangdong Province may also be negligible, referred to the results from the western US and Chinese regulatory standards for radioactive discharges from NPPs.

**C2 Biospheric exchange.** Biospheric carbon fluxes associated with photosynthesis, autotrophic respiration, and annual biomass burning generally do not alter atmospheric $\Delta(^{14}C)$ levels, as the carbon exchanged through these processes largely maintains isotopic equilibrium with contemporary atmospheric $CO_2$ (Turnbull et al., 2009). In contrast, heterotrophic respiration and multi-year biomass burning (e.g., wildfire consuming legacy organic matter) release carbon fixed during periods of elevated atmospheric $\Delta(^{14}C)$, such as the 1960s nuclear bomb testing peak. This temporal decoupling between

carbon uptake and release introduces a measurable positive bias in modern $\Delta(^{14}C)$, reflecting the delayed contribution of older carbon pools. Therefore, we use estimates of heterotrophic respiration (Rh) and biomass burning (BB) fluxes to correct for biospheric influence on $C_{ff}$ calculation.

**C2.1 Heterotrophic respiration.** The heterotrophic respiration correction term ($\beta_{Rh}$) is calculated by the following equation:

$$\beta_{Rh} = \frac{C_{Rh}(\Delta_{bg} - \Delta_{Rh})}{\Delta_{bg} + 1000‰} \qquad (C1)$$

where $C_{Rh}$ is the $CO_2$ mole fraction estimated by coupling hourly FLEXPART footprints with the heterotrophic respiration fluxes extracted from the Carnegie Ames Stanford Approach Global Fire Emissions Database Version 4 (CASA-GFED4s) (Randerson et al., 2017; Van Der Werf et al., 2017). We imposed the diurnal cycle from the CASA-GFED3 (Van Der Werf et al., 2010) heterotrophic respiration fluxes (estimated as half of the ecosystem respiration, which is calculated as the difference between net ecosystem exchange and gross ecosystem exchange; [NEE − GEE]/2) onto the nearest neighbor

CASA-GFED4s monthly mean fluxes to approximate hourly resolved fluxes. By aggregating these flux estimates, we created flux maps matching the spatial resolution of the hourly FLEXPART footprints. We then calculated $C_{Rh}$ by multiplying the FLEXPART footprints with heterotrophic respiration flux maps. The simulated $C_{Rh}$ concentrations were 1.5 ± 0.7 µmol mol$^{-1}$ (range: 0.5−3.1 µmol mol$^{-1}$) in summer and 2.2 ± 0.9 µmol mol$^{-1}$ (range: 0.4−3.8 µmol mol$^{-1}$) in winter. We used a value of 40 ± 35 ‰ for the $\Delta(^{14}CO_2)$ signature of heterotrophic respiration ($\Delta_{Rh}$), based on the value of 75 ± 35 ‰

in 2015 (Graven et al., 2018) and considering a decrease of 5 ‰ per year (Zazzeri et al., 2023). The disequilibrium correction from heterotrophic respiration ($\beta_{Rh}$) were estimated to be –0.06 ± 0.03 ppm (range: –0.14 to –0.02 ppm) in summer and –0.11 ± 0.04 ppm (range: –0.20 to –0.02 ppm) in winter.

**C2.2 Biomass burning.** For the influence from biomass burning, we extracted $CO_2$ emissions from biomass burning in Guangdong Province and four cities from the CASA-GFED4s (Randerson et al., 2017; Van Der Werf et al., 2017), and $C_{bio}$

emissions from the Emissions Database for Global Atmospheric Research (EDGAR) (Edgar, 2024). Key methodological distinctions arise from the differing scopes of these datasets: CASA-GFED4s quantifies open-environment fires (i.e.,



satellite-observable combustion events including wildfires, agricultural residue burning, savanna/rangeland fires, and small fires), whereas EDGAR also incorporates emissions from closed-system combustion (e.g., industrial or residential biomass use).

Biomass burning emissions from CASA-GFED4s accounted for <2 % of Rh emissions across Guangdong Province and the four studied cities. In contrast, EDGARv2024ghg $C_{bio}$ estimates represented a substantially higher proportion of Rh emissions, ranging from 7–29 % (Guangdong), 24–92 % (Guangzhou), 16–97 % (Shenzhen), 13–38 % (Zhanjiang), to 73–248 % (Shaoguan). Given the small disequilibrium corrections for Rh, the simulated contribution of BB emissions to $C_{ff}$ estimates was negligible when using CASA-GFED4s data. However, EDGAR-derived $C_{bio}$ scenarios indicated a potentially

higher influence on $C_{ff}$ quantification, particularly in regions with high biomass combustion activity.

To estimate the biomass burning correction term ($\beta_{BB}$) using the EDGAR2024 $C_{bio}$ inventory (Edgar, 2024), we first derived total $C_{BB}$ simulations ($C_{BB}$) by applying a biomass burning fraction ($\alpha_{BB}$) to the $C_{bio}$ simulations ($C_{bio}$). This parameter $\alpha_{BB}$ represents the proportion of $C_{bio}$ emissions attributable to biomass burning:

$$C_{BB} = C_{bio} \cdot \alpha_{BB} \qquad (C2)$$

The correction term $\beta_{BB}$ was subsequently calculated as:

$$\beta_{BB} = \frac{C_{BB}(\Delta_{bg} - \Delta_{BB})}{\Delta_{bg} + 1000‰} \qquad (C3)$$

For simulated $C_{bio}$ mole fraction estimation in 2022, we implemented a three-stage process: (1) Generating 0.1°×0.1° resolution flux maps through integration of EDGAR2024 $C_{bio}$ emission fluxes with FLEXPART atmospheric transport footprints, (2) performing spatiotemporal aggregation to align with FLEXPART model output specifications, and (3)

calculating concentrations via convolution operations between transport footprints and optimized flux fields.

We adopted $\Delta(^{14}CO_2)$ signatures of −14.8 ± 2.2 ‰ (annual biomass burning) and 116.2 ± 17.6 ‰ (multi-year biomass burning) calculated in Appendix B. For 2022 $\Delta(^{14}C)$ endmembers for biomass burning, we estimated values of 116.2 ± 17.6 ‰ (0 % annual biomass), 103.1 ± 15.8 ‰ (10 %), 90.0 ± 14.1 ‰ (20 %), 76.9 ± 12.3 ‰ (30 %), 63.8 ± 10.6 ‰ (40 %), and 50.7 ± 8.9 ‰ (50 %), corresponding to incremental annual biomass burning fractions from 0 % to 50 %.

We quantified disequilibrium correction terms under the maximum biomass burning (BB) contribution scenario ($f_a$ = 0 % and $\alpha_{BB}$=100 %). The simulated $C_{BB}$ concentrations were 0.8 ± 0.7 µmol mol$^{-1}$ (range: 0.0−1.9 µmol mol$^{-1}$) in summer and 1.9 ± 1.0 µmol mol$^{-1}$ (range: 0.1−4.4 µmol mol$^{-1}$) in winter. The BB-specific correction term ($\beta_{BB}$) exhibited seasonal variations: −0.09 ± 0.08 µmol mol$^{-1}$ (range: −0.46 to −0.01 µmol mol$^{-1}$) in summer and −0.24 ± 0.12 µmol mol$^{-1}$ (range: −0.56 to −0.02 µmol mol$^{-1}$) in winter under 0 % annual biomass burning contribution. The combined correction factor β, integrating contributions from both heterotrophic respiration (Rh) and biomass burning (BB), showed broader ranges: −0.16

± 0.09 µmol mol$^{-1}$ (range: −0.55 to −0.03 µmol mol$^{-1}$) in summer and −0.35 ± 0.15 µmol mol$^{-1}$ (range: −0.72 to −0.04 µmol mol$^{-1}$) in winter.



**Appendix D: Background selection**

As we summarized in Zhou et al. (2024), Turnbull et al. (2015) concluded that for Indianapolis, a city with relatively simple

boundary conditions, the upwind background site (Tower 1) is more appropriate compared with continental and regional background sites (LEF and NWR). In contrast, for Los Angeles, a city with relatively complex boundary conditions, Newman et al. (2016) and Miller et al. (2020) tend to use the neighboring regional or continental background sites (MWO and LJO; BRW and NWR), because the upwind background within the city may be influenced by emissions from neighboring cities and therefore cannot represent the local urban background. In this study, the cities concerned are central

cities or neighboring cities of the Pearl River Delta (PRD) urban agglomeration with relatively complex boundary conditions, so we chose the nearest regional background site NL (i.e., SG5) as the background for determining the $C_{ff}$ concentrations.

The Nanling site serve as an ideal site for Guangdong's atmospheric background monitoring station due to its remote location (over 100 km from the PRD urban agglomeration), high-altitude terrain (>1,000 m) avoiding localized pollution, and strategic position as a climate/watershed boundary intercepting seasonal airflows, collectively enabling precise

monitoring of regional atmospheric transport patterns while meeting strict background station criteria for pollution isolation and cross-boundary impact assessment.

On the other hand, the "annual" $CO_2$ and $\Delta(^{14}C)$ averages at NL, which are close to those at the Jungfraujoch, are positioned in the upper-right section of the Keeling plot of $\Delta(^{14}C)$ and $CO_2$ representing the background, by comparison with the values at the Waliguan (Fig. 3). Additionally, the $\Delta(^{14}C)$ and $CO_2$ averages at NL were the highest and the lowest among the 30

sampling sites, respectively (Table A1), which are consistent with background-level criteria.



**Appendix E: Comparison of $C_{ff}$ fractions and concentrations among various cities**

**Table E1** Comparison of $C_{ff}$ and $C_{bio}$ fractions derived from $\Delta(^{14}CO_2)$ measurements in various cities

| City | Time | Background | $C_{ff}$ (%) | $C_{bio}$ (%) | References |
|------|------|-----------|-------------|--------------|-----------|
| Paris | 2010 | MHD [a] | 77 | 23 | (Lopez et al., 2013) |
| Los Angeles | 2006-2013 winter | LJO [b] | 86 | 14 | (Newman et al., 2016) |
| Los Angeles | 2006-2013 summer | LJO | 93 | 7 | (Newman et al., 2016) |
| Los Angeles | 2014.11-2016.03 | MWO [c] | 80 | 20 | (Miller et al., 2020) |
| Beijing | 2014 | WLG [d] | $75.2 \pm 14.6$ | 24.8 | (Niu et al., 2016) |
| Xiamen | 2014 | WLG | $59.1 \pm 26.8$ | 40.9 | (Niu et al., 2016) |
| Xi'an | 2014 winter | WLG | $92.7 \pm 9.7$ | $7.3 \pm 9.7$ | (Zhou et al., 2020) |
| Xi'an, urban | 2016 summer | WLG | $82.5 \pm 23.8$ | 17.5 | (Wang et al., 2018) |
| Xi'an, urban | 2016 winter | WLG | $61.8 \pm 10.6$ | 38.2 | (Wang et al., 2018) |
| Xi'an, suburban | 2016 summer | WLG | $90.0 \pm 24.8$ | 10.0 | (Wang et al., 2018) |
| Xi'an, suburban | 2016 winter | WLG | $57.4 \pm 9.7$ | 42.6 | (Wang et al., 2018) |
| Guangzhou | 2022 winter | NL [e] | $79 \pm 5$ | 21 | this study |
| Shenzhen | 2022 winter | NL | $73 \pm 6$ | 27 | this study |
| Zhanjiang | 2022 winter | NL | $59 \pm 4$ | 41 | this study |
| Shaoguan | 2022 winter | NL | $53 \pm 13$ | 47 | this study |

[a] Mace Head, [b] La Jolla, [c] Mount Wilson Observatory, [d] Waliguan, [e] Nanling




**Table E2** Comparison of $C_{ff}$ concentrations derived from $\Delta(^{14}CO_2)$ measurements in various cities

| Country | City | Time | $C_{ff}$ (µmol mol$^{-1}$) | References |
|---|---|---|---|---|
| Poland | Krakow | 1989 | 27.5 | (Kuc et al., 2003) |
| Poland | Krakow | 1994 | 10 | (Kuc et al., 2003) |
| Poland | Krakow | 2005−2009 | 1.98−2.18 | (Zimnoch et al., 2012) |
| Poland | Kasprowy Wierch | 2005−2009 | 1.95−2.08 | (Zimnoch et al., 2012) |
| Poland | Gliwice | 2011.01−2013.01 | 23−24 | (Piotrowska et al., 2020) |
| Czech Republic | Prague | 2001−2018 | 25.51 ± 11.45 | (Svetlik et al., 2010) |
| Slovakia | Bratislava | 1999−2007 | 25.56 ± 6.90 | (Svetlik et al., 2010) |
| Germany | Heidelberg | 1986−1996 | 11.09 ± 0.24 | (Levin and Rödenbeck, 2008) |
| Germany | Heidelberg | 1997−2007 | 10.92 ± 0.34 | (Levin and Rödenbeck, 2008) |
| Hungary | Debrecen | 2009/10.01 | 10−15 | (Molnár et al., 2010) |
| France | Paris | 2010.01−02 | 26.4 | (Lopez et al., 2013) |
| United Kingdom | London | 2020.06−07 | 17.3 ± 3.0 | (Zazzeri et al., 2023) |
| United States | Los Angeles | 2006−2013 | 22.9 ± 5.6 | (Newman et al., 2016) |
| United States | Los Angeles | 2014.11−2016.03 | 13.2 ± 9.4 | (Miller et al., 2020) |
| United States | Indianapolis | 2010−2015 | 10.8 ± 1.0 | (Turnbull et al., 2015) |
| China | Urumqi | 2014−2016 | 45.6 ± 12.9 | (Zhou et al., 2020) |
| China | Lanzhou | 2014−2016 | 36.4 ± 8.8 | (Zhou et al., 2020) |
| China | Xi'an | 2011−2013 | 40.1 ± 3.8 | (Zhou et al., 2022) |
| China | Xi'an | 2014−2016 | 25.7 ± 1.1 | (Zhou et al., 2022) |
| China | Suburban Xi'an | 2016 | 23.5 ± 6.5 | (Wang et al., 2018) |
| China | Near Xi'an | 2021.04−2022.03 | 13.1 ± 10.9 | (Liu et al., 2024) |
| China | Beijing | 2014 | 39.7 ± 36.1 | (Niu et al., 2016) |
| China | Beijing | 2014−2016 | 27.0 ± 0.3 | (Zhou et al., 2020) |
| China | Beijing | 2020 winter | 19.7 ± 22.0 | (Wang et al., 2022b) |
| China | Wuhan | 2014−2016 | 34.5 ± 10.0 | (Zhou et al., 2020) |
| China | Xiamen | 2014 | 13.6 ± 12.3 | (Niu et al., 2016) |
| China | Guangzhou | 2011 | 23.7 ± 12.9 | (Ding et al., 2013) |



| China | Guangzhou | 2022 | $15.3 \pm 5.2$ | this study |
|-------|-----------|------|----------------|------------|
| China | Shenzhen | 2022 | $13.7 \pm 10.2$ | this study |
| China | Zhanjiang | 2022 | $10.0 \pm 5.2$ | this study |
| China | Shaoguan | 2022 | $8.2 \pm 7.0$ | this study |


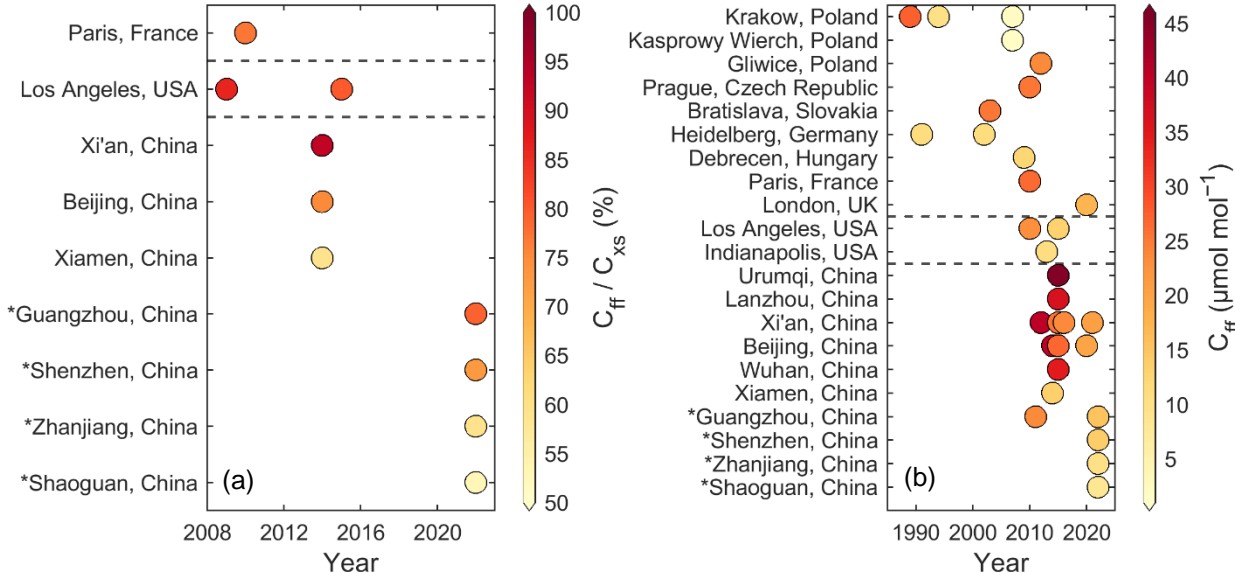

**Figure E1** Comparison of **(a)** $C_{ff}$ fractions in $C_{xs}$ and **(b)** $C_{ff}$ concentrations derived from $\Delta(^{14}CO_2)$ measurements from previous studies and this study (*) in various cities across European countries (Kuc et al., 2003; Levin and Rödenbeck, 2008;
Molnár et al., 2010; Svetlik et al., 2010; Zimnoch et al., 2012; Lopez et al., 2013; Piotrowska et al., 2020; Zazzeri et al., 2023), United States (Turnbull et al., 2015; Newman et al., 2016; Miller et al., 2020), and China (Ding et al., 2013; Niu et al., 2016; Zhou et al., 2020; Wang et al., 2022b; Liu et al., 2024; Zhou et al., 2022). In (a), Los Angeles, Beijing, Guangzhou, and Shenzhen are megacities, Xi'an is a supercity, and other are large and medium cities (Council, 2022). Values in (a) and (b) refer to Tables E1 and E2, respectively.




**Appendix F: HYSPLIT back trajectories and FLEXPART footprints**

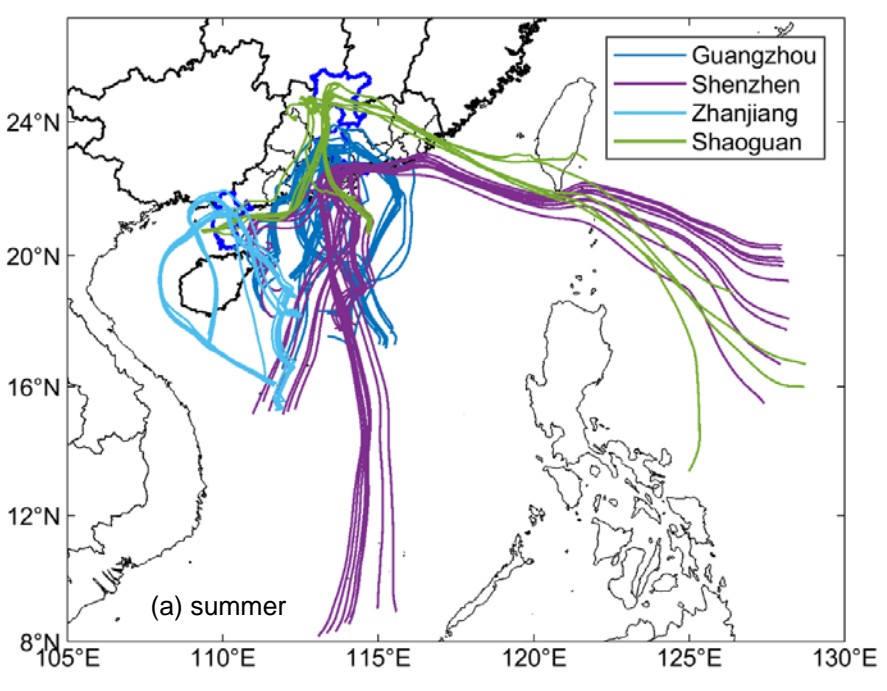

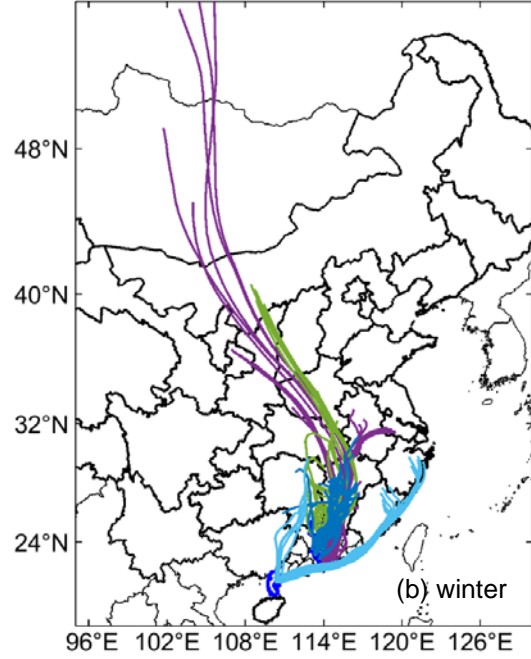



**Figure F1** HYSPLIT back trajectories for a 72-h duration for Guangzhou, Shenzhen, Zhanjiang, and Shaoguan in **(a)**
summer and **(b)** winter. National boundaries were taken from Natural Earth (https://www.naturalearthdata.com/, last
accessed: 9 March 2024).

**Figure F2** Annual mean $C_{ff}$ emissions in China from four gridded inventory datasets: **(a)** ODIAC, **(b)** EDGAR, **(c)** MIXv2,
and **(d)** MEIC, and FLEXPART footprints simulating $C_{ff}$ emissions for **(e)** Guangzhou (GZ), **(f)** Shenzhen (SZ) in winter (w)
by releasing particles at the blue sampling sites at heights from 0–100 m a.s.l. over a period of 30 days. The YRD and PRD



represent Yangtze River Delta and Pearl River Delta urban agglomeration labelled in (e) and (f). Boundaries of nations and Chinese provinces were obtained from Natural Earth (https://www.naturalearthdata.com/, last accessed: 9 March 2024).

**Appendix G: Comparison of contributions of coal, oil and natural gas to $C_{ff}$ concentrations in various cities**

**Table G1** Comparison of contributions of coal, oil and natural gas to $C_{ff}$ concentrations in various cities

| City | Time | $F_{coal}$ (%) | $F_{oil}$ (%) | $F_{ng}$ (%) | References |
|------|------|------|------|------|------------|
| Paris | 2010 | < 1 | 30 | 70 | (Lopez et al., 2013) |
| Los Angeles | 2007.10 | < 1 | 69 | 31 | (Djuricin et al., 2010) |
| Los Angeles | 2007.12 | < 1 | 61 | 39 | (Djuricin et al., 2010) |
| Los Angeles | 2008.02 | < 1 | 58 | 42 | (Djuricin et al., 2010) |
| Los Angeles | 2008.04 | < 1 | 52 | 48 | (Djuricin et al., 2010) |
| Los Angeles | 2006-2013 winter | < 1 | 68 | 32 | (Newman et al., 2016) |
| Los Angeles | 2006-2013 summer | < 1 | 55 | 45 | (Newman et al., 2016) |
| Xi'an | 2014 winter | 72.6 ± 10.4 | 13.8 ± 10.4 | 13.6 | (Zhou et al., 2014) |
| Xi'an | 2019.12-2020.01 | 54 ± 4 | 24 ± 14 | 22 ± 13 | (Wang et al., 2022b) |
| Beijing | 2020.12-2021.01 | 17 ± 10 | 28 ± 19 | 55 ± 9 | (Wang et al., 2022b) |
| Guangzhou | 2022 winter | 49 ± 25 | 29 ± 22 | 22 ± 19 | this study |
| Shenzhen | 2022 winter | 47 ± 25 | 29 ± 21 | 24 ± 20 | this study |
| Zhanjiang | 2022 winter | 43 ± 24 | 29 ± 21 | 28 ± 21 | this study |
| Shaoguan | 2022 winter | 39 ± 24 | 34 ± 23 | 27 ± 21 | this study |





## Appendix H: $R_{CO/CO2ff}$ for sites, cities and comparison

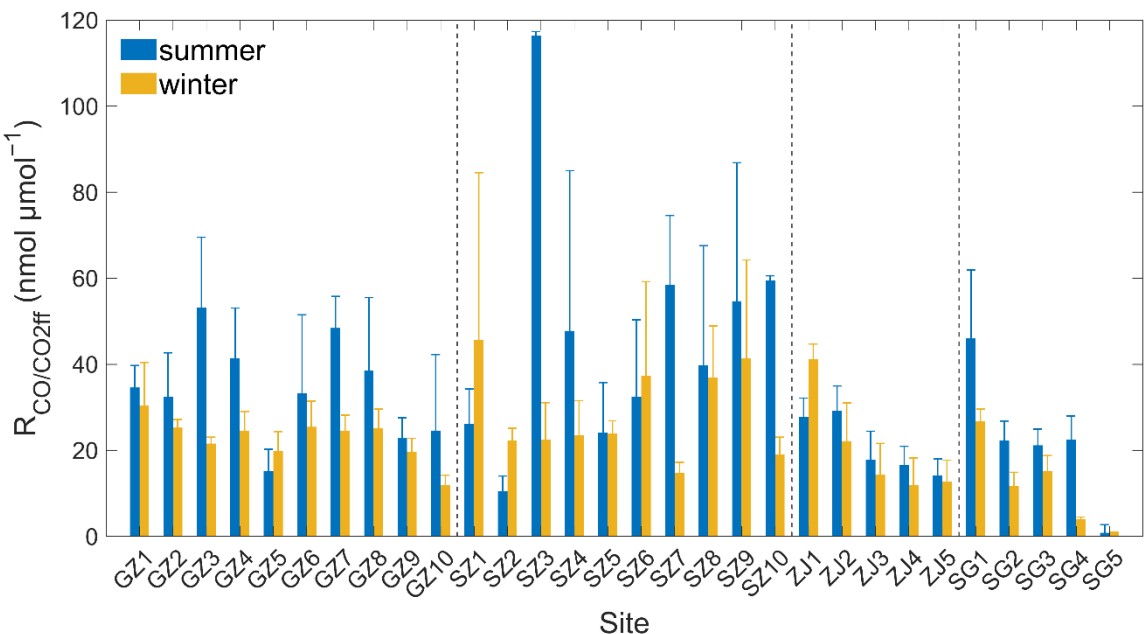

**Figure H1** $R_{CO/CO2ff}$ averages at the 30 sampling sites


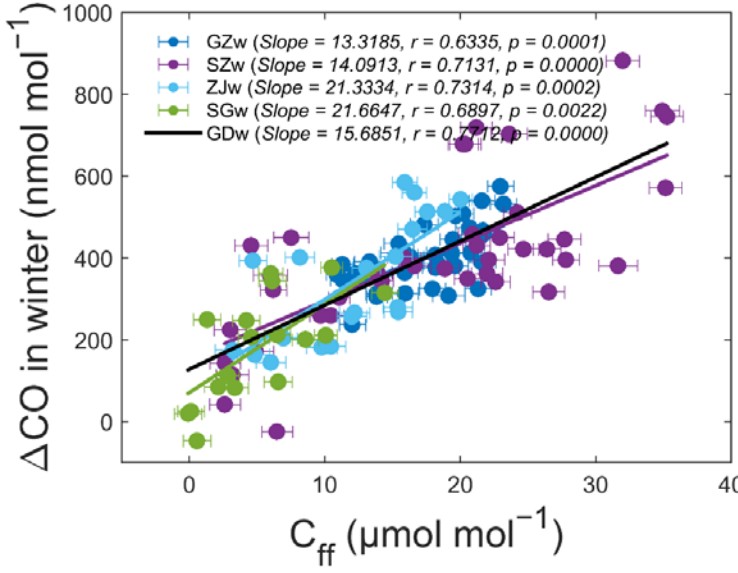

**Figure H2** $\Delta CO$: $C_{ff}$ for Guangzhou, Shenzhen, Zhanjiang, and Shaoguan in winter



**Table H1** Observational emission ratios of CO to $C_{ff}$ ($R_{CO/CO2ff}$) for China and Chinese cities

| City | Time | $R_{CO/CO2ff}$ (nmol µmol⁻¹) | Method [a] | References |
|------|------|------|------|------|
| China | 2001 | 68.8 | I | (Suntharalingam et al., 2004) |
| China | 2004−2010 | 44 ± 3 | II | (Turnbull et al., 2011) |
| China | 1998 | 56.3 | I | (Tohjima et al., 2014) |
| China | 2010 | 37.5 | I | (Tohjima et al., 2014) |
| Mainland China | 2009 | 36.3 | I | (Fu et al., 2015) |
| Yellow Sea | 2016 | 35.0 | I | (Tang et al., 2018) |
| China, CN [b] | 2014−2016 | 31 ± 8 | II | (Lee et al., 2020) |
| China, CE [b] | 2014−2016 | 36 ± 2 | II | (Lee et al., 2020) |
| China, CB [b] | 2014−2016 | 29 ± 8 | II | (Lee et al., 2020) |
| China, OB [b] | 2014−2016 | 31 ± 4 | II | (Lee et al., 2020) |
| Beijing | 2005 | 54.3 | I | (Han et al., 2009) |
| Beijing/Tianjin | 2009−2010 | 59.5 | I | (Silva et al., 2013) |
| Beijing | 2014 | 30.4 ± 1.6 | II | (Niu et al., 2018) |
| Rural Beijing | 2004 | 72.3 | I | (Wang et al., 2010) |
| Rural Beijing | 2005 | 52.5 | I | (Wang et al., 2010) |
| Rural Beijing | 2006 | 48.1 | I | (Wang et al., 2010) |
| Rural Beijing | 2007 | 43.7 | I | (Wang et al., 2010) |
| Rural Beijing | 2008 | 47.0 | I | (Wang et al., 2010) |
| Rural Beijing | 2009−2010 | 47 ± 2 | II | (Turnbull et al., 2011) |
| Xi'an | 2016 | 46 ± 13 | II | (Wang et al., 2018) |
| Near Xi'an | 2021 | 23 ± 6 | II | (Liu et al., 2024) |
| Xiamen | 2014 | 29.6 ± 3 | II | (Niu et al., 2018) |
| Guangzhou | 2009−2010 | 35.8 | I | (Silva et al., 2013) |
| Guangzhou | 2014−2017 winter | 23.8 | I | (Mai et al., 2021) |
| Guangzhou | 2022 winter | 13.3 ± 0.00002 | II | Fig. H2, this study |
| Shenzhen | 2022 winter | 14.1 ± 0.000003 | II | Fig. H2, this study |
| Zhanjiang | 2022 winter | 21.3 ± 0.00002 | II | Fig. H2, this study |



| Shaoguan | 2022 winter | $21.7 \pm 0.00004$ | II | Fig. H2, this study |

[a] by correction from $R_{CO/CO_2}$ by increased 20 % (Method I) and estimation from $\Delta^{14}C$ measurements (Method II), [b] CN represents the air masses from northeast China, CE for central eastern China around the Shandong area, CB for continental background air, and OB for ocean background.

**Code availability**

The FLEXPART 10.4 model is available at https://www.flexpart.eu/. The MixSIAR 3.1.12 model is available via GitHub at
https://brianstock.github.io/MixSIAR/index.html. The HYSPLIT model is available at https://www.arl.noaa.gov/hysplit/. In this study, commercial software such as MATLAB R2023a, and public software such as R 4.3.2 and Python 3.9 are used for data processing and result visualization.

**Data availability**

Data generated in this study are available in Supplement Dataset 1 and seasonal averages in Table A1. Additional data
related to this paper may be requested from the corresponding authors. The Carnegie Ames Stanford Approach Global Fire Emissions Database Version 4 (CASA-GFED4s) dataset is available at https://daac.ornl.gov/VEGETATION/guides/fire_emissions_v4_R1.html. The CASA-GFED3 dataset is available at http://nacp-files.nacarbon.org/nacp-kawa-01/. The Open-source Data Inventory for Anthropogenic $CO_2$ (ODIAC) is available from https://db.cger.nies.go.jp/dataset/ODIAC/. The Emissions Database for Global Atmospheric Research
(EDGAR) Global Greenhouse Gas and Air Pollutant Emissions are from https://edgar.jrc.ec.europa.eu. The Multi-resolution Emission Inventory for China (MEIC) and the Open Biomass Burning Emission Inventory for China (OBBEIC) are available from http://meicmodel.org.cn. The MIXv2 Asian emission inventory (MIXv2) is available from https://csl.noaa.gov/groups/csl4/modeldata/data/Li2023/. The National Centers for Environmental Prediction's Climate Forecast System (CFSv2) Reanalysis data that drive the FLEXPART model is available at
https://rda.ucar.edu/datasets/ds094.0/. The National Centers for Environmental Prediction's Global Data Assimilation System (GDAS) Reanalysis data that drives the HYSPLIT model is available at ftp.arl.noaa.gov/pub/archives/reanalysis.

**Author contribution**

G.Z., D.C., Jun Li, and P.L. conceived and designed the study. Almost all authors participated in the sampling organized by Jun Li and P.L.. Z.N. provided data in Beijing and Xi'an. P.D. provided data from Ding et al., (2013) in Guangzhou. R.L.
conducted the sample graphitization. Sanyuan Z. handled the $^{14}C$ measurement by AMS. P.L. and B.L. performed the



simulations. P.L. conducted the data search and analysis, and wrote the article, with contributions from G.Z., Jun Li, Z.C., Jing L., and T.Z. for revisions and improvements.

## Competing interests

The authors declare no competing interest.

## Acknowledgements

We gratefully acknowledge the research team from Professor Gan Zhang's group for their essential support in air sampling, including staff scientists, postdoctoral researchers, and graduate students. Special thanks are extended to Mr. Jiangtao Li for his dedicated assistance with field sampling and laboratory extraction procedures.

## Financial support

This study was supported by the National Natural Science Foundation of China (NSFC; nos. 42330715, 42103082, 42030715, and 42177241), Guangdong Provincial Applied Science and Technology Research and Development Program (Grant nos. 2022A1515011271, and 2022A1515011851), the Alliance of International Science Organizations (Grant no. ANSO-CR-KP-2021-05), China Postdoctoral Science Foundation (Grant no. 2022T150652), Special Research Assistant Program of the Chinese Academy of Sciences (CAS), and Director's Fund of Guangzhou Institute of Geochemistry, CAS
(Grant no. 2021SZJJ-3).

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
