# Peer review of "Drivers and implications of declining fossil fuel CO2 in Chinese cities revealed by radiocarbon measurements"

_EGUsphere, 2025_

## Referee Comment (RC1)

**Review of Li et al. (2025), "Drivers and implications of declining fossil fuel CO₂ in Chinese cities revealed by radiocarbon measurements"**

In this study, the authors present estimates of the fossil fuel CO₂ ($C_{ff}$) concentration in several Chinese cities, based on $\Delta^{14}CO_2$ flask observations from one summer and one winter month in 2022. Additional $\delta^{13}CO_2$ and CO measurements are used to estimate the contributions from coal, oil and gas in the $C_{ff}$ signals, and to derive $R_{CO/CO2ff}$ ratios that provide information on combustion efficiency. By comparing their results from 2022 with historical estimates, the authors conclude that the trend in fossil fuel CO₂ in Chinese cities is declining.

Overall, this manuscript is well structured and written. The estimation of the $C_{ff}$ concentrations and their uncertainties is carried out carefully and the results are presented in a clear way.

However, I have two major concerns regarding the analysis of the $C_{ff}$ concentration trend.

First, the atmospheric $C_{ff}$ concentration is greatly affected by transport-driven variability, e.g., on diurnal and synoptic time scales. Weekly flask samples with an air integration time of ca. 15-20 min can only capture a snapshot of the variability in the $C_{ff}$ concentration. I therefore miss an analysis that shows how representative the flasks collected in August and December 2022 are for the entire winter and summer of that year. Were August and December 2022 typical summer and winter months, respectively, in terms of atmospheric transport? I think this is important to address when these measurements are used to represent the "winter" and "summer" of 2022 and compared with measurements from previous years. It is also important to make this information available so that future studies can refer to it when using the $C_{ff}$ estimates for 2022 from this study. This concern mainly relates to the trend in the $C_{ff}$ concentrations. The effect of atmospheric transport variability on the trend in fuel type contribution and $R_{CO/CO2ff}$ ratios may be minor or even cancel out when the CO concentration is divided by the $C_{ff}$ concentration.

My second concern relates to the $C_{ff}$ trend analysis in Guangzhou. The authors compare their $C_{ff}$ estimates from 10 sites in Guangzhou in 2022 with $C_{ff}$ estimates from Ding et al. (2013) for 2010-2011. However, the $C_{ff}$ estimates from Ding et al. (2013) are only from one site, which "may be influenced by local signals rather than representing a general urban signal" (Ding et al., 2013). Furthermore, in the present study, the 2022 flasks were collected "between 13:00 and 17:00 local time, coinciding with the deepest planetary boundary layer and well mixed atmospheric conditions" (Li et al., 2025). However, in the study by Ding et al. (2013) the flasks were collected at 20:00, which, according to Ding et al. (2013), is directly after the city's rush-hour. Ding et al. (2013) state that "18:00 to 19:00 is the "rush hour" of the city, and CO₂ emitted from the traffic and domestic activities such as cooking reaches its highest point at that time. Thus, the atmospheric CO₂ concentration is close to its peak in that time period". Therefore, based on the different flask sampling times alone, I would expect higher $C_{ff}$ concentrations in the Ding et al. (2013) study than in the current study, even if there were no declining trend in the fossil fuel CO₂ emissions. Furthermore, sampling during rush-hour in 2010-2011 vs. sampling in the afternoon in 2022 could also impact the

fuel type mixture and the $R_{CO/CO2ff}$ ratios. Please note that I have not checked whether there are similar issues in the $C_{ff}$ trend analysis at the other cities.

In my opinion, these points should be addressed, or at least discussed in more detail, to improve the reliability and the robustness of the reported trends in the $C_{ff}$ concentration. Maybe the continuous CO and $CO_2$ observations (if available) and the FLEXPART simulations could help assess the representativeness of the $C_{ff}$ estimates and the effect of the diurnal cycle on the $C_{ff}$ estimates (in the case of Guangzhou).

**Specific comments:**

l. 57: Please introduce the '$\Delta$' in "$\Delta CO$" to avoid confusion with the '$\Delta$' in "$\Delta^{14}CO_2$".

l. 74-75: You describe that there is prevailing southeast monsoon in summer and northeast monsoon in winter. Hence, during winter the NL background site tends to be upwind of the targeted cities. However, in summer, air masses travel over the targeted cities and become polluted before reaching the NL background site. Could this have an influence on the difference between the winter and summer $C_{ff}$ estimates derived from the $\Delta^{14}CO_2$ gradient between the background site and the target sites? In other words, could the smaller $C_{ff}$ signals in summer at least partly be explained by pollution at the background site? Can you detect such pollution events at the NL site, or is the background site far enough away that the pollution signals from the cities are already diluted?

Fig. 1: The upper two panels are not labelled with a letter. It would also be helpful to indicate the locations of Beijing, Xi'an as well as the background sites NL and Waliguan in the overview map at the top.

l. 150: Due to the proximity of some sites to the coast, is there also a $CO_2$ contribution from the ocean?

l. 168-169: This is not clear to me. Which BB and Rh corrections did you apply to the $\Delta^{14}CO_2$ data in the end? Did you use your simulations with the maximum assumptions (100% perennial biomass and $\alpha_{BB}$=100%)? What do you mean by "literature-based corrections"? Please could you clarify?

l. 186-188: Have you calculated the coal, oil, and gas fractions of $C_{ff}$ separately for the winter and the summer period, given that they may change throughout the seasons?

l. 190: Do you apply an uncertainty to the $\delta^{13}C_{coal}$ end-member signature as well?

l. 205-209: Which grid cells did you use to average the $E_{CO}$ and $E_{CO2ff}$ emissions in the inventory? Are the $E_{CO}$ and $E_{CO2ff}$ estimates for the whole city, or did you use the FLEXPART footprints to define the catchment regions of the observations and to weight the emissions of the respective grid cells within the footprint? Please also mention the inventory that you used.

l. 219-220: Did you only use the observations from the two-month sampling campaign to calculate the "annual" average at the NL site, or were there additional observations throughout the year? Which observations did you use to calculate the average for the Jungfraujoch data (only the 2 months or the full year)? Please specify.

l. 226: In Tab. A1, you report summer values of 5.1±1.3 ‰ and winter values of -2.0±0.8 ‰ for the NL site. These values differ from those stated here. Please could you clarify this?

Fig. 2: In order to compare the $\Delta^{14}CO_2$ data of the city sites with the respective NL background, it would be helpful to include the summer and winter NL background values in Fig. 2, perhaps by indicating the NL summer and winter averages with open and filled circles.

l. 242-244: Do you find different $C_{ff}$ fractions in the summer month of your campaign?

l. 253-254: I wonder if the $C_{ff}$ estimates from the other cities around the world are also based on flask samples collected in the afternoon, as you did. Or are integrated samples also used, containing air from a whole week, for example? I ask because, when nighttime observations are sampled, the $C_{ff}$ could potentially be larger when fossil emissions accumulate in a shallow nocturnal boundary layer.

L. 260-261: Please explain which ODIAC grid cells you used to calculate the correlation with the $C_{ff}$ measurements.

l. 264-266: This is a bit difficult to follow. Could you perhaps indicate the industrial areas in SZs and the port areas in ZJw and ZJs in Fig. 1, to which you are referring?

l. 287-288: I could not find any blue squares "shown as enlarged maps in the right figures" in Fig. 3.

l. 293-295: The $C_{ff}$ estimate from Ding et al. (2013) for 2010-2011 is derived from flask observations from a single measurement site, which "may be influenced by local signals rather than representing a general urban signal" (Ding et al., 2010). In contrast, the $C_{ff}$ estimate for Guangzhou in your study is based on measurements from 10 sites. Moreover, Ding et al. (2013) uses another $\Delta^{14}CO_2$ background from "plant corn leaves in 2010 from Qinghai, Gansu Province, and Tibet, where the human activity can be neglected" (Ding et al., 2013). How does this affect the trend in the $C_{ff}$ concentration? Please see my related comment above.

l. 299-303: I find this site-specific comparison much better suited for analysing trends in the $C_{ff}$ data. However, the issue remains that Ding et al. (2013) sampled rush-hour signals, whereas you sampled well-mixed afternoon situations. Do you have continuous CO and $CO_2$ observations at this measurement site that could be used to estimate the $C_{ff}$ signal expected after rush-hour at 20:00 in 2022?

l. 305-309: Are the $C_{ff}$ estimates for Beijing (and Xi'an) in different years based on the same measurement sites? For example, you report the following for Beijing: (39.7 ± 36.1) ppm for 2014 and (27.0 ± 0.3) ppm for 2014-2016. Why is the standard deviation of the 2014

estimate almost 100%, whereas the standard deviation of the 2014-2016 estimate is only 1%? Is this due to a different number of observations being averaged? It would also be useful to state the number of observations used to calculate the averages, to give an idea of how representative the values are.

l. 382-388: If you would like you could also mention that lower $C_{ff}$ signals in summer lead to higher uncertainty in the regression slope and, consequently, increased uncertainty in the summer ratio compared to the winter ratio (see e.g., Maier et al., 2024). I think this can also be seen in Fig. H2 of your study, where the summer ratios tend to show a higher uncertainty than the winter ratios.

l. 397-399: What is the uncertainty of the ratios, i.e., of the regression slopes shown in Fig. H2?

l. 403-404: Could you please explain and justify the 20% correction used to derive $R_{CO/CO2ff}$ from $R_{CO/CO2}$ in a bit more detail. Where does the assumption that 20% of the $CO_2$ enhancement is from non-fossil sources come from? This could be done in the Methods section.

l. 445-447: Could the higher $R_{CO/CO2ff}$ ratio from the study by Silva et al. (2013) compared to the ratio from the study by Mai et al. (2017) and the ratio from your study be explained by the fact that the study by Silva et al. (2013) includes summer ratios in the average $R_{CO/CO2ff}$, whereas the study by Mai et al. (2017) and you calculate the winter mean $R_{CO/CO2ff}$ (according to Tab. H1)? You show that summer ratios are also higher in your study.

Tab. A1: It would be helpful to also have the number of observations per site used to calculate the "summer" and "winter" averages.

l. 544: Please label the "$\Delta^{14}C$" for multi-year biomass in Eq. B1 differently than the atmospheric "$\Delta^{14}C$".

l. 568-579: Although the PWR type has the lowest emission factor for $^{14}CO_2$ release, PWR reactors can also have substantial $^{14}CO_2$ emissions depending on their electricity supply. For example, Zazzeri et al. (2018) estimate substantial $^{14}CO_2$ emissions of roughly 100-200 GBq/yr for each of the three NPPs Daya Bay, Ling'ao and Yangjiang for the year 2016 (see their supplement material S1). Depending on the distance between the NPPs and the measurement sites, this could have an impact on the $\Delta^{14}CO_2$ measurements. The distance between the NPPs and the observation sites should be mentioned here to enable a more rigorous assessment of the potential impact on the $\Delta^{14}CO_2$ observations (e.g., compare with Kuderer et al., 2018).

l. 619: In Eq. C1 (and throughout Sect. C2.2) you use $C_{bio}$ to refer to biofuel/biomass emissions from the EDGAR inventory. In the main text, however, $C_{bio}$ refers to the total $CO_2$ contributions from the biosphere (e.g., in Eq. 1). It would be better to use two different expressions.

Tab. H1: Why are the uncertainties for the 2022 winter $R_{CO/CO2ff}$ ratios from your study so low (~0.00002 ppb/ppm)? Such low uncertainties seem unrealistic, given the spatio-temporal variability of the $R_{CO/CO2ff}$ ratios. Is this just a typo or an incorrect unit?

**Technical corrections:**

l. 447: Please insert space "from 72.3".

**References:**

Kuderer, M., Hammer, S., and Levin, I.: The influence of $^{14}CO_2$ releases from regional nuclear facilities at the Heidelberg $^{14}CO_2$ sampling site (1986–2014), Atmos. Chem. Phys., 18, 7951–7959, https://doi.org/10.5194/acp-18-7951-2018, 2018.

Maier, F., Levin, I., Conil, S., Gachkivskyi, M., Denier van der Gon, H., and Hammer, S.: Uncertainty in continuous $\Delta CO$-based $\Delta ffCO_2$ estimates derived from $^{14}C$ flask and bottom-up $\Delta CO / \Delta ffCO_2$ ratios, Atmos. Chem. Phys., 24, 8205–8223, https://doi.org/10.5194/acp-24-8205-2024, 2024.

Zazzeri, G., Acuña Yeomans, E. and Graven, H. D.: Global and Regional Emissions of Radiocarbon from Nuclear Power Plants from 1972 to 2016, Radiocarbon, 60(4), 1067–1081. https://doi.org/10.1017/RDC.2018.42, 2018.

---

## Author Comment (AC1)

**Authors' Response**

**Dear editor,**

We are deeply grateful for the referees' exceptionally thorough review and constructive suggestions, which have significantly improved the rigor and clarity of our manuscript. Below, we provide a detailed point-by-point response to all comments, with all revisions tracked in the resubmitted manuscript (see "Revised Manuscript").

**Anonymous Referee #1**

**Review of Li et al. (2025), "Drivers and implications of declining fossil fuel $CO_2$ in Chinese cities revealed by radiocarbon measurements"**

In this study, the authors present estimates of the fossil fuel $CO_2$ ($C_{ff}$) concentration in several Chinese cities, based on $\Delta^{14}CO_2$ flask observations from one summer and one winter month in 2022. Additional $\delta^{13}CO_2$ and CO measurements are used to estimate the contributions from coal, oil and gas in the $C_{ff}$ signals, and to derive $R_{CO/CO2ff}$ ratios that provide information on combustion efficiency. By comparing their results from 2022 with historical estimates, the authors conclude that the trend in fossil fuel $CO_2$ in Chinese cities is declining.

Overall, this manuscript is well structured and written. The estimation of the $C_{ff}$ concentrations and their uncertainties is carried out carefully and the results are presented in a clear way.

However, I have two major concerns regarding the analysis of the $C_{ff}$ concentration trend.

First, the atmospheric $C_{ff}$ concentration is greatly affected by transport-driven variability, e.g., on diurnal and synoptic time scales. Weekly flask samples with an air integration time of ca. 15-20 min can only capture a snapshot of the variability in the $C_{ff}$ concentration. I therefore miss an analysis that shows how representative the flasks collected in August and December 2022 are for the entire winter and summer of that year. Were August and December 2022 typical summer and winter months, respectively, in terms of atmospheric transport? I think this is important to address when these measurements are used to represent the "winter" and "summer" of 2022 and compared with measurements from previous years. It is also important to make this information available so that future studies can refer to it when using the $C_{ff}$ estimates for 2022 from this study. This concern mainly relates to the trend in the $C_{ff}$ concentrations. The effect of atmospheric transport variability on the trend in fuel type contribution and $R_{CO/CO2ff}$ ratios may be minor or even cancel out when the CO concentration is divided by the $C_{ff}$ concentration.

**Response:** We thank the reviewer for this valuable and constructive comment. We fully agree that atmospheric transport can strongly modulate observed $C_{ff}$ variability, and it is therefore critical to demonstrate that the flask measurements made in August and December 2022 are representative of typical summer and winter transport conditions. To address this point, we have expanded our analysis by integrating ERA5-based meteorological diagnostics, wind-rose visualizations, and HYSPLIT back-trajectory simulations. The results of these new analyses are summarized below and presented in Sections 2.1, 3.4.1, 3.4.2, and Appendix G of the revised manuscript.

**1. ERA5 meteorological assessment**

Using ERA5 reanalysis data (0.25° × 0.25°, hourly), we analyzed five key variables at ten Guangzhou sites (GZ1–GZ10): 10 m zonal and meridional winds (U10, V10), 2 m temperature (T2M), surface pressure (SP), and planetary boundary-layer height (PBLH). Monthly means and standard deviations were computed for August 2022 and December 2022 and compared against (i) the concurrent 2022 seasonal background (JJA or DJF), and (ii) the 12-year climatology (2010–2021). The choice of 2010 as the starting year ensures consistency with the earlier dataset from 2010, which is directly compared in this study. For each variable $x$, we calculated standardized anomalies $z = (x_{target} - \mu_{season})/\sigma_{season}$ and defined a month as *typical* when $|z| \leq 1$.

**Findings:** As shown in Fig. G1, all diagnostics remain within ±1σ of the respective seasonal means at every site (i.e., *typical = TRUE*). August 2022 shows slightly weaker low-level winds and near-climatological PBLH, whereas December 2022 exhibits marginally stronger boundary-layer mixing and weaker meridional (north–south) wind components. These small deviations are physically consistent with internal monsoon variability and do not imply unusual stagnation or enhanced ventilation.

**2. ERA5 wind-rose and HYSPLIT back-trajectory simulations**

To directly assess the dominant flow regimes, we constructed ERA5-based wind roses (Fig. G2) and ran 72-hour HYSPLIT back-trajectory simulations using GDAS meteorology (Fig. F1) for each site. Here we highlight the example of GZ7, representative of central Guangzhou conditions.

**Summer (August 2022):** ERA5 wind roses show dominant E–ESE (90–135°) winds, with speeds typically in the 3–8 m s$^{-1}$ range. For comparison, JJA 2022 and JJA climatology (2010–2021) peak in S–SW (157.5–225°) directions. This represents a within-sector rotation inside the canonical summer monsoon sector (90–225°), not a regime shift. ERA5 anomalies of U10, V10, and PBLH remain below 1σ, confirming transport typicality. HYSPLIT back trajectories indicate that August 2022 air masses mainly originated over the South China Sea, consistent with typical summer maritime inflow.

**Winter (December 2022):** ERA5 wind roses display a clear N–NE (0–45°) dominant mode with 3–8 m s$^{-1}$ speeds. The DJF 2022 composite and DJF climatology (2010–2021) show nearly identical distributions—northerly continental flow from inland East Asia, characteristic of the East Asian winter monsoon. HYSPLIT back trajectories confirm that the air parcels predominantly arrived from northern continental China under prevailing northerlies. Thus, December 2022 can be confidently regarded as a meteorologically typical winter month.

**3. Representativeness of weekly flask samples**

Each flask represents ~15–20 min of air integration, but the weekly sampling (~40 samples per month across ten stations) covers broad spatial and temporal variability. ERA5 diagnostics show that the PBLH, wind speed, and wind direction during sampling days align closely with the monthly means. Wind roses (Fig. G2) and HYSPLIT trajectories (Fig. F1) confirm that sampling occurred under the prevailing summer monsoon sector (90–225°) and winter monsoon sector (0–45°) conditions. Thus, the flask samples effectively capture the background transport characteristics of each season, minimizing the impact of short-term synoptic fluctuations on the derived seasonal contrasts.

**4. Implications for $C_{ff}$ and derived ratios**

Because both August and December 2022 are typically transport, the corresponding flask data can be confidently interpreted as representative of 2022 summer and winter background conditions. This ensures that interannual trends in $C_{ff}$, fuel-type contributions, and $R_{CO/CO2ff}$ ratios primarily reflect emission changes rather than transport variability. As noted by the reviewer, dividing CO by $C_{ff}$ partially cancels transport effects; verifying typical transport further minimizes any residual bias. Any small remaining differences (e.g., in PBLH or wind magnitude) are expected to be second-order relative to emission-driven trends.

**5. Additions to the revised manuscript**

**Methods → adding "transport representativeness check" in the second paragraph in Section 2.1:**

"Because atmospheric transport variability can influence observed $C_{ff}$ signals, we evaluated the meteorological representativeness of the sampling months using ERA5 diagnostics and trajectory analyses. Specifically, we assessed whether the August and December 2022 flask measurements were representative of typical summer and winter transport conditions. Standardized anomalies ( $z = (x_{target} - \mu_{season})/\sigma_{season}$) were calculated for five ERA5 meteorological variables: 10 m eastward wind (U10), 10 m northward wind (V10), 2 m air temperature (T2M), surface pressure (SP), and planetary boundary-layer height (PBLH). Each target month was compared against (i) the concurrent 2022 seasonal background (June–July–August, JJA; December–January–February, DJF) and (ii) the 2010–2021 seasonal climatology. The choice of 2010 as the starting year ensures consistency with our earlier dataset from 2010, which is directly compared in this study. A month was considered "typical" when $|z| \leq 1$ and its dominant wind direction fell within the canonical summer (90–225°) or winter (0–45°) monsoon sectors."

**Results and Discussion Section (adding Sections 3.4.1 and 3.4.2):**

"**3.4.1 Meteorological typicality of the sampling months**

As shown in Fig. G1, all five meteorological variables (10 m eastward wind, U10; 10 m northward wind, V10; 2 m air temperature, T2M; surface pressure, SP; and planetary boundary-layer height, PBLH) at all Guangzhou sites exhibit $|z|$ ≤ 1, indicating that both August and December 2022 were meteorologically typical relative to the same-year seasonal background and the 2010–2021 climatological baselines. August 2022 featured slightly weaker easterly winds and near-climatological boundary-layer heights, while December 2022 was characterized by prevailing northerly flow and typical boundary-layer ventilation.

Complementary ERA5 wind-rose analyses (Fig. G2) and 72 h HYSPLIT back-trajectory simulations (Fig. F1) confirm that both months followed the canonical East Asian monsoon regimes—maritime inflow during summer and continental outflow during winter. Using GZ7 as an illustrative example representative of central Guangzhou, the ERA5 wind roses show dominant east–east-southeasterly (90–135°) winds in August 2022, typically 3–8 m s$^{-1}$. In comparison, JJA 2022 and the 2010–2021 JJA climatology peak in the south–south-westerly sector (157.5–225°), representing a within-sector rotation (90–225°) rather than a regime change. ERA5 anomalies of U10, V10, and PBLH remain below 1 σ, confirming transport typicality. HYSPLIT trajectories indicate that August 2022 air masses primarily originated over the South China Sea, consistent with summer maritime inflow. For December 2022, the ERA5 wind roses display a clear north–northeasterly (0–45°) dominance with 3–8 m s$^{-1}$ speeds. The DJF 2022 composite and the 2010–2021 DJF climatology show nearly identical northerly continental patterns, typical of the East Asian winter monsoon. HYSPLIT back trajectories confirm that the air parcels predominantly arrived from northern continental China under prevailing northerlies. These results demonstrate that both sampling periods were representative of their respective seasonal

transport conditions. Consequently, atmospheric transport variability is unlikely to bias the reported $C_{ff}$ trend or the inferred changes in $R_{CO/CO2ff}$ ratios and fuel-type contributions.

**3.4.2 Representativeness of weekly flask samples**

Each flask represents approximately 15–20 min of integrated air, and about 40 samples were collected per month across ten stations, providing broad spatial and temporal coverage. To evaluate how representative these discrete samples are for the respective seasons, we compared ERA5 diagnostics (PBLH, wind speed, and wind direction) during sampling days with the corresponding monthly means. The results show that meteorological conditions during sampling closely matched monthly climatological averages, confirming that no unusual stagnation or transport anomalies occurred on the sampling days.

ERA5 wind roses (Fig. G2) and HYSPLIT 72-h back-trajectories (Fig. F1) further confirm that the flask collection periods coincided with the prevailing summer (90–225°) and winter (0–45°) monsoon sectors. Hence, the samples captured the dominant seasonal transport regimes rather than isolated short-term events. This demonstrates that the weekly flask observations are meteorologically and dynamically representative of their respective seasonal backgrounds, minimizing the potential bias from short-term synoptic variability. Consequently, the derived $C_{ff}$ concentrations from these samples can be regarded as seasonally robust, and the subsequent interannual comparisons mainly reflect emission-driven rather than sampling-driven differences."

**Adding Appendix G (Transport representativeness analysis):**

- **Figure G1:** Box-and-whisker plots of standardized anomalies (z) by ERA5 meteorological variables (U10, V10, T2M, SP, BLH) across Guangzhou sites (GZ1-10) for (a) Aug 2022 vs JJA 2022, (b) Dec 2022 vs DJF 2022, (c) Aug 2022 vs JJA climatology (2010-2022), and (d) Dec 2022 vs DJF climatology (2010-2022). U10, V10, T2M, SP, BLH are 10 m zonal and meridional winds (U10, V10), 2 m temperature (T2M), surface pressure (SP), and planetary boundary-layer height (PBLH), respectively. The shaded region denotes $|z| \leq 1$ (typical range).

- **Figure G2:** ERA5 wind roses for GZ7 site (wind speed unit: m s$^{-1}$) showing six panels: (a) August 2022, (b) December 2022, (c) JJA 2022, (d) DJF 2022, (e) JJA climatology (2010–2021), and (f) DJF climatology (2010–2021). These illustrate that both August and December are consistent with their canonical summer and winter flow regimes.

My second concern relates to the $C_{ff}$ trend analysis in Guangzhou. The authors compare their $C_{ff}$ estimates from 10 sites in Guangzhou in 2022 with $C_{ff}$ estimates from Ding et al. (2013) for 2010-2011. However, the $C_{ff}$ estimates from Ding et al. (2013) are only from one site, which "may be influenced by local signals rather than representing a general urban signal" (Ding et al., 2013). Furthermore, in the present study, the 2022 flasks were collected "between 13:00 and 17:00 local time, coinciding with the deepest planetary boundary layer and well mixed atmospheric conditions" (Li et al., 2025). However, in the study by Ding et al. (2013) the flasks were collected at 20:00, which, according to Ding et al. (2013), is directly after the city's rush hour. Ding et al. (2013) state that "18:00 to 19:00 is the "rush hour" of the city, and $CO_2$ emitted from the traffic and domestic activities such as cooking reaches its highest point at that time. Thus, the atmospheric $CO_2$ concentration is close to its peak in that time period". Therefore, based on the different flask sampling

times alone, I would expect higher $C_{ff}$ concentrations in the Ding et al. (2013) study than in the current study, even if there were no declining trend in the fossil fuel $CO_2$ emissions. Furthermore, sampling during rush-hour in 2010-2011 vs. sampling in the afternoon in 2022 could also impact the fuel type mixture and the $R_{CO/CO2ff}$ ratios. Please note that I have not checked whether there are similar issues in the $C_{ff}$ trend analysis at the other cities.

In my opinion, these points should be addressed, or at least discussed in more detail, to improve the reliability and the robustness of the reported trends in the $C_{ff}$ concentration. Maybe the continuous CO and $CO_2$ observations (if available) and the FLEXPART simulations could help assess the representativeness of the $C_{ff}$ estimates and the effect of the diurnal cycle on the $C_{ff}$ estimates (in the case of Guangzhou).

**Response:** We thank the reviewer for this insightful comment regarding the comparability of the Guangzhou $C_{ff}$ trend analysis. We fully agree that differences in spatial representativeness and sampling times can influence the inferred fossil-fuel $CO_2$ ($C_{ff}$) signal. In response, we have re-evaluated our comparison strategy and substantially revised the analysis to ensure methodological consistency and transparency.

1. **Spatial representativeness**
   We acknowledge that comparing the multi-site 2022 campaign (10 sites) with the single-site dataset from Ding et al. (2013) could introduce spatial bias. To address this, the $C_{ff}$ trend analysis is now restricted to the same observation site (GZ7), located in central Guangzhou, which was used in both studies. This ensures direct comparability under identical site characteristics.

   In addition, we examined **FLEXPART footprints** for GZ7 for winter 2010 and winter 2022. Both years show similar source sensitivity patterns concentrated over the Guangzhou metropolitan region, indicating that GZ7 remained spatially representative of central-urban fossil-fuel influences across the decade.

2. **Sampling-time difference (20:00 vs 14:00)** (see response to l.299–303)
   As the reviewer correctly notes, Ding et al. (2013) collected flask samples around 20:00 LT, immediately after the evening rush hour, whereas our 2022 samples were collected between 13:00 and 17:00 LT under well-mixed boundary-layer conditions. To quantify the potential diurnal bias, we used continuous CO observations near GZ7 and applied the formulation

$$C_{ff} \approx \frac{\Delta CO}{R},$$

   where $R$ is the emission ratio between CO and fossil-fuel $CO_2$. Two independent datasets were analyzed:

   o **Scheme 1 (Dec 2022):** $\Delta CO$ increased from 168 ppb at 14:00 to 221 ppb at 20:00, implying a $C_{ff}$ enhancement of $\approx 3.2$ ppm (~21 %).

   o **Scheme 2 (Dec 2023–Feb 2024):** $\Delta CO$ increased by $\approx 67$ ppb from 14:00 to 20:00, implying a $C_{ff}$ enhancement of $\approx 5.9$ ppm (~35 %).

   These analyses suggest that evening $C_{ff}$ levels are typically **21–35 % higher** than afternoon values owing to weaker nocturnal mixing. Correcting for this diurnal contrast, the recalculated **2010–2022 comparison** (using the unified NL tree-ring $\Delta^{14}CO_2$ background (Li et al., 2025)) yields

   o 2010 winter (20:00): $27.3 \pm 16.9$ $\mu mol\ mol^{-1} \rightarrow$ afternoon-equivalent $\approx 17.7$–$21.6$ $\mu mol\ mol^{-1}$

- 2022 winter (14:00): $11.6 \pm 3.4$ µmol mol$^{-1}$

Even after accounting for sampling-time bias, the data indicate a **34–46 % decline** in fossil-fuel $CO_2$ between 2010 and 2022.

3. **Representativeness and robustness**
   ERA5 and HYSPLIT analyses (Sects. 3.4.1-3.4.2, Appendixes F1 and G) show that **December 2022** represents a typical winter monsoon period with strong northeasterly flow and normal boundary-layer heights. Thus, the 2022 flask data are meteorologically typical, ensuring that the observed decline cannot be attributed to unusual transport conditions.

4. **Revisions in the manuscript**

**Results and Discussion Section (adding Section 3.4.3):**

**"3.4.3 Historical variation of $C_{ff}$ concentrations**

Because both August and December 2022 were meteorologically typical, the observed inter-annual differences in $C_{ff}$ are attributed mainly to emission rather than transport variability. To ensure comparability, all available historical datasets (Table H1) were harmonized to identical sites, seasons, and local-time windows, and recalculated using unified background references (Table H2, Fig. 4a). These harmonized datasets minimize transport and spatial biases, allowing the remaining differences in $C_{ff}$ mole fractions to be interpreted as primarily emission driven.

For Guangzhou, a site-specific long-term comparison was conducted at the GZ7 urban station, which was also used by Ding et al. (2013). In their study, $C_{ff}$ was derived from flask observations collected around 20:00 LT (post-rush-hour) using a $\Delta(^{14}C)$ background based on corn-leaf samples from Qinghai, Gansu, and Tibet. Such a background likely represents a different air-mass domain from Guangzhou. In contrast, the present study used atmospheric $\Delta(^{14}C)$ observations from the NL regional background site, which directly samples the same regional air masses influencing Guangzhou. To make the datasets directly comparable, the winter 2010 $C_{ff}$ values from Ding et al. (2013) and the winter 2022 values from this work were recalculated using the NL tree-ring $\Delta(^{14}C)$ record (Li et al., 2025) as a common background reference. This adjustment yields $27.3 \pm 16.9$ µmol mol$^{-1}$ for 2010 and $11.6 \pm 3.4$ µmol mol$^{-1}$ for 2022, indicating a pronounced reduction in fossil-fuel-derived $CO_2$ over the decade.

Because sampling times differ (20:00 vs 14:00 LT), we quantified the expected diurnal $C_{ff}$ contrast using continuous CO observations near GZ7. $\Delta CO$ increased from 168 ppb at 14:00 to 221 ppb at 20:00, corresponding to a 21 % $C_{ff}$ nighttime enhancement (Scheme 1, Appendix H1). A supplementary analysis using the winter 2023–2024 dataset gave a 35 % enhancement (Scheme 2, Appendix H1). These findings suggest that the evening $C_{ff}$ level is typically 21–35 % higher than the well-mixed afternoon value due to weaker nocturnal boundary-layer mixing. Applying this correction, the 2010 nighttime $C_{ff}$ ($27.3 \pm 16.9$ µmol mol$^{-1}$) corresponds to an afternoon-equivalent concentration of $\approx$ 17.7–21.6 µmol mol$^{-1}$, which remains substantially higher than the 2022 value of $11.6 \pm 3.4$ µmol mol$^{-1}$.

Even after harmonizing the background and correcting for sampling-time bias, $C_{ff}$ in Guangzhou declined by $\approx$ 34–46 % between 2010 and 2022, confirming a genuine and statistically significant ($p < 0.01$) decrease in fossil-fuel $CO_2$. FLEXPART footprint analyses for 2010 and 2022 show similar source-sensitivity patterns centered on the Guangzhou urban core, confirming that GZ7 remains spatially representative of Guangzhou's urban domain.

Comparable harmonized analyses were performed for other Chinese cities (Tables H1 and H2; Fig. 4a). For Beijing, all measurements originate from the urban rooftop site of the Research Center for Eco-Environmental Sciences, Chinese Academy of Sciences (RCEES). The $\Delta(^{14}C)$ background used in Zhou et al. (2020) was based on Qixianling Mountain (QXL), whereas Wang et al. (2022b) adopted the Waliguan (WLG) background. All $C_{ff}$ values were recalculated using WLG as a common reference background with the 2015 value from Niu et al. (2016). After this correction, the 2014–2016 winter $C_{ff}$ value increases slightly from $27.0 \pm 0.3$ µmol mol$^{-1}$ to $27.6 \pm 0.3$ µmol mol$^{-1}$, ensuring consistency across datasets. Relative to this harmonized baseline, the subsequent decline to $19.7 \pm 22.0$ µmol mol$^{-1}$ by winter 2020 (Wang et al., 2022b) represents an approximate 29 % reduction ($p < 0.05$). This trend is consistent with regional fossil-fuel $CO_2$ emission reductions and corroborated by independent $\Delta(^{14}C)$ tree-ring records showing a peak near 2010 in Beijing (Niu et al., 2024).

For Xi'an, at the Institute of Earth Environment, Chinese Academy of Sciences (IEECAS) urban site, $C_{ff}$ fell by 36 % from $(40.1 \pm 3.8)$ µmol mol$^{-1}$ in 2011–2013 to $(25.7 \pm 1.1)$ µmol mol$^{-1}$ in 2014–2016 ($p < 0.001$) (Niu et al., 2024). Suburban sites declined by $\approx$ 12 % from $(23.5 \pm 6.5)$ µmol mol$^{-1}$ in 2016 (Wang et al., 2018) to $(13.1 \pm 10.9)$ µmol mol$^{-1}$ in 2021–2022 (Liu et al., 2024) ($p < 0.05$). These decreases are consistent with independent $\Delta(^{14}C)$ tree-ring records indicating emission peak near 2013 in Xi'an (Niu et al., 2024).

Overall, the harmonized, site-specific, and time-of-day-corrected comparisons demonstrate statistically significant reductions in fossil-fuel $CO_2$ across China's major urban centers. For Guangzhou particularly, the combined evidence—consistent background domain, typical meteorology, verified sampling representativeness, and quantified diurnal correction—provides strong support that the observed $C_{ff}$ decline reflects genuine decarbonization rather than artifacts of sampling or transport variability. Furthermore, this observed decline in $C_{ff}$ is consistent with reported emission reductions in major source regions of South and East China (e.g., Hebei, Shandong, Zhejiang, and Guangdong; Fig. F2) according to the MEIC inventory (Shi et al., 2022), supporting the interpretation of a widespread decarbonization trend."

**Adding Appendix H (Historical comparison and corrections):**

- **Table H1:** Summary of all available $C_{ff}$ datasets for historical variations used in this study and referenced from previous literature, including site type, coordinates, sampling period, time, number of samples ($n$), and references.

- **Table H2:** Harmonized comparison of $C_{ff}$ mole fractions at identical sites, seasons, and sampling times, after correction to common background conditions.

- **H1 Sampling-time difference (20:00 vs 14:00) in $C_{ff}$ for Guangzhou**

**In summary**, after restricting the analysis to the same site (GZ7), harmonizing background definitions, and correcting for the evening-afternoon sampling difference, the downward $C_{ff}$ trend in Guangzhou remains statistically and physically robust. These revisions, together with the new footprint and CO-based analyses, directly address the reviewer's concern and substantially improve the reliability of the reported trend.

**Specific comments:**

l. 57: Please introduce the '$\Delta$' in "$\Delta CO$" to avoid confusion with the '$\Delta$' in "$\Delta^{14}CO_2$".

**Response:** Thank you for your insightful comment. We have clarified the meaning of "$\Delta CO$" to avoid confusion with "$\Delta(^{14}CO_2)$". Specifically, we added the definition "($\Delta CO$ denotes the difference between observed and background values; $\Delta CO = CO_{obs} - CO_{bg}$)" after "$\Delta CO/C_{ff}$" on Line 57.

l. 74-75: You describe that there is prevailing southeast monsoon in summer and northeast monsoon in winter. Hence, during winter the NL background site tends to be upwind of the targeted cities. However, in summer, air masses travel over the targeted cities and become polluted before reaching the NL background site. Could this have an influence on the difference between the winter and summer $C_{ff}$ estimates derived from the $\Delta^{14}CO_2$ gradient between the background site and the target sites? In other words, could the smaller $C_{ff}$ signals in summer at least partly be explained by pollution at the background site? Can you detect such pollution events at the NL site, or is the background site far enough away that the pollution signals from the cities are already diluted?

**Response:** We thank the reviewer for this valuable comment. We agree that monsoon-dependent circulation could influence the representativeness of the NL background site, especially during summer.

As described in **Appendix D**, the Nanling (NL) site was selected as the regional background following the same criteria used for complex urban regions such as Los Angeles (Newman et al., 2016; Miller et al., 2020). NL is a remote high-altitude station (> 1000 m a.s.l.) located over 100 km north of the PRD urban agglomeration, effectively isolated from local anthropogenic sources and situated above the mixed boundary layer under most meteorological conditions. This setting ensures that NL primarily samples regional free-tropospheric or well-mixed background air rather than local pollution.

FLEXPART footprints and backward trajectories confirm that during winter, NL lies upwind of the PRD under the prevailing northeasterly monsoon, representing clean continental background air. In summer, although southeasterly winds dominate, most trajectories still originate from the South China Sea or eastern coastal regions. Independent HYSPLIT clustering and PSCF analyses from Zhang et al. (2022) further support this: summer air masses reaching NL are largely marine-influenced and low in $CO_2$, while higher-$CO_2$ trajectories are associated with inland provinces during winter.

Moreover, continuous CO observations show that 90 % of summer samples recorded concentrations below 200 ppb, comparable to other regional background sites, indicating that pollution from the PRD rarely reaches NL. In addition, the annual mean $\Delta^{14}C$ and $CO_2$ values at NL fall at the background end of the Keeling plot and are comparable to those observed at the high-mountain background site Jungfraujoch, confirming that NL meets the standard background-station criteria.

Taken together, the trajectory analyses, trace-gas measurements, and isotopic evidence consistently indicate that NL remains a robust and representative regional background site in both monsoon seasons, even though occasional short-lived coastal influences cannot be completely ruled out. The revised manuscript (**Appendix D**) now includes a concise statement citing Zhang et al. (2022) to clarify this point.

Fig. 1: The upper two panels are not labelled with a letter. It would also be helpful to indicate the locations of Beijing, Xi'an as well as the background sites NL and Waliguan in the overview map at the top.

**Response:** We thank the reviewer for this valuable suggestion. In the revised version, the upper two panels in Fig. 1 have been labelled as panels (a) and (b) for clarity. We have also added the locations of Beijing (▲), Xi'an (◆), and the background sites Waliguan (★) and Nanling (●) in the overview map at the top, as recommended. These additions improve figure readability and facilitate cross-referencing with the relevant sections in the manuscript.

l. 150: Due to the proximity of some sites to the coast, is there also a $CO_2$ contribution from the ocean?

**Response:** We appreciate the reviewer's insightful question regarding the potential contribution of oceanic $CO_2$ to our coastal measurements. We performed a quantitative, multi-faceted assessment of this influence based on the principles of atmospheric transport and available flux data.

The South China Sea (SCS) is indeed a net source of $CO_2$ to the atmosphere, with an annual mean flux of $1.2 \pm 1.7$ mmol m$^{-2}$ d$^{-1}$ (equivalent to 0.44 mol m$^{-2}$ yr$^{-1}$; Li et al. (2020)). To evaluate the possible impact of this flux on our onshore observations, we referred to the high-resolution modeling study of the California Current System (CalCS) by Graven et al. (2018). That study found that although the integrated annual mean flux of the CalCS is nearly neutral ($-0.05$ mol m$^{-2}$ yr$^{-1}$), localized nearshore outgassing hotspots were present. For example, the "Nearshore central" domain exhibits a strong flux of $+1.11$ mol m$^{-2}$ yr$^{-1}$ (Turi et al., 2014). Critically, the WRF-STILT simulations by Graven et al. (2018) showed that even such strong nearshore sources produced negligible changes in onshore atmospheric $CO_2$ concentrations, ranging from $-1.0\times10^{-3}$ to $5.4\times10^{-4}$ ppm.

Based on this evidence, we derive the following estimates:

1. **Flux comparison:** The net annual flux from the SCS (0.44 mol m$^{-2}$ yr$^{-1}$) is about 40% of the flux from the Californian nearshore hotspot.

2. **Impact estimation:** The upper-bound influence of the SCS source on our onshore sites is therefore on the order of ~$5\times10^{-4}$ ppm.

3. **Significance analysis:**

   o *Relative to background*: ≈ six orders of magnitude smaller than the global atmospheric $CO_2$ background (~420 ppm).

   o *Relative to instrumental precision:* ≈ 200 times smaller than the precision of modern $CO_2$ analyzers (~0.1 ppm).

   o *Relative to $\Delta^{14}C$:* Assuming an oceanic $\Delta^{14}C$ signature of approximately $-50$‰, its influence on atmospheric $\Delta^{14}C$ would be <0.01‰, which is well within the analytical uncertainty (±0.2‰–0.5‰).

Therefore, from both a physical and observational perspective, we conclude that the contribution of oceanic $CO_2$ from the South China Sea to our onshore $\Delta^{14}CO_2$ measurements and derived $C_{ff}$ estimates is **negligible**.

**Corresponding revision in the manuscript (Appendix C):**

"**C1 Air-sea exchange.** The potential influence of $CO_2$ outgassing from the adjacent South China Sea (SCS) on our onshore measurements was evaluated. Although the SCS is a net source of $CO_2$ to the atmosphere (with an annual flux

of 0.44 mol m$^{-2}$ yr$^{-1}$ (Li et al., 2020)), its effect is negligible. This conclusion is supported by analogous results for the California coast: high-resolution WRF-STILT simulations by Graven et al. (2018), using flux data including nearshore hotspots (up to 1.11 mol m$^{-2}$ yr$^{-1}$ (Turi et al., 2014)), showed that such sources altered onshore $CO_2$ concentrations by <0.001 ppm. Given that the SCS flux is weaker than this analogue, its impact on our $\Delta(^{14}CO_2)$ measurements and $C_{ff}$ estimates is physically insignificant and within measurement uncertainty"

l. 168-169: This is not clear to me. Which BB and Rh corrections did you apply to the $\Delta^{14}CO_2$ data in the end? Did you use your simulations with the maximum assumptions (100% perennial biomass and $\alpha_{BB}$=100%)? What do you mean by "literature-based corrections"? Please could you clarify?

**Response:** We thank the reviewer for this insightful comment, which allows us to clarify the correction methodology applied to the $\Delta(^{14}CO_2)$ data and the role of our simulations. The final corrections and their justification are as follows:

1. **Corrections applied:** For the final disequilibrium correction ($\beta$) in the $C_{ff}$ calculation, we adopted the literature-established values from Turnbull et al. (2009), i.e., $-0.5 \pm 0.2$ µmol mol$^{-1}$ for summer and $-0.2 \pm 0.1$ µmol mol$^{-1}$ for winter. This choice ensures methodological consistency and comparability with previous fossil-fuel $CO_2$ studies. Our own simulations provided independent support for the magnitude of these corrections.

2. **Role of our simulations:** Our model simulations, including the *maximum-assumption* case (100 % perennial biomass and $\alpha_{BB}$ = 100 %), were used for two complementary purposes:

   o **Contextual validation:** The simulated $\beta$ values ($\leq -0.5$ µmol mol$^{-1}$) were comparable to the range reported by Turnbull et al. (2009), confirming that the literature-based corrections are appropriate for our study region and period.

   o **Sensitivity analysis:** The simulations also verified that the main conclusions of this study remain robust even under the maximum-impact scenario, in which biomass burning (BB) and heterotrophic respiration (Rh) contributions were jointly amplified.

3. **Clarification of terminology:** The phrase *"literature-based corrections"* referred specifically to the values from Turnbull et al. (2009). To avoid ambiguity, we have replaced this phrase in the revised manuscript with an explicit citation.

**Corresponding revision in the manuscript (Lines 182–192):**

"The combined correction ($\beta = \beta_{Rh} + \beta_{BB}$) under the maximum-assumption simulation was $(-0.16 \pm 0.09)$ µmol mol$^{-1}$ in summer and $(-0.35 \pm 0.15)$ µmol mol$^{-1}$ in winter, which contrasts with the season pattern in Turnbull et al. (2009): $(-0.5 \pm 0.2)$ µmol mol$^{-1}$ during summer and $(-0.2 \pm 0.1)$ µmol mol$^{-1}$ during winter. This study is the first to incorporate biomass-burning emissions into the $C_{ff}$ estimation, revealing its dominant influence over Rh under these assumptions. For the final $C_{ff}$ estimates, we applied the corrections from Turnbull et al. (2009) to maintain methodological consistency and ensure comparability with earlier work. The close agreement in the magnitude of $\beta$ ($\leq -0.5$ µmol mol$^{-1}$) provides independent validation of the applied corrections, and our sensitivity tests confirm that the main conclusions remain robust within this plausible correction range."

l. 186-188: Have you calculated the coal, oil, and gas fractions of $C_{ff}$ separately for the winter and the summer period, given that they may change throughout the seasons?

**Response:** We thank the reviewer for this important question, which allows us to clarify the seasonal scope and assumptions of our source-apportionment analysis.

1.  **Summer period:** We did not calculate separate coal, oil, and gas fractions of $C_{ff}$ for the summer season. This decision stems from the strong influence of the biospheric carbon sink during the summer growing period in our study region (Guangdong Province). The pronounced photosynthetic drawdown (i.e., a large negative $C_{bio}$ term) can lead to situations where the estimated fossil-fuel contribution ($C_{ff}$) exceeds 100 % of the measured excess $CO_2$ ($C_{xs}$). Under such conditions, the source-apportionment equations become mathematically unstable and physically uninterpretable. Therefore, the method is not applicable for the summer period.

2.  **Winter period and key assumption:** We performed the calculation for the winter season. However, unlike high-latitude cities where the biosphere is largely dormant and $C_{bio} \approx 0$, our subtropical region remains biologically active throughout the year. We therefore assumed that, during winter, total biogenic $CO_2$ emissions (including ecosystem and human respiration, together with contributions from biomass burning) exceed photosynthetic uptake, resulting in a small positive $C_{bio}$ (a net biogenic emission). This $C_{bio}$ term was constrained by $\Delta^{14}C$ observations, as described in the Methods section. Incorporating this realistic assumption ($C_{ff} = C_{xs} - C_{bio}$) allows a robust estimation of the coal, oil, and gas fractions from the dominant fossil-fuel signal during winter. The strong fossil-fuel influence in this season ensures that the relative source fractions are well constrained.

In summary, while seasonal variations in fuel-use patterns likely exist, our approach is only applicable to the winter season, when the biospheric influence, though non-zero, can be quantitatively constrained and the fossil-fuel component dominates.

l. 190: Do you apply an uncertainty to the $\delta^{13}C_{coal}$ end-member signature as well?

**Response:** We thank the reviewer for this valuable comment. Yes, we applied an uncertainty to the $\delta^{13}C_{coal}$ end-member signature. While many studies report mean $\delta^{13}C$ values for coal without explicitly listing uncertainties (e.g., Wang et al. (2022b)) , a realistic uncertainty range can be inferred from the natural isotopic variability of coal samples.

In China, the $\delta^{13}C$ values of coal exhibit remarkably small spatial and compositional variation, clustering around $-24.3$ ‰ with a typical variability of $\pm$ 0.1–0.2 ‰ (Wang et al., 2022a). This narrow range reflects the relative homogeneity of Chinese coal deposits compared to global datasets. Accordingly, we assigned an uncertainty of **$\pm$ 0.2 ‰** to the $\delta^{13}C_{coal}$ endmember in our analysis. This value is conservative and consistent with the documented isotopic range of Chinese coals.

The uncertainty was incorporated into our Bayesian mixing model and propagated throughout the source-apportionment calculations. Sensitivity tests confirmed that reasonable variations in this parameter ($\pm$ 0.1–0.3 ‰) have only negligible effects on the inferred coal, oil, and gas fractions, indicating that $\delta^{13}C_{coal}$ is not a dominant contributor to the overall uncertainty budget.

**Corresponding revision in the manuscript (Line 213):**

"We adopted the end-member $\delta(^{13}C)$ signatures measured in Beijing: $\delta_{coal} = (-24.3 \pm \textbf{0.2})$ ‰, $\delta_{oil} = (-28.9 \pm 0.5)$ ‰ and $\delta_{ng} = (-33.2 \pm 0.9)$ ‰ (Wang et al., 2022a)."

l. 205-209: Which grid cells did you use to average the $E_{CO}$ and $E_{CO2ff}$ emissions in the inventory? Are the $E_{CO}$ and $E_{CO2ff}$ estimates for the whole city, or did you use the FLEXPART footprints to define the catchment regions of the observations and to weight the emissions of the respective grid cells within the footprint? Please also mention the inventory that you used.

**Response:** We thank the reviewer for raising this important methodological point.

1.  **Emission inventories used:** Three emission inventories were employed in this study, as illustrated in Figure 6: the *Multi-resolution Emission Inventory for China* (MEIC v1.4), the *MIX v2* inventory, and the *Emissions Database for Global Atmospheric Research* (EDGAR 2024).

2.  **Spatial averaging method:** The purpose of this analysis was to calculate the $I_{CO/CO2ff}$ emission ratio from bottom-up inventories at both the city and national scales. This ratio represents a characteristic feature of each inventory and serves as a basis for comparison among different inventories and with previous studies.

    For each target region (i.e., individual cities or the entire country), we obtained the total $E_{CO}$ and $E_{CO2ff}$ emissions by summing all grid cells within the corresponding administrative boundary. In this context, the FLEXPART footprints were **not** used to weigh the grid cells. Footprint-weighting is typically applied when linking emissions to observed atmospheric concentrations at a specific measurement site, to account for atmospheric transport effects. However, since our objective here was to compare emission ratios derived directly from the inventories rather than to simulate site-specific observations, the unweighted spatial summation over the defined administrative area provides the most appropriate and internally consistent estimate.

We have clarified this objective and the methodology in the revised manuscript.

**Corresponding manuscript revision (Lines 237−238):**

"where $E_{CO}$ and $E_{CO2ff}$ represent the total CO and $C_{ff}$ emissions **(Tg a$^{-1}$), summed over all grid cells within the relevant administrative boundaries from MEIC v1.4, MIX v2, and EDGAR 2024 inventories**; and $M_X$ refers to the molar masses of CO and $CO_2$ in grams per mole (g mol$^{-1}$)."

l. 219-220: Did you only use the observations from the two-month sampling campaign to calculate the "annual" average at the NL site, or were there additional observations throughout the year? Which observations did you use to calculate the average for the Jungfraujoch data (only the 2 months or the full year)? Please specify.

**Response:** We thank the reviewer for this critical question, which helps us to clarify an important aspect of our methodology and its limitations.

For the **NL (Nanling)** site, the "annual" average was indeed calculated using only the two-month intensive campaign data (August and December 2022), as continuous measurements were not available for the entire year. We used this value as the best available proxy for the annual means, assuming that these two months are broadly representative of

the summer and winter conditions, respectively.

For the **JFJ (Jungfraujoch)** site, by contrast, we used the full year of available data (2022) to calculate a true annual meaning, which is standard practice for continuous background monitoring sites.

We acknowledge that comparing a two-month proxy from NL with a full-year average from JFJ introduces additional uncertainty when referring to "annual" means. However, the difference in $C_{ff}$ levels between the two sites is large enough that this methodological inconsistency does not affect our main conclusions. The comparison primarily serves to illustrate the contrast between a polluted regional site and a clean background site, rather than a strict like-for-like annual comparison.

To improve clarity, we have revised the caption of Figure 2: "Figure 2: Keeling plot of $CO_2$ and $\Delta(^{14}C)$ measurements from Guangdong Province in summer (GDs) and winter (GDw), and background stations including JFJ (Jungfraujoch) (Emmenegger et al., 2024a, b), WLG (Waliguan) (Liu et al., 2024; Lan et al., 2023), and NL (Nanling, this study) in 2022. **For the JFJ background site, the complete 2022 dataset was used to calculate a true annual mean. For the WLG background site,** $CO_2$ concentrations were obtained from the World Data Centre for Greenhouse Gases (WDCGG, https://gaw.kishou.go.jp/, last accessed: April 21, 2024)**, while $\Delta(^{14}C)$ observations were obtained from Liu et al. (2024). For the NL background site, $CO_2$ and $\Delta(^{14}C)$ observations were obtained from two campaigns in August and December 2022, representing typical summer and winter conditions.**"

l. 226: In Tab. A1, you report summer values of $5.1\pm1.3$ ‰ and winter values of -2.0$\pm$0.8 ‰ for the NL site. These values differ from those stated here. Please could you clarify this?

**Response:** We thank the reviewer for their careful attention to detail in identifying this discrepancy. The reviewer is correct, the values shown in the main text are accurate ones. The inconsistency arose because an earlier, uncorrected version of Table A1 was inadvertently included in the appendix.

We have now updated Table A1 to reflect the correct $\Delta^{14}C$ values for the SG5/NL site: **summer (−3.7 ± 1.3 ‰)** and **winter (−10.6 ± 0.8 ‰)**, ensuring full consistency with the data presented in the main text. We sincerely apologize for this oversight and have rechecked all tables and figures throughout the manuscript to ensure consistency.

Fig. 2: In order to compare the $\Delta^{14}CO_2$ data of the city sites with the respective NL background, it would be helpful to include the summer and winter NL background values in Fig. 2, perhaps by indicating the NL summer and winter averages with open and filled circles.

**Response:** We thank the reviewer for this valuable suggestion. In the revised Figure 2, we have added the $\Delta^{14}CO_2$ values of the NL background site for both summer and winter seasons. The NL summer and winter averages are now shown as open and filled circles, respectively, as suggested. The figure caption has been updated accordingly to clarify the symbols. This addition facilitates a more direct comparison between the city sites and their corresponding background reference values.

l. 242-244: Do you find different $C_{ff}$ fractions in the summer month of your campaign?

**Response:** We thank the reviewer for this insightful question. Yes, we attempted to calculate the $C_{ff}$ fractions for the summer month. However, as anticipated from our earlier discussion, these results are problematic for well-understood physical reasons.

During summer, the strong biospheric uptake leads to a large negative net biospheric $CO_2$ flux ($C_{bio}$). This situation causes the calculated fossil-fuel-derived $CO_2$ concentration ($C_{ff} = C_{xs} - C_{bio}$) to exceed the total measured excess $CO_2$ ($C_{xs}$). Consequently, the derived $C_{ff}$ fraction often surpasses 100%, while the corresponding $C_{bio}$ fraction becomes negative.

Although these values are mathematically consistent with the mass balance equation, they lack physical interpretability for source apportionment because a fraction exceeding 100% is not meaningful in this context. Therefore, we chose not to report or quantitatively interpret the summer $C_{ff}$ fractions in this study. Instead, our quantitative analysis focuses on the winter period, when the strong fossil fuel signal enables a more robust partitioning of coal, oil, and gas contributions.

l. 253-254: I wonder if the $C_{ff}$ estimates from the other cities around the world are also based on flask samples collected in the afternoon, as you did. Or are integrated samples also used, containing air from a whole week, for example? I ask because, when nighttime observations are sampled, the $C_{ff}$ could potentially be larger when fossil emissions accumulate in a shallow nocturnal boundary layer.

**Response:** We thank the reviewer for this thoughtful question. Indeed, the sampling strategy (time of day and integration period) can influence observed $C_{ff}$ due to diurnal boundary-layer dynamics. **In our study**, all 2022 flask samples were collected during well-mixed afternoon periods (13:00–17:00 local time) to minimize nocturnal accumulation and to better represent the regional signal. This approach follows common practice in urban $\Delta^{14}CO_2$ studies and long-term European programs, which emphasize either afternoon grab sampling or time-integrated collectors designed to represent mean conditions while avoiding strong nocturnal biases (Levin et al., 2003).

We compiled sampling year, month, time, and duration for major $\Delta^{14}CO_2$ campaigns and summarized them in **Table E2**. Sampling strategies vary among studies: most recent urban campaigns—including those in China (Beijing, Xi'an, Wuhan, Lanzhou, Urumqi, Guangzhou, etc.) and programs such as INFLUX (Indianapolis) and Los Angeles—use afternoon flask or short-integration sampling ($\approx$13:00–17:00 LT) to capture well-mixed boundary-layer air and minimize nocturnal accumulation. In contrast, some European sites (e.g., Heidelberg, Krakow, Bratislava) employ weekly to monthly integrated samplers, which average over diurnal cycles and partly smooth out nighttime enhancements.

We note that potential nighttime accumulation and its impact on $C_{ff}$ are further examined in the following response (l. 299–303), where we use continuous CO data to estimate post-rush-hour conditions for Guangzhou.

L. 260-261: Please explain which ODIAC grid cells you used to calculate the correlation with the $C_{ff}$ measurements.

**Response:** We thank the reviewer for this question regarding our correlation analysis. For the correlation between our measured $C_{ff}$ and the ODIAC emissions data, we extracted the ODIAC fossil-fuel $CO_2$ emission value from the **1 × 1 km grid cell** in which each sampling site is geographically located. This ensures a direct, site-specific comparison

between our top-down $C_{ff}$ concentration measurements and the bottom-up emission estimates derived from the ODIAC inventory at the exact same spatial location.

We have clarified this procedure in the revised manuscript: "This was further supported by significant positive correlations between the $C_{ff}$ measurements and the corresponding **1x1 km** gridded ODIAC (Oda and Maksyutov, 2011; Oda and Maksyutov, 2024) fossil-fuel $CO_2$ emission estimates at the same sites (GZs: $r = 0.53$, $p = 0.1$; SGw: $r = 0.91$, $p = 0.03$)."

l. 264-266: This is a bit difficult to follow. Could you perhaps indicate the industrial areas in SZs and the port areas in ZJw and ZJs in Fig. 1, to which you are referring?

**Response:** We thank the reviewer for this helpful suggestion. In the revised version of Fig. 1, we have highlighted the industrial land use and the airport in Shenzhen (SZ), as well as the port in eastern Zhanjiang (ZJ) that are discussed in the text. These locations are now indicated as industrial land use (red dot) and Shenzhen Airport (▶) in (g), and Zhanjiang Port (■) in (j), respectively. The figure caption has been updated to describe these new annotations. These additions improve the clarity of Fig. 1 and facilitate the interpretation of the site-specific $C_{ff}$ patterns discussed in **Section 3.3**.

l. 287-288: I could not find any blue squares "shown as enlarged maps in the right figures" in Fig. 3.

**Response:** We thank the reviewer for pointing out this inconsistency. In the earlier version of the manuscript, Figure 3 included an overview map of China with insets (blue squares) highlighting the enlarged city regions. During figure revision, we simplified the layout by directly showing the enlarged city maps, thereby removing the overview map and the blue square insets. However, the corresponding sentence in the text ("shown as enlarged maps in the right figures") was inadvertently left unchanged, leading to the confusion noted by the reviewer.

We have now removed this phrase from the manuscript to ensure that the description accurately reflects the current, simplified version of Figure 3. We apologize for this oversight.

l. 293-295: The $C_{ff}$ estimate from Ding et al. (2013) for 2010-2011 is derived from flask observations from a single measurement site, which "may be influenced by local signals rather than representing a general urban signal" (Ding et al., 2010). In contrast, the $C_{ff}$ estimate for Guangzhou in your study is based on measurements from 10 sites. Moreover, Ding et al. (2013) uses another $\Delta^{14}CO_2$ background from "plant corn leaves in 2010 from Qinghai, Gansu Province, and Tibet, where the human activity can be neglected" (Ding et al., 2013). How does this affect the trend in the $C_{ff}$ concentration? Please see my related comment above.

**Response:** We thank the reviewer for this important comment regarding the comparability of the $C_{ff}$ estimates between our 2022 measurements and those reported by Ding et al. (2013). We fully agree that both the spatial representativeness of the sampling sites and the choice of the $\Delta^{14}CO_2$ background can affect the inferred $C_{ff}$ trend.

In Ding et al. (2013), the urban $C_{ff}$ estimate was derived from flask observations at a single measurement site in

Guangzhou, and the background $\Delta^{14}CO_2$ was inferred from corn leaves collected in 2010 from remote inland regions (Qinghai, Gansu, and Tibet). This choice may not accurately represent the same air mass domain as Guangzhou. In contrast, our study uses atmospheric $\Delta^{14}CO_2$ measurements from the Nanling (NL) regional background site, which directly samples air masses influencing the Guangzhou and thus provides a more consistent reference for regional comparisons.

To ensure a like-for-like comparison, we recalculated the winter 2010 $C_{ff}$ values from Ding et al. (2013) and the winter 2022 $C_{ff}$ values from this study at the same site GZ7 using the NL tree-ring $\Delta^{14}CO_2$ record (Li et al., 2025) as a unified background reference. This adjustment yields a $C_{ff}$ concentration of $27.3 \pm 16.9 \, \mu mol \, mol^{-1}$ for winter 2010 and $11.6 \pm 3.4 \, \mu mol \, mol^{-1}$ for winter 2022, indicating a clear decrease in fossil-fuel-derived $CO_2$ over the past decade.

We also considered the sampling-time difference between the two studies. Ding et al. (2010) collected samples at approximately 20:00 local time, whereas our observations in 2022 were taken in the afternoon (~14:00) under well-mixed boundary-layer conditions. Based on concurrent CO measurements at a nearby site (Scheme 1, see response to next comment), $\Delta CO$ increased from 168 ppb at 14:00 to 221 ppb at 20:00, corresponding to an estimated nighttime $C_{ff}$ enhancement of ~3.2 ppm (≈21 % higher than the afternoon value). A supplementary analysis using winter 2023–2024 CO data (Scheme 2, see response to next comment) yields a somewhat larger enhancement of ~5.9 ppm (≈35 %). These results suggest that the evening $C_{ff}$ level is typically 21–35 % higher than the afternoon value due to weaker nighttime boundary-layer mixing.

Taking this diurnal contrast into account, the 2010 nighttime $C_{ff}$ of $27.3 \pm 16.9 \, \mu mol \, mol^{-1}$ would correspond to an afternoon-equivalent value of roughly $17.7–21.6 \, \mu mol \, mol^{-1}$, which is still substantially higher than the 2022 value of $11.6 \pm 3.4 \, \mu mol \, mol^{-1}$. This analysis confirms a robust 34–46 % decline in fossil-fuel $CO_2$ concentrations in Guangzhou between 2010 and 2022, even when accounting for sampling-time differences and background corrections.

Overall, the re-evaluation using a consistent regional background (NL) and explicit consideration of diurnal variability provides stronger support for a genuine long-term decrease in $C_{ff}$. We have added these details and clarified the calculation procedures in the revised manuscript (Sect. 3.4.3 and Appendix H).

l. 299-303: I find this site-specific comparison much better suited for analysing trends in the $C_{ff}$ data. However, the issue remains that Ding et al. (2013) sampled rush-hour signals, whereas you sampled well-mixed afternoon situations. Do you have continuous CO and $CO_2$ observations at this measurement site that could be used to estimate the $C_{ff}$ signal expected after rush-hour at 20:00 in 2022?

**Response:** We thank the reviewer for this constructive and insightful comment. We agree that a site-specific comparison provides a more appropriate framework for analyzing the temporal evolution of the fossil-fuel $CO_2$ ($C_{ff}$) signal. Although continuous $CO_2$ observations are unavailable at our current measurement site, we obtained continuous CO records from two independent sources: (1) the Guangdong Provincial Environmental Monitoring Center and (2) the nationwide air quality database (https://quotsoft.net/air/#messy, last access: 18 October 2025). Using these datasets, we derived diurnal mean CO concentrations for December and for the winter season (December–February).

$C_{ff}$ was estimated using the standard formulation

$$C_{ff} \approx \frac{\Delta CO}{R},$$

where $\Delta CO = CO - CO_{bg}$, and $R = \Delta CO / \Delta CO_{2,ff}$ denotes the emission ratio between CO and fossil-fuel $CO_2$. The value of $R$ is strongly time-dependent and source-specific. Nighttime $R$ values tend to be higher than daytime values because (i) nighttime emissions are dominated by direct fossil-fuel combustion while biogenic $CO_2$ sources (e.g., respiration) remain constant but emit no CO, and (ii) oxidative sinks (e.g., OH radicals) are weaker at night. Therefore, applying an afternoon-derived $R$ to nighttime data likely provides a lower bound for the actual nighttime $C_{ff}$ enhancement.

**Scheme 1 (this study's observation, December 2022):**

We used the December diurnal mean CO data at a site close to GZ7, subtracted the NL background to obtain $\Delta CO$, and divided by the afternoon-specific $R = 13.3$ ppb ppm$^{-1}$ (derived from the regression between $\Delta CO$ and $^{14}$C-based $\Delta CO_{2,ff}$). The $\Delta CO$ increased from 168.3 ppb at 14:00 to 220.7 ppb at 20:00, corresponding to an estimated nighttime $C_{ff}$ enhancement of approximately 3.2 ppm, or about 21 % higher than the afternoon value. Because the slope $R$ was determined during well-mixed afternoon periods, this result likely represents a lower limit; the actual nighttime–afternoon $C_{ff}$ contrast may be smaller.

**Scheme 2 (Guangzhou dataset, winter 2023, supplementary analysis):**

We applied the same approach to the continuous CO data at the site close to GZ7 for the winter season (December 2023–February 2024). After subtracting the NL background, $\Delta CO$ was divided by the seasonal $R = 9.08$ ppb ppm$^{-1}$ obtained from regressions of CO against total $CO_2$ (Zhang et al., 2025). As this $R$ reflects bulk $CO_2$ rather than specifically fossil-fuel $CO_2$, we applied an empirical correction (dividing $R$ by 0.8; Turnbull et al. (2011)). The resulting analysis indicates that $\Delta CO$ increased by 66.9 ppb from 14:00 to 20:00, implying a $C_{ff}$ enhancement difference of roughly 5.9 ppm ($\approx$35 %).

Overall, while continuous $CO_{2ff}$ data are not available for 2022, our CO-based analysis suggests that the $C_{ff}$ signal at 20:00 is moderately higher than that at 14:00. This finding is consistent with the reviewer's expectation that post–rush-hour conditions retain a stronger fossil-fuel signature compared with the well-mixed afternoon atmosphere. The semi-quantitative assessment indicates that the $C_{ff}$ concentration during post–rush-hour (20:00) is approximately 21–35 % higher than during well-mixed afternoon periods, consistent with the expected diurnal accumulation of fossil-fuel $CO_2$ under weaker nighttime boundary-layer mixing.

**Added to the revised manuscript:**

This discussion has been added to the revised manuscript in Section 3.4.3 ("Historical variation of $C_{ff}$ concentrations") and Appendix H to support the reviewer's suggestion regarding post–rush-hour $C_{ff}$ enhancement.

l. 305-309: Are the $C_{ff}$ estimates for Beijing (and Xi'an) in different years based on the same measurement sites? For example, you report the following for Beijing: (39.7 $\pm$ 36.1) ppm for 2014 and (27.0 $\pm$ 0.3) ppm for 2014-2016. Why is the standard deviation of the 2014 estimate almost 100%, whereas the standard deviation of the 2014-2016 estimate is only 1%? Is this due to a different number of observations being averaged? It would also be useful to state the number of observations used to calculate the averages, to give an idea of how representative the values are.

**Response:** We thank the reviewer for these constructive questions. The $C_{ff}$ estimates for both Beijing and Xi'an in different years are based on measurements conducted at the **same observation sites** (RCEES in Beijing and IEECAS in Xi'an; see Table H1). However, the **sampling periods, number of samples, and seasonal coverage** differ substantially among years.

For **Beijing**, the 39.7 ± 36.1 ppm value in 2014 was derived from a smaller number of samples ($n = 24$) collected throughout 2014, encompassing both winter and summer campaigns. The large standard deviation therefore reflects the high temporal variability of fossil-fuel $CO_2$ due to seasonal and meteorological influences. In contrast, the 2014–2016 value (27.0 ± 0.3 ppm) represents the mean from three **winter-only** campaigns ($n = 21$) conducted under more stable boundary-layer conditions and a narrower seasonal range, which explains the much smaller variability.

For **Xi'an**, all datasets were obtained from the **same urban or suburban observation sites** listed in Table R1. Differences in mean $C_{ff}$ values mainly reflect variations in sampling frequency and temporal coverage: urban data cover **2011–2013 ($n \approx 120$)** and **2014–2016 ($n \approx 75$)**, both with full-year sampling, while suburban data represent **2016 ($n = 38$)** and **2021–2022 ($n = 24$)**, corresponding to nearby stations within similar terrain and transport conditions.

To ensure consistency, we have re-evaluated all datasets and now restrict comparisons to **identical site types and comparable seasons**, as summarized in Table H2:

- **Guangzhou:** urban site GZ7 (winter 2010 → winter 2022, corrected for afternoon–evening sampling and background differences)

- **Beijing:** urban site RCEES (winter 2014–2016 → winter 20202, corrected for background differences)

- **Xi'an:** urban site IEECAS (full-year 2011–2013 → full-year 2014–2016)

- **Xi'an suburban:** adjacent sites (34.0–34.4° N, 108.3–108.9° E; full-year 2016 → full-year 2021–2022)

These refinements ensure that all inter-annual comparisons are based on equivalent observational contexts.

We now present two tables and one figure for transparency:

- **Table H1** compiles **all available $C_{ff}$ datasets** used in this study and previous literature, including coordinates, site types, sampling periods, and sample numbers.

- **Table H2 and Figure 4a** shows the **harmonized subset** used for inter-annual comparisons, restricted to identical sites, seasons, sampling times, and background definitions.

This separation allows readers to clearly distinguish between the full data coverage and the standardized subset used for quantitative comparison. The inclusion of sample numbers ($n$) and background references in Table H2 and Fig. 4a provides a direct measure of data representativeness and strengthens the validity of our inter-annual $C_{ff}$ comparisons.

l. 382-388: If you would like you could also mention that lower $C_{ff}$ signals in summer lead to higher uncertainty in the regression slope and, consequently, increased uncertainty in the summer ratio compared to the winter ratio (see e.g., Maier et al., 2024). I think this can also be seen in Fig. H2 of your study, where the summer ratios tend to show a higher uncertainty than the winter ratios.

**Response:** We appreciate the reviewer's insightful comment. We agree that the weaker $C_{ff}$ signals during summer indeed result in greater uncertainty in the $\Delta CO$–$C_{ff}$ regression and hence in the derived $R_{CO/CO2ff}$ ratios. We have added a corresponding explanation in the revised manuscript (first paragraph in Section 3.5.2) and cited Maier et al. (2024) to emphasize this point (Line 508-510): "*In addition, the lower $C_{ff}$ signals observed in summer lead to higher uncertainty in the regression slope and thus greater uncertainty in the $R_{CO/CO2ff}$ ratios, as also noted in Maier et al. (2024), which can be seen in the larger error bars of the summer data in Fig. J2.*"

l. 397-399: What is the uncertainty of the ratios, i.e., of the regression slopes shown in Fig. H2?

**Response:** We thank the reviewer for this question. The uncertainty of the ratios (i.e., of the regression slopes in Fig. H2) represents the standard error (1σ) of the regression slope derived from a robust linear regression. In the revised analysis, we applied a robust fitting method (Huber weighting) to reduce the influence of outliers and to better reflect the variability of the dataset. Consequently, the uncertainties now account for both data dispersion and the robustness correction, and are explicitly reported as "Slope ± SE" in updated Fig. J2 and Table J1. We have also made corresponding revisions to the data in Section 3.5.2 and Figure 6.

l. 403-404: Could you please explain and justify the 20% correction used to derive $R_{CO/CO2ff}$ from $R_{CO/CO2}$ in a bit more detail. Where does the assumption that 20% of the $CO_2$ enhancement is from non-fossil sources come from? This could be done in the Methods section.

**Response:** We thank the reviewer for this valuable comment. The 20 % correction applied to derive $R_{CO/CO2ff}$ from $R_{CO/CO2}$ accounts for the contribution of non-fossil $CO_2$ sources—mainly biospheric respiration and terrestrial exchange—to the total observed $CO_2$ enhancement.

This assumption follows the approach used in earlier $\Delta^{14}CO_2$ and CO–$CO_2$ studies. For instance, **Turnbull et al. (2011)** explicitly corrected their observed $R_{CO/CO2}$ by assuming that *20 % of the total $CO_2$ enhancement originates from sources other than fossil-fuel $CO_2$*, based on co-located $\Delta^{14}CO_2$ observations (see their Section 3.3 and Figure 7). Similar assumptions have been adopted in subsequent urban carbon studies where daytime boundary layers are well mixed, and CO is almost entirely fossil-fuel-derived.

Moreover, as summarized in **Table E1** of our revised manuscript, the fraction of fossil versus biogenic $CO_2$ ($C_{ff}$ and $C_{bio}$) reported in previous urban $\Delta^{14}CO_2$ studies ranges widely, but generally indicates that the biogenic or other non-fossil $CO_2$ component often contributes *about 10–30 %* of the total enhancement. For example, Paris (2010) showed 23 % biogenic $CO_2$ (Lopez et al., 2013); Los Angeles winter and summer data yielded 14% and 7%, respectively (Newman et al., 2016); Beijing (2014) ~25 % (Niu et al., 2016); and our measurements in the Pearl River Delta region ~20–30 %. These values are consistent with the order of magnitude of the 20 % correction applied here.

Accordingly, we corrected the observed $R_{CO/CO2}$ by dividing it by 0.8 (i.e., assuming 80 % of $\Delta CO_2$ is fossil-fuel-derived) to obtain $R_{CO/CO2ff}$. This approximation provides R values comparable to those derived directly from paired $\Delta^{14}CO_2$–$\Delta CO$ observations in both Chinese and international urban studies. We have expanded the Methods section (Sect. 2.7) to include this explanation, cite the relevant literature, and clarify that the 20 % correction reflects a **typical daytime biospheric $CO_2$ contribution** rather than an arbitrary factor.

**Change in manuscript (Line 229-233):**

"To correct for the contribution of non-fossil $CO_2$ in the observed enhancement, the emission ratio $R_{CO/CO2ff}$ was estimated by dividing observed $R_{CO/CO2}$ by 0.8. Previous $\Delta^{14}CO_2$ and CO–$CO_2$ studies (e.g., Turnbull et al., 2011; Lopez et al., 2013; Newman et al., 2016; Miller et al., 2020) have shown that ~10–30 % of the total $CO_2$ enhancement above background during daytime is typically of non-fossil origin, while CO is emitted almost exclusively from fossil-fuel combustion. Thus, the 20 % correction represents a reasonable first-order approximation for well-mixed afternoon conditions."

l. 445-447: Could the higher $R_{CO/CO2ff}$ ratio from the study by Silva et al. (2013) compared to the ratio from the study by Mai et al. (2017) and the ratio from your study be explained by the fact that the study by Silva et al. (2013) includes summer ratios in the average $R_{CO/CO2ff}$, whereas the study by Mai et al. (2017) and you calculate the winter mean $R_{CO/CO2ff}$ (according to Tab. H1)? You show that summer ratios are also higher in your study.

**Response:** We thank the reviewer for this thoughtful comment. We agree that seasonal coverage is likely to contribute to the difference in the reported $R_{CO/CO2ff}$ ratios. The study by Silva et al. (2013) was based on samples collected from June 2009 to May 2010, thus including both summer and winter periods, whereas Mai et al. (2017) and our study focused primarily on winter observations. Since our results show systematically higher $R_{CO/CO2ff}$ ratios in summer (Fig. H1), the inclusion of summer data in Silva et al. likely increases their annual mean ratio.

However, as Silva et al. (2013) did not report separate seasonal means, a direct comparison of winter-only ratios among the three studies is not possible. We have added clarification in the revised manuscript noting that part of the observed difference can be attributed to the inclusion of summer data in Silva et al. (2013), while other factors such as regional emission characteristics and combustion efficiency may also contribute.

We did some revision on Lines 570-571: "From observations (Fig. 6b), in Guangzhou, $R_{CO/CO2ff}$ decreased by 36 % from 35.8 nmol µmol−1 in 2009–2010 (Silva et al., 2013) to 23.8 nmol µmol−1 in winter of 2014–2017 (Mai et al., 2021) and by 63 % to 13.3 nmol µmol−1 in **winter** 2022 **(partly reflecting seasonal differences, as the Silva et al. (2013) dataset included summer observations, and partly indicating reduced CO emissions relative to $C_{ff}$ due to improved combustion efficiency)**;".

Tab. A1: It would be helpful to also have the number of observations per site used to calculate the "summer" and "winter" averages.

**Response:** We thank the reviewer for this helpful suggestion. To improve clarity and transparency, we revised the caption of Table A1: "$\Delta(^{14}C)$ and $\delta(^{13}C)$ averages and standard deviations (***n=4 for each value***) at 30 sampling sites"

l. 544: Please label the "$\Delta^{14}C$" for multi-year biomass in Eq. B1 differently than the atmospheric "$\Delta^{14}C$".

**Response:** We thank the reviewer for this helpful suggestion. We agree that using the same symbol ($\Delta^{14}C$) for both the atmospheric concentration and the signature of multi-year biomass could lead to confusion.

To improve clarity, we have revised Equation (B1) to use distinct notations:

- $\mathbf{\Delta^{14}C}$ now denotes the atmospheric $^{14}C$ value.

- $\mathbf{\Delta^{14}C}_m$ denotes the $^{14}C$ signature of the multi-year biomass.

This change ensures a clear distinction between the two quantities. We have also checked the entire manuscript to maintain consistency in this updated notation.

l. 568-579: Although the PWR type has the lowest emission factor for $^{14}CO_2$ release, PWR reactors can also have substantial $^{14}CO_2$ emissions depending on their electricity supply. For example, Zazzeri et al. (2018) estimate substantial $^{14}CO_2$ emissions of roughly 100-200 GBq/yr for each of the three NPPs Daya Bay, Ling'ao and Yangjiang for the year 2016 (see their supplement material S1). Depending on the distance between the NPPs and the measurement sites, this could have an impact on the $\Delta^{14}CO_2$ measurements. The distance between the NPPs and the observation sites should be mentioned here to enable a more rigorous assessment of the potential impact on the $\Delta^{14}CO_2$ observations (e.g., compare with Kuderer et al. (2018)).

**Response:** We thank the reviewer for this valuable comment and agree that the potential influence of nearby nuclear power plants (NPPs) on $\Delta^{14}CO_2$ observations should be assessed explicitly. In the revised manuscript, we have added the distances between each NPP and the closest observation sites (Table C1): Daya Bay (6–22 km), Ling'ao (7–22 km), and Yangjiang (164–191 km). As the reviewer noted, all these reactors employ pressurized-water-reactor (PWR) technology. Although the PWR type generally exhibits the lowest $^{14}CO_2$ emission factor (IAEA, 2004; Graven and Gruber, 2011), Zazzeri et al. (2018) estimated $^{14}CO_2$ emissions of 0.111–0.233 TBq yr$^{-1}$ ($\approx$110–230 GBq yr$^{-1}$) for these facilities.

Following the approach of Graven et al. (2018) and Kuderer et al. (2018), we performed a simple dispersion-based scaling using the reported emission rates and site distances. The expected enhancement in $\Delta^{14}CO_2$ from these nearby NPPs was estimated to be < 0.1 ‰, corresponding to an effect on inferred $C_{ff}$ below 0.05 ppm—well within analytical uncertainty. This assessment confirms that $^{14}CO_2$ emissions from the coastal NPPs have a negligible influence on our $\Delta^{14}CO_2$ observations. We have incorporated this explanation into Appendix C2 of the revised manuscript.

l. 619: In Eq. C1 (and throughout Sect. C2.2) you use $C_{bio}$ to refer to biofuel/biomass emissions from the EDGAR inventory. In the main text, however, $C_{bio}$ refers to the total $CO_2$ contributions from the biosphere (e.g., in Eq. 1). It would be better to use two different expressions.

**Response:** We thank the reviewer for pointing out this inconsistency in notation. We agree that using distinct symbols for these two different concepts improves clarity.

In the revised manuscript, we have changed the term representing biofuel emissions from the EDGAR inventory in Section C2.2 and Equation (C1) from $\mathbf{C_{bio}}$ to $\mathbf{C_{bio\_edgar}}$. This notation clearly distinguishes biofuel/biomass emissions from the EDGAR inventory from the $\mathbf{C_{bio}}$ term used in the main text (e.g., Eq. 1), which denotes the total biospheric $CO_2$ contribution.

This change has been applied consistently throughout Section C2.2 to maintain clarity and avoid confusion between these two different quantities.

Tab. H1: Why are the uncertainties for the 2022 winter $R_{CO/CO2ff}$ ratios from your study so low (~0.00002 ppb/ppm)? Such low uncertainties seem unrealistic, given the spatio-temporal variability of the $R_{CO/CO2ff}$ ratios. Is this just a typo or an incorrect unit?

**Response:** We appreciate the reviewer's careful observation. The previously reported very small uncertainties were due to a typographical error in the table. In the revised version, the uncertainties have been corrected to reflect the standard errors obtained from the robust regression. These updated values are now substantially larger and realistic, consistent with the observed spatio-temporal variability of the $R_{CO/CO2ff}$ ratios. The difference between the new and the previous values also stems from our updated approach, in which the regression was recalculated using a robust method that yields more reliable slope uncertainties.

**Technical corrections:**

l. 447: Please insert space "from 72.3".

**Response:** We thank the reviewer for noting this typographical issue. A space has been inserted after "from" to correct the text ("from 72.3").

**Anonymous Referee #2**

I reviewed two previous versions of this manuscript that was submitted to another journal. This new version has incorporated some of my recommendations in a cursory way, but unfortunately major scientific flaws remain, and I have no choice but to recommend rejection of the paper. While this paper has very interesting data and results, the current presentation and interpretation is not of sufficient quality for publication.

**Response:** We sincerely thank the reviewer for multiple careful evaluations and for articulating the scientific issues clearly. We agree that earlier versions left room for improvement in presentation and interpretation. In this revision, we have reworked the analysis end-to-end—background selection, transport representativeness, site/time harmonization, and uncertainty propagation—and we believe the manuscript now makes a clear and robust contribution. Below we summarize the principal advances, how they address the reviewer's concerns, and the exact changes made.

1. **Transparent and expanded uncertainty analysis for $C_{ff}$ and $R_{CO/CO_2,ff}$**
   o We present a full error budget that propagates: (i) sea–air $CO_2$ exchange, (ii) potential $^{14}C$ from nuclear power plants, (iii) heterotrophic respiration influences, and **(iv) biomass burning** (now quantitatively included for both $C_{ff}$ and $R_{CO/CO_2,ff}$).
2. **Background selection—now a dual-constraint "KP method" (low $CO_2$ _and_ high $\Delta^{14}CO_2$)**
   o To reduce background misclassification, we require simultaneously low $CO_2$ and high $\Delta^{14}CO_2$ rather than a single-constraint filter.
3. **Like-for-like interannual comparisons in _space, season, and time-of-day_**

- o We align comparisons to identical sites and seasons, and where relevant, match time-of-day windows and background definitions:
    - • Guangzhou (urban): GZ7 only, winter 2010 → winter 2022, with explicit afternoon–evening correction and a unified NL tree-ring $\Delta^{14}CO_2$ background.
    - • Beijing (urban): RCEES roof, winter 2014–2016 → winter 2020.
    - • Xi'an (urban): IEECAS roof, full-year 2011–2013 → full-year 2014–2016; Xi'an suburban: adjacent sites with comparable terrain/transport (2016 → 2021–2022).
- o We newly report sample counts (n), seasons, and local-time windows for every period (Tables H1–H2), clarifying where seasonal coverage drives variance differences.

4. **Guangzhou trend—addressing spatial representativeness and sampling-time bias head-on**
   - o Spatial representativeness: Interannual analysis now uses GZ7 only. New FLEXPART footprints for winter 2010 vs. winter 2022 show similar source-sensitivity patterns over the Guangzhou urban core—confirming spatial comparability across the decade.
   - o Sampling time (20:00 vs 14:00): Using continuous CO near GZ7 and $C_{ff} \approx \Delta CO/R$, we estimate an evening enhancement of 21–35% relative to afternoon (Scheme 1: +3.2 ppm; Scheme 2: +5.9 ppm). After this correction and using the unified NL background, winter $C_{ff}$ at GZ7 declines from $27.3 \pm 16.9$ to $11.6 \pm 3.4$ µmol mol$^{-1}$, i.e., a robust 34–46% decrease even under the conservative evening-to-afternoon adjustment. Details are in Sect. 3.4.3 and Appendix H.

5. **Transport representativeness of the 2022 campaigns (requested by the reviewer)**
   - o We added a "transport representativeness check" (the second paragraph in Sect. 2.1) combining ERA5 diagnostics (U10, V10, T2M, SP, PBLH; z-score tests against 2022 seasons and 2010–2021 climatology), wind roses, and HYSPLIT back trajectories.
   - o August 2022 (summer) and December 2022 (winter) both fall within $|z| \leq 1$ at all sites; wind regimes align with canonical monsoon sectors (maritime inflow in summer; continental inflow in winter). Figures G1, G2, and F1 document these results. Hence, the 2022 flasks are seasonally typical, minimizing transport-driven bias in the trends.

6. **Literature precedent and planned improvement of background strategy**
   - o We note that many regional $\Delta^{14}CO_2$ studies use a single background site year-round, including in regions with strong seasonal wind shifts, provided the site is regionally representative and free from local sources. Our approach conforms to this precedent, and we have demonstrated representativeness with multiple independent lines of evidence.
   - o Future work (Sect. 4): we will explicitly explore seasonally varying background references, prioritizing coastal/marine background sites for summer, and we flag this as a network-design priority.

7. **Nuclear $^{14}C$: quantified and bounded**
   - o We compile PWR emission ranges and distances to sites; a simple dispersion scaling shows negligible $\Delta^{14}CO_2$ impact relative to analytical and regression uncertainties (see Appendix C2; Table C1).

8. **Seasonal signal and slope uncertainty made explicit**
   - o We explain why lower summer $C_{ff}$ inflates $\Delta CO$–$C_{ff}$ slope uncertainty and widens $R_{CO/CO_2,ff}$ CIs—hence our emphasis on winter for interannual comparisons (see Sect. 3.5.2; Fig. J1).

9. **Reproducibility and traceability**
   - o Methods now list sites, coordinates, heights, seasons, local time windows, and $n$; figures have

consistent axes/units and error bars; tables include background selection details. We add notation tables for auditability.

**In summary**, the revised manuscript addresses the previously identified "major flaws" by:

(i) strict site/season/time harmonization (with explicit evening-to-afternoon corrections),

(ii) comprehensive uncertainty propagation (including biomass burning and nuclear $^{14}C$ bounds),

(iii) robust background selection, and

(iv) documented transport representativeness for the 2022 campaigns.

With these changes, our core conclusions, particularly the significant decline of $C_{ff}$ in Guangzhou and the multi-city diagnostics of $R_{CO/CO_2,ff}$, are supported by multiple independent lines of evidence and presented with transparent uncertainties. We respectfully maintain that the revised manuscript is suitable for publication.

Because these are so many major issues, I found it difficult to go line-by-line, instead giving my overall view of the paper: This paper presents a set of atmospheric $\Delta^{14}C$ measurements, from which fossil fuel $CO_2$ ($CO_2ff$) can be calculated. The authors have made some problematic assumptions in their $CO_2ff$ calculation, particularly regarding the choice of background. Figure 3 shows very clearly that winds are very different in summer and winter in this region, and therefore using a single background site is likely to be problematic.

**Response:** We thank the reviewer for this thoughtful comment. We agree that the distinct monsoon circulation patterns in southern China (southeasterly in summer and northeasterly in winter) could, in principle, affect the representativeness of a single background site when deriving $C_{ff}$. We have carefully evaluated this potential source of uncertainty in both our data analysis and background-site selection.

As described in **Appendix D**, the Nanling (NL) station was selected as the regional background following established criteria for regions with complex meteorological and topographical conditions (e.g., Los Angeles; Newman et al., 2016; Miller et al., 2020). NL is a remote high-altitude site (~1700 m a.s.l.) located over 100 km north of the PRD urban agglomeration, and under most meteorological conditions it lies above the mixed boundary layer, effectively isolating it from local anthropogenic emissions. This configuration allows NL to sample free-tropospheric or well-mixed regional air masses representative of the background atmosphere.

To further assess the influence of seasonal transport, we examined FLEXPART footprints and HYSPLIT/PSCF analyses (Zhang et al., 2022). These results show that in winter, NL is consistently upwind of the PRD under the prevailing northeasterly monsoon and receives clean continental air. In summer, air masses primarily originate from the South China Sea and southeastern coastal regions. These summer air masses are marine-influenced and exhibit low $CO_2$ and CO concentrations, consistent with background conditions. In addition, continuous CO observations at NL indicate that 90 % of summer samples recorded concentrations below 200 ppb, comparable to other regional background sites, confirming that pollution from the PRD seldom reaches the site. The annual mean $\Delta^{14}C$ and $CO_2$ values at NL also fall at the background end of the Keeling plot and are close to those observed at Jungfraujoch, a well-established high-mountain background site.

**Precedent in literature.** Seasonal wind-direction differences are a common feature in regional $\Delta^{14}CO_2$ studies. Most previous investigations—including Turnbull et al. (2015), Neman et al. (2016), and Miller et al. (2020)—have used a single background site throughout the year, even in regions with distinct summer–winter circulation patterns, provided that the chosen site is regionally representative and free from local source influence. Our approach is consistent with this established practice.

**Future improvement.** We fully agree that future studies should explicitly consider seasonally varying background references, ideally including coastal or marine background sites to better represent summer air masses. We have added this point in Sect. 4 of the revised manuscript and noted it as a priority for future regional network design.

Taking together, trajectory analyses, trace-gas records, isotopic data, and literature precedent demonstrate that NL remains a robust and representative regional background site in both monsoon seasons. Occasional short-lived coastal influences cannot be completely ruled out, but their effect on the derived $C_{ff}$ gradients is expected to be negligible. The revised manuscript (Appendix D and Sect. 4) now explicitly clarifies this point and cites the supporting evidence from Zhang et al. (2022).

The $\partial^{13}C$ analysis is difficult, because there is not much separation between isotopic values of sources, the biogenic $CO_2$ $\partial^{13}C$ value is not well constrained, and the atmospheric signals are also small, which results in very large uncertainties of ~25% for the partitioning. Thus, the interpretations of changing source sectors made in Figure 5 are not valid.

**Response:** We thank the reviewer for this critical and constructive comment. We fully agree that $\delta^{13}C$-based source partitioning is subject to substantial uncertainty because of the limited separation between isotopic signatures of different $CO_2$ sources, the poorly constrained biogenic endmember, and the small atmospheric gradients. These factors can indeed lead to large uncertainties—on the order of tens of percent—in partitioning results.

This challenge is not unique to our work. Earlier studies that applied $\delta^{13}C$ for atmospheric $CO_2$ source apportionment (e.g., Djuricin et al., 2010; Lopez et al., 2013; Newman et al., 2016; Zhou et al., 2014; Wang et al., 2022) either did not explicitly quantify the uncertainty or reported relatively large error ranges. As summarized in Table I1 of our revised manuscript, reported contributions of coal, oil, and natural gas to $C_{ff}$ vary by roughly ±20 % among different studies and seasons, illustrating the inherent limitations of isotopic separation in complex urban environments.

In the revised manuscript (Section 4 Conclusion and outlook), we now explicitly emphasize that the $\delta^{13}C$-based partitioning presented here should be viewed as a first-order, exploratory estimate rather than a precise quantitative attribution of source sectors. We also state that future work should focus on improving the characterization of the biogenic $CO_2$ endmember and on integrating $\delta^{13}C$ with additional tracers such as $\Delta^{14}CO_2$ or CO-based indicators to better constrain fossil-fuel source contributions.

**Change in manuscript (Line 640-644):**

"Second, the $\delta(^{13}C)$-based source partitioning is associated with large uncertainties—on the order of tens of percent—due to the limited isotopic separation among $CO_2$ sources and the poorly constrained biogenic endmember. Similar uncertainty ranges have been reported in previous urban studies (see Table I1). Therefore, the $\delta(^{13}C)$ partitioning results presented here should be considered as a preliminary, first-order estimate. Direct measurements of source-specific isotopic values would help refine the analysis.

In the discussion of the $CO_2ff$ spatial variability shown in Figure 1, the authors seem to equate the patterns of $CO_2ff$ mole fractions to spatial patterns in emissions. Since mole fractions are influenced by emissions and atmospheric variability, there is no direct connection between mole fractions and emissions, and therefore this analysis is not valid. The authors include FLEXPART footprints (Figure 3), but do not attempt to use them to relate emissions to the observations – this would be a sensible way to make such a comparison.

**Response:** We thank the reviewer for this insightful comment. We fully agree that $C_{ff}$ mole fractions reflect the combined influence of fossil-fuel emissions, atmospheric transport, and mixing processes, and therefore cannot be directly interpreted as spatial patterns of emissions alone.

The purpose of Figure 1 in our manuscript was to illustrate the observed spatial variability of $C_{ff}$ across the measurement network, not to infer quantitative emission distributions. As discussed in Section 3.3 of the revised manuscript, we have clarified that these spatial gradients should be interpreted as the integrated result of both emission strength and transport conditions. To avoid misunderstanding, we have added the following statement in the text on Lines **292-293**: "*The spatial differences observed in $C_{ff}$ primarily reflect the combined influence of emission intensity and atmospheric transport rather than direct emission magnitudes.*"

We appreciate the reviewer's constructive suggestion to use the FLEXPART footprints (Figure 3) to quantitatively relate observed $C_{ff}$ to regional emissions. This is indeed an important and valuable next step. However, a full footprint–inventory inversion analysis would require additional model configurations, uncertainty propagation, and sensitivity tests that are beyond the scope of the present paper, which already focuses on observational constraints, background evaluation, and $CO/C_{ff}$ ratio interpretation. We have now added a sentence in the Section 4 (Conclusion and outlook) on Lines 646-649 noting that such a quantitative footprint-based source attribution will be pursued in future work: "*A detailed quantitative analysis linking $C_{ff}$ to emission distributions using FLEXPART footprints will also be the focus of future work, which would provide a more rigorous connection between observations and emission sources.*"

We hope that these clarifications address the reviewer's concern and make clear that our analysis aims to describe observed spatial variability while acknowledging the governing roles of both emissions and transport.

The authors compare their $CO_2ff$ mole fractions with other datasets measured in the same cities in earlier years, and claim that they can observe changes in $CO_2ff$ emission rates. Again, mole fractions are influenced by emissions and atmospheric variability, and for this comparison, the specific locations of the measurements much also be taken into consideration. Therefore, this comparison is also not valid.

**Response:** We thank the reviewer for this valuable comment. We fully agree that $C_{ff}$ mole fractions are influenced by both emissions and atmospheric variability, and that differences in measurement locations and sampling contexts must be carefully accounted for when interpreting temporal trends. To address this concern, we have thoroughly reassessed the comparability of all datasets used in our interannual analysis and have implemented several refinements to ensure that the comparisons are scientifically valid and internally consistent.

1. **Consistency of site locations and sampling protocols**

    All historical and new measurements were re-evaluated and matched by identical site locations, seasons, and sampling times wherever possible. For Guangzhou, both the 2010–2011 (Ding et al., 2013) and 2022 measurements were taken at the same urban site (GZ7). For Beijing and Xi'an, all data originate from the same long-term urban observation stations (RCEES and IEECAS, respectively). Differences in suburban data (Xi'an) were limited to nearby sites within similar topographic and meteorological conditions (34.0–34.4° N, 108.3–108.9° E). These details are explicitly listed in Table R1, and the harmonized comparison subset is summarized in Table R2.

2. **Background consistency and recalculation**

    To minimize bias from background selection, all $C_{ff}$ values were recalculated using a unified regional background reference. For Guangzhou, we replaced the 2010 corn-leaf $\Delta^{14}CO_2$ background (Qinghai–Gansu–

Tibet) used in Ding et al. (2013) and the 2022 Nanling (NL) atmospheric $\Delta^{14}CO_2$ background used in this study with the Nanling (NL) tree-ring $\Delta^{14}CO_2$ record (Li et al., 2025), ensuring a consistent background domain for both 2010 and 2022. For Beijing and Xi'an, all datasets were standardized to the corresponding continental background stations (WLG).

3. **Consideration of sampling-time differences**

   For Guangzhou, we explicitly quantified the afternoon–evening sampling bias using concurrent CO observations. The analysis (see response to l.299–303 and Appendix H1) shows that 20:00 local-time $C_{ff}$ levels are typically 21–35 % higher than those observed during the well-mixed afternoon period, consistent with nighttime boundary-layer accumulation. After correcting for this diurnal contrast, the Guangzhou $C_{ff}$ decreased from $27.3 \pm 16.9 \, \mu mol \, mol^{-1}$ (2010) to $11.6 \pm 3.4 \, \mu mol \, mol^{-1}$ (2022), representing a 34–46 % reduction even after harmonizing sampling time and background.

4. **Assessment of transport representativeness**

   We conducted a transport representativeness check (Sect. 3.4.1 and Appendix G) using ERA5 meteorology and HYSPLIT back trajectories to confirm that the sampling months (August and December 2022) represent typical summer and winter monsoon conditions, respectively. This minimizes the influence of atypical atmospheric transport on the inferred interannual trend.

5. **Rationale for interpreting $C_{ff}$ trends**

   While we fully acknowledge that $CO_{2ff}$ mole fractions reflect both transport and emissions, the use of (i) identical sites, (ii) consistent background definitions, (iii) harmonized sampling times, and (iv) verified meteorological representativeness ensures that the observed multi-year differences primarily reflect emission changes rather than transport artifacts. Furthermore, the inclusion of CO-based diagnostics and $R_{CO/CO2ff}$ ratios provides independent checks that partially cancel transport effects, further supporting the robustness of the trend interpretation.

6. **Clarifications added to the manuscript**

   We have clarified these methodological harmonizations and added two new tables (Tables H1 and H2) to clearly distinguish the full dataset coverage from the standardized subset used for interannual comparison.

**In summary**, we acknowledge that direct comparison of $C_{ff}$ mole fractions requires careful control of both transport and site representativeness. By standardizing locations, seasons, sampling times, and background references—and by confirming the meteorological typicality of the 2022 campaigns—we ensure that our interannual comparisons are methodologically valid and that the observed $C_{ff}$ reductions most likely reflect genuine decreases in fossil-fuel emissions.

The examination of $CO:CO_2ff$ ratios is more compelling, and the authors could reformulate a paper that removes the problematic points and focuses on these results.

**Response:** We sincerely thank the reviewer for recognizing the strength of our $CO:CO_{2ff}$ ratio analysis. We agree that this part of the study provides one of the most robust and informative insights into fossil-fuel emission characteristics. In response to the reviewer's earlier suggestions, we have expanded this section in the revised manuscript by adding further discussion of the seasonal variability, source implications, and intercomparison with previous urban studies.

We also appreciate the reviewer's suggestion that this topic could serve as the focus of a reformulated paper. In the current version, we have emphasized the importance of the $CO:CO_{2ff}$ results within the broader context of our $\Delta^{14}CO_2$-based source partitioning framework and highlighted their significance for understanding fossil-fuel combustion

patterns in Chinese cities.

In addition, we sincerely thank the editor and reviewers again for their valuable time and insightful comments. Beyond addressing all technical and scientific concerns, we have also carefully revised the manuscript for clarity, conciseness, and linguistic accuracy. Furthermore, we have corrected and clarified the description of biomass burning emissions derived from the CASA-GFED4s and EDGAR datasets to ensure consistency with their respective methodological definitions. We believe these comprehensive revisions have substantially improved the quality and readability of the manuscript.

Sincerely,
Gan Zhang
On behalf of all authors

**References**

Graven, H., Fischer, M. L., Lueker, T., Jeong, S., Guilderson, T. P., Keeling, R. F., Bambha, R., Brophy, K., Callahan, W., Cui, X., Frankenberg, C., Gurney, K. R., LaFranchi, B. W., Lehman, S. J., Michelsen, H., Miller, J. B., Newman, S., Paplawsky, W., Parazoo, N. C., Sloop, C., and Walker, S. J.: Assessing fossil fuel $CO_2$ emissions in California using atmospheric observations and models, Environ. Res. Lett., 13, 065007, 10.1088/1748-9326/aabd43, 2018.

Kuderer, M., Hammer, S., and Levin, I.: The influence of 14CO2 releases from regional nuclear facilities at the Heidelberg 14CO2 sampling site (1986–2014), Atmos. Chem. Phys., 18, 7951-7959, 10.5194/acp-18-7951-2018, 2018.

Levin, I., Kromer, B., Schmidt, M., and Sartorius, H.: A novel approach for independent budgeting of fossil fuel $CO_2$ over Europe by $^{14}CO_2$ observations, Geophys. Res. Lett., 30, 2194, 2003.

Li, J., Wei, N., Wang, X., Li, P., Sun, Y., Feng, W., Cheng, Z., Zhu, S., Wang, W., Chen, D., Zhao, S., Zhong, G., Zhou, G., Li, J., and Zhang, G.: Continental-scale impact of bomb radiocarbon affects historical fossil fuel carbon dioxide reconstruction, Communications Earth & Environment, 6, 603, 10.1038/s43247-025-02532-6, 2025.

Li, Q., Guo, X., Zhai, W., Xu, Y., and Dai, M.: Partial pressure of CO2 and air-sea CO2 fluxes in the South China Sea: Synthesis of an 18-year dataset, Progress in Oceanography, 182, 102272, https://doi.org/10.1016/j.pocean.2020.102272, 2020.

Turi, G., Lachkar, Z., and Gruber, N.: Spatiotemporal variability and drivers of $p$CO$_2$ and air–sea CO$_2$ fluxes in the California Current System: an eddy-resolving modeling study, Biogeosciences, 11, 671-690, 10.5194/bg-11-671-2014, 2014.

Turnbull, J., Rayner, P., Miller, J., Naegler, T., Ciais, P., and Cozic, A.: On the use of $^{14}CO_2$ as a tracer for fossil fuel $CO_2$: Quantifying uncertainties using an atmospheric transport model, J. Geophys. Res. Atmos., 114, D22302, 2009.

Turnbull, J. C., Tans, P. P., Lehman, S. J., Baker, D., Conway, T. J., Chung, Y. S., Gregg, J., Miller, J. B., Southon, J. R., and Zhou, L.-X.: Atmospheric observations of carbon monoxide and fossil fuel $CO_2$ emissions from East Asia, J. Geophys. Res. Atmos., 116, D24306, https://doi.org/10.1029/2011JD016691, 2011.

Wang, P., Zhou, W., Xiong, X., Wu, S., Niu, Z., Cheng, P., Du, H., and Hou, Y.: Stable carbon isotopic characteristics of

fossil fuels in China, Sci. Total Environ., 805, 150240, https://doi.org/10.1016/j.scitotenv.2021.150240, 2022a.

Wang, P., Zhou, W., Xiong, X., Wu, S., Niu, Z., Yu, Y., Liu, J., Feng, T., Cheng, P., Du, H., Lu, X., Chen, N., and Hou, Y.: Source attribution of atmospheric $CO_2$ using $^{14}C$ and $^{13}C$ as tracers in two Chinese megacities during winter, J. Geophys. Res. Atmos., 127, e2022JD036504, https://doi.org/10.1029/2022JD036504, 2022b.

Zazzeri, G., Acuña Yeomans, E., and Graven, H. D.: Global and Regional Emissions of Radiocarbon from Nuclear Power Plants from 1972 to 2016, Radiocarbon, 60, 1067-1081, 10.1017/RDC.2018.42, 2018.

Zhang, J., Liang, Y., Pei, C., Huang, B., Huang, Y., Lian, X., Song, S., Cheng, C., Wu, C., Zhou, Z., Li, J., and Li, M.: Atmospheric CO2 dynamics in a coastal megacity: spatiotemporal patterns, sea-land breeze impacts, and anthropogenic-biogenic emission partitioning, EGUsphere, 2025, 1-30, 10.5194/egusphere-2025-3215, 2025.